# IPBench: Benchmarking the Knowledge of Large Language Models in Intellectual Property

## Abstract

Intellectual Property (IP) is a highly specialized domain that integrates technical and legal knowledge, making it inherently complex and knowledge-intensive. Recent advancements in LLMs have demonstrated their potential to handle IP-related tasks, enabling more efficient analysis, understanding, and generation of IP-related content. However, existing datasets and benchmarks focus narrowly on patents or cover limited aspects of the IP field, lacking alignment with real-world scenarios. To bridge this gap, we introduce **IPBench**, the first comprehensive IP task taxonomy and a large-scale bilingual benchmark encompassing **8 IP mechanisms and 20 distinct tasks**, designed to evaluate LLMs in real-world IP scenarios. We benchmark **17 main LLMs**, ranging from general purpose to domain-specific, including chat-oriented and reasoning-focused models, under zero-shot, few-shot, and chain-of-thought settings. Our results show that even the top-performing model, DeepSeek-V3, achieves only 75.8% accuracy, indicating significant room for improvement. Notably, open-source IP and law-oriented models lag behind closed-source general-purpose models. To foster future research, we publicly release IPBench, and will expand it with additional tasks to better reflect real-world complexities and support model advancements in the IP domain. We provide the data and code in the supplementary materials.

## 1 Introduction

Intellectual property (IP) is the embodiment of human creativity and innovation (WIPO, 2020a) protected through legal frameworks such as patents, copyrights, and trademarks. Owing to its intersection of technical and legal domains, IP-related tasks are inherently knowledge-intensive, highly applicable to real-world scenarios, and hold substantial practical value. Beyond domain-specific expertise, these tasks demand robust capabilities in information processing, logical reasoning, decision-making, and creative generation.

With the advancement of large language models (LLMs) (Achiam et al., 2023; DeepSeek-AI et al., 2024), there is increasing potential to automate tasks across domains, including those in IP. LLMs offer a generalizable framework for understanding, processing, and generating complex content, paving the way for more efficient IP information management and decision support. Nowadays, NLP researchers have been paying increasing attention to the field of intellectual property. This has spurred growing interest among NLP researchers in IP applications. For example, Jiang & Goetz (2024) provide a comprehensive survey of patent-related NLP tasks, classifying them into analysis and generation categories, but their focus is limited to patent text.

Recent efforts have introduced datasets such as HUPD (Suzgun et al., 2023), which compiles a corpus of patent and defines tasks including subject classification, language modeling, and summarization. While practically useful, HUPD emphasizes linguistic attributes and neglects the deeper technical and legal aspects essential to IP evaluation. Similarly, benchmarks like PatentEval (Zuo et al., 2024), MoZIP (Ni et al., 2024), and IPEval (Wang et al., 2024b) concentrate on narrow and specific IP task scopes. Moreover, most existing benchmarks center exclusively on patents, leaving other critical IP mechanisms-such as trademarks and copyrights-largely unaddressed. Despite the

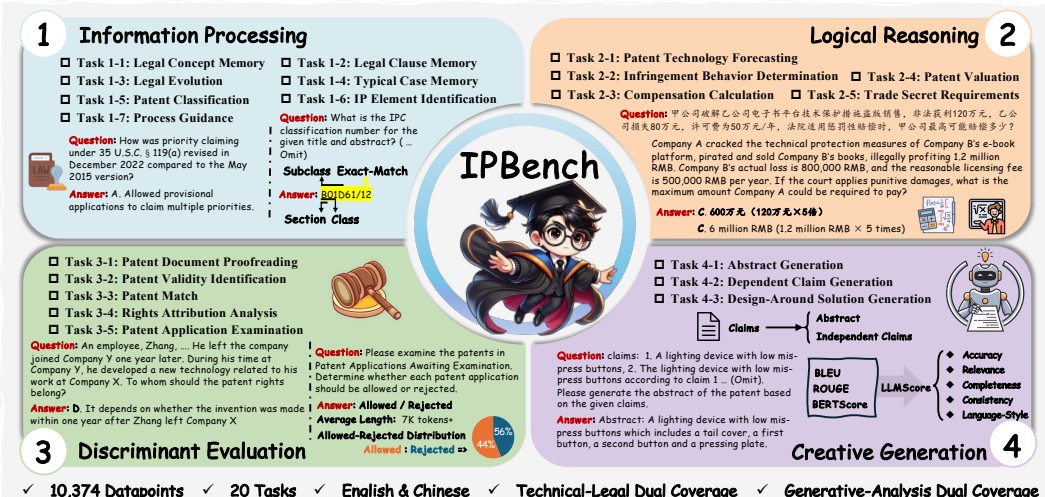

Figure 1: Overview of the comprehensive IP task taxonomy and IPBench.

field's real-world impact, there remains a lack of a comprehensive task taxonomy and benchmark that fully captures the breadth and complexity of IP scenarios.

To address the growing demand for effective LLMs applications in IP, we introduce the first comprehensive task taxonomy tailored to real-world IP challenges, as illustrated in Figure 1. It is grounded in Webb's Depth of Knowledge (DOK) theory (Webb, 2002) and extended to capture four hierarchical levels: *Information Processing*, *Logical Reasoning*, *Discriminant Evaluation*, and *Creative Generation*. These levels reflect the cognitive complexity inherent in IP tasks and provide a structured framework to assess the depth of LLMs understanding. Our taxonomy incorporates intrinsic knowledge evaluation and in-depth textual analysis from both point-wise and pairwise perspectives, covering the interplay between technical and legal reasoning.

Building on this taxonomy, we present **IPBench**, the first large-scale, comprehensive benchmark for evaluating LLMs on IP knowledge. **IPBench** comprises 10,374 data points across 20 diverse tasks, aligned with 8 core IP mechanisms. Our benchmark is bilingual (English and Chinese), and is grounded in the legal frameworks of the United States and mainland China, allowing cross-jurisdictional evaluation. IPBench tasks are carefully designed to span a spectrum of difficulty and task formats, including classification, retrieval, and open-ended generation, enabling holistic evaluation of model capabilities in knowledge recall, reasoning, legal judgment, and creative synthesis. We evaluate 17 leading LLMs on IPBench-including general-purpose models, law-oriented models, and IP-specialized models-covering both chat and reasoning-focused architectures, under zero-shot, few-shot and chain-of-thought settings. Our key contributions and findings are as follows:

- We propose the first hierarchical taxonomy for IP domain, rooted in cognitive theory, and introduce **IPBench**, a bilingual benchmark with 10,374 examples spanning 20 tasks and 8 IP mechanisms. This enables realistic, and multi-faceted evaluation of LLMs in IP contexts.

- Our experimental results reveal that even the best-performing model achieves only 75.8% accuracy overall, indicating that current LLMs fall short in reliably handling IP tasks. Notably, closed-source general-purpose models consistently outperform domain-specific open-source models, highlighting a pressing need for more capable and interpretable IP-focused LLMs.

- We include both IPC/CPC classification and conditional generation tasks in IPBench. DeepSeek-R1 achieves the best IPC classification accuracy at 10.8%, while DeepSeek-V3 leads in CPC classification at 9.5%. For generative tasks, we introduce *LLMScore*, a novel evaluation metric based on LLM-as-a-judge methodology, which exhibits stronger alignment with human judgments than traditional automatic metrics.

- We conduct comprehensive analyses, including cross-lingual performance comparisons, variations in prompt design, and a taxonomy of 7 major error types.

We believe **IPBench** offers a timely and essential tool for advancing the application of LLMs in IP. From a machine learning perspective, the complexity of IP language can serve as a robust stress test for LLMs. From a legal and innovation standpoint, automation in this domain can enhance service intelligence, reduce operational costs, and ultimately accelerate global technological advancement. We plan to continuously expand IPBench by incorporating additional languages, modalities, and tasks in future iterations.

## 2 RELATED WORK

Prior to the emergence of LLMs, researchers applied NLP techniques IP tasks, particularly within the domain of patent analysis. These efforts focused on applications such as patent classification (Lee & Hsiang, 2020), and abstract or claims generation (Sharma et al., 2019; Lee & Hsiang, 2020). However, traditional models used in these studies typically lacked generalization capabilities and required extensive task-specific adaptation, limiting their scalability and real-world applicability. With the advent of LLMs based on the decoder-only transformer architecture (Radford et al., 2019), models trained using next-token prediction have demonstrated impressive zero-shot (Kojima et al., 2022) and few-shot (Brown et al., 2020) capabilities across diverse tasks. This paradigm shift introduced a new approach to handling IP-related challenges using prompt-based inference, reducing the need for task-specific training and enabling more versatile applications in the IP domain.

Recent work has explored the adaptation of LLMs specifically for IP. Ni et al. (2024) developed MoZi, a multilingual IP-oriented LLM based on BLOOMZ and ChatGLM. Bai et al. (2024) proposed a cost-efficient training framework to fine-tune LLMs for IP tasks, claiming performance on par with human experts. Other studies, such as Pap2Pat (Knappich et al., 2024), AutoPatent (Wang et al., 2024b), and PatentFormer (Wang et al., 2024a), focus on long-context generation for patent documents using LLMs. These works predominantly emphasize the technical aspects of patent language and overlook broader IP mechanisms, such as trademarks, trade secrets, and copyrights. Moreover, they rarely consider legal reasoning and decision-making, which are essential for real-world applications.

Table 1: Comparison of IP related benchmark with Ours. *Gen.-Ana. Dual Cover.* refers to benchmarks that encompass both text generation and analysis tasks. *Tech.-Legal Dual Cover.* refers to benchmarks that contain both aspects of technical and legal content. Meanwhile, *Compre. Taxonomy* refers to a benchmark that possesses a comprehensive taxonomy.

| Benchmark | PatentEval | IPEval | MoZIP | Ours |
|---|---|---|---|---|
| Evaluation for LLMs | ✓ | ✓ | ✓ | ✓ |
| Multilingual | | ✓ | ✓ | ✓ |
| Multi-IP Mechanisms | | ✓ | ✓ | ✓ |
| Tech.-Legal Dual Cover. | | | ✓ | ✓ |
| Gen.-Ana. Dual Cover. | | | | ✓ |
| Compre. Taxonomy | | | | ✓ |
| LLMs Evaluated # | 6 | 15 | 5 | 17 ⋆ |
| Task # | 2 | 1 | 3 | 20 ⋆ |
| Testset Size | 400 | 2657 | 3121 | 10374 ⋆ |

Our work differs fundamentally in both scope and design. IPBench builds upon and expands these earlier efforts by introducing a unified, comprehensive IP task taxonomy grounded in Webb's Depth of Knowledge (DOK) theory. Notably, we include under-explored areas such as trade secret and trademark, offering a holistic evaluation of LLM performance across the IP landscape. This makes IPBench more comprehensive than prior benchmarks like IPEval, MoZIP, and PatentEval. A detailed comparison is presented in Table 1, highlighting our benchmark's task diversity, linguistic coverage, and legal granularity.

## 3 IPBENCH

### 3.1 TASK TAXONOMY

While previous patent-related benchmarks have primarily focused on textual content such as classification or summarization they often overlook the broader real-world implications of IP tasks. To address this gap, we introduce the first comprehensive intellectual property task taxonomy that extends beyond in-domain textual analysis to encompass the multifaceted real-world demands of the

Table 2: Task taxonomy of IPBench. The *EN* in the Language column indicates English, while *ZH* represents Chinese. The *AE* in the Metric column indicates Automated Evaluation, while *HE* represents Human Evaluation.

| Level | Index | Task Name | Metric | Data Source | Language | Size |
|---|---|---|---|---|---|---|
| Information Processing | 1-1 | Legal Concept Memory | Accuracy | Expert Annotation | EN/ZH | 500 |
| | 1-2 | Legal Clause Memory | Accuracy | Expert Annotation | EN/ZH | 502 |
| | 1-3 | Legal Evolution | Accuracy | Expert Annotation | EN/ZH | 500 |
| | 1-4 | Typical Case Memory | Accuracy | USTPO / CNIPA | EN/ZH | 504 |
| | 1-5-1 | Patent IPC Classification | Exact Match | USTPO / CNIPA | EN/ZH | 1125 |
| | 1-5-2 | Patent CPC Classification | Exact Match | USTPO | EN | 600 |
| | 1-6 | IP Element Identification | Accuracy | Expert Annotation | EN/ZH | 557 |
| | 1-7 | Process Guidance | Accuracy | Expert Annotation | EN/ZH | 548 |
| Logical Reasoning | 2-1 | Patent Technology Forecasting | Accuracy | Expert Annotation | EN/ZH | 500 |
| | 2-2 | Infringement Behavior Determination | Accuracy | Expert Annotation | EN/ZH | 500 |
| | 2-3 | Compensation Calculation | Accuracy | Expert Annotation | EN/ZH | 316 |
| | 2-4 | Patent Valuation | Accuracy | Expert Annotation | EN/ZH | 301 |
| | 2-5 | Trade Secret Requirements | Accuracy | Expert Annotation | ZH | 301 |
| Discriminant Evaluation | 3-1 | Patent Document Proofreading | Accuracy | Expert Annotation | EN/ZH | 300 |
| | 3-2 | Patent Validity Identification | Accuracy | Expert Annotation | EN/ZH | 308 |
| | 3-3 | Patent Match | Accuracy | MoZIP | EN/ZH | 1000 |
| | 3-4 | Rights Attribution Analysis | Accuracy | Expert Annotation | EN/ZH | 400 |
| | 3-5 | Patent Application Examination | Accuracy | USTPO | EN | 314 |
| Creative Generation | 4-1 | Abstract Generation | AE & HE | USTPO / CNIPA | EN/ZH | 400 |
| | 4-2 | Dependent Claim Generation | AE & HE | USTPO / CNIPA | EN/ZH | 400 |
| | 4-3 | Design-Around Solution Generation | Accuracy | Expert Annotation | EN/ZH | 499 |

IP field, spanning both technical and legal dimensions. Given the intrinsic complexity of IP knowledge, effective modeling in this domain requires more than domain-specific understanding. LLMs must be capable of integrating diverse IP mechanisms, simulating real-world procedural reasoning, and interpreting varied linguistic styles present in different IP documents and legal jurisdictions. This necessitates a structured evaluation framework that captures different levels of cognitive depth and reasoning complexity.

To this end, our taxonomy is grounded in the Depth of Knowledge (DOK) theory by American educator Norman L. Webb, which categorizes cognitive complexity into four levels: *Recall and Reproduction*, *Skills and Concepts*, *Strategic Thinking*, and *Extended Thinking*. Originally developed to guide educational assessment, this framework aligns well with the stratified nature of IP reasoning. We adapt and reinterpret DOK into a legal and technical context, resulting in four hierarchical levels tailored for IP evaluation: **Information Processing**, **Logical Reasoning**, **Discriminant Evaluation**, and **Creative Generation**, as illustrated in Figure 1. These levels enable us to map tasks to specific reasoning capacities required by LLMs, ranging from simple fact recall to complex synthesis and decision-making. The taxonomy provides a principled foundation for evaluating LLMs not only in terms of accuracy but also cognitive depth and functional applicability. Table 2 summarizes the 20 tasks included in IPBench and we also provide further details on the task taxonomy, along with comprehensive definitions of each task, in Appendix D.

## 3.2 DATA PROCESSING AND ANNOTATION

**Data Source and Collection.** Our dataset is constructed from three primary sources: expert-curated annotations, databases maintained by national IP offices, and previously published public datasets. This diverse sourcing approach ensures broad coverage of real-world scenarios and IP mechanisms. For tasks grounded in statutory interpretation-such as *Legal Concept Memory*-data are drawn from official legal texts and documentation published on the public websites of IP offices, including the United States and China. For litigation-oriented tasks-such as *Infringement Behavior Determination*-we utilize publicly available judicial decisions, including case repositories such as China Judgements Online. Patent-related tasks leverage structured data from the USPTO and the China National Intellectual Property Administration (CNIPA). All sources used in IPBench are publicly accessible, ensuring transparency and reproducibility.

**Data Processing and Annotation.** Our IPBench is constructed as a gold-standard benchmark through extensive human expert annotation. Given the highly structured nature of patent documents,

both the USPTO and CNIPA datasets offer well-organized metadata-enabling the systematic creation of paired inputs, such as sequential claim pairs that reflect logical progression in legal language. To ensure annotation quality and domain relevance, we engaged 21 trained annotators, including senior undergraduate and PhD students, all supervised by four certified and experienced patent agents. Most annotators hold academic backgrounds in IP, equipping them with foundational knowledge of both technical and legal aspects of IP. This subject matter expertise was critical to generating high-fidelity annotations across legal, technical, and procedural tasks.

The annotation team is organized into four subgroups, each dedicated to one of the hierarchical levels in our taxonomy. Each task underwent a rigorous two-stage workflow: one team conducted the initial annotation while another team reviewed and validated the results. The roles were then rotated to ensure objectivity and consistency across all data points. Following annotation, we perform automatic quality filtering using cosine similarity based on the BGE-M3 model (Chen et al., 2024). This step eliminate semantically redundant examples and further enhanced the dataset's diversity and representativeness. Our complete annotation and examination protocol is in Appendix E.

### 3.3 FEATURE OF IPBENCH

IPBench consists of 10,374 expertly curated questions spanning 20 tasks. These tasks are systematically organized across 4 hierarchical levels and cover 8 IP mechanisms, including patents and trade secrets, etc. The benchmark integrates both technical and legal domains and includes a mix of task formats, ranging from classification and comprehension to open-ended generative reasoning. This diverse coverage enables comprehensive evaluation of LLM capabilities, including factual recall, legal reasoning, procedural understanding, and content synthesis. As shown in Table 1, IPBench surpasses existing IP benchmarks across multiple dimensions, including task diversity, jurisdictional representation, cognitive complexity, and linguistic variation.

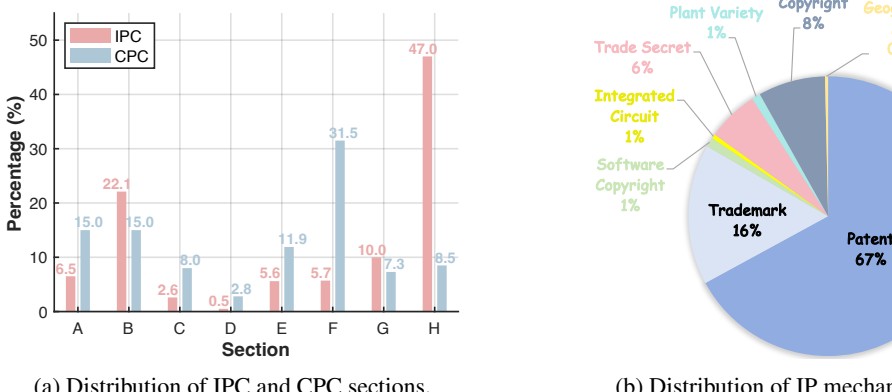

(a) Distribution of IPC and CPC sections.    (b) Distribution of IP mechanisms.

Figure 2: Distributions across IPC/CPC sections and IP mechanisms.

Given the wide scope of IP mechanisms and technical domains represented, we provide detailed statistical analysis of IPBench's data characteristics. These include the distributions of International Patent Classification (IPC) and Cooperative Patent Classification (CPC) codes, as shown in Figure 2a, and IP mechanisms, as shown in Figure 2b. More feature details of IPBench can be found in Appendix F including the distributions of IPC/CPC classification codes, text lengths, and domain coverage.

## 4 BENCHMARKING RESULTS

### 4.1 EVALUATION SETUP

**Evaluated Models.** We evaluate 17 language models covering a broad range of sizes, architectures, and domain specializations, with details provided in Appendix I. Among them, 14 are general-purpose large language models, 2 are law-oriented models specifically fine-tuned for legal tasks, and one is an IP-oriented model developed for intellectual property applications.

Table 3: Main results of IPBench. The best-performing model in each task is in darker red , and the second best is in lighter red . The model DS-Qwen refers to DeepSeek-R1-Distill-Qwen.

| Model | OA | 1-1 | 1-2 | 1-3 | 1-4 | 1-6 | 1-7 | 2-1 | 2-2 | 2-3 | 2-4 | 2-5 | 3-1 | 3-2 | 3-3 | 3-4 | 3-5 | 4-3 |
|---|---|---|---|---|---|---|---|---|---|---|---|---|---|---|---|---|---|---|
| GPT-4o | 75.3 | 96.0 | 92.0 | 82.2 | 83.7 | 64.2 | 71.9 | 54.8 | 62.6 | 63.9 | 78.5 | 84.1 | 71.0 | 70.1 | 81.3 | 83.5 | 50.0 | 75.4 |
| GPT-4o-mini | 72.6 | 94.4 | 87.5 | 80.2 | 82.1 | 58.8 | 67.5 | 50.2 | 64.0 | 59.5 | 76.7 | 83.4 | 67.3 | 75.0 | 81.6 | 78.5 | 44.0 | 66.3 |
| DeepSeek-V3 | 75.8 | 96.6 | 90.2 | 88.4 | 82.8 | 66.1 | 69.9 | 56.8 | 64.2 | 66.1 | 76.7 | 84.1 | 72.0 | 75.0 | 78.9 | 83.5 | 44.6 | 78.8 |
| Qwen3 | 70.6 | 94.4 | 83.1 | 75.0 | 76.6 | 60.9 | 66.8 | 51.4 | 66.8 | 60.4 | 75.1 | 82.7 | 69.7 | 74.4 | 70.5 | 78.0 | 44.0 | 67.9 |
| Qwen2.5-72B-it | 74.7 | 96.0 | 90.4 | 84.2 | 83.5 | 61.3 | 69.2 | 54.4 | 66.6 | 63.0 | 80.4 | 82.1 | 71.7 | 73.4 | 79.9 | 80.7 | 43.3 | 75.3 |
| Qwen2.5-7B-it | 68.0 | 92.4 | 83.3 | 77.2 | 77.2 | 58.4 | 62.0 | 49.4 | 64.4 | 57.3 | 74.4 | 77.1 | 67.7 | 71.1 | 65.8 | 78.2 | 38.9 | 58.9 |
| Llama3.1-70B-it | 70.5 | 93.8 | 85.3 | 77.6 | 79.8 | 59.3 | 67.0 | 53.0 | 64.8 | 53.5 | 74.8 | 81.1 | 70.3 | 74.4 | 67.1 | 78.0 | 45.2 | 71.3 |
| Llama3.1-8B-it | 61.7 | 90.4 | 75.9 | 68.2 | 71.3 | 53.0 | 60.4 | 47.6 | 57.5 | 44.6 | 71.4 | 75.7 | 60.0 | 61.7 | 50.6 | 77.2 | 41.7 | 52.3 |
| Gemma-2-27B-it | 68.1 | 90.6 | 80.5 | 73.2 | 77.6 | 54.5 | 61.3 | 53.4 | 65.0 | 56.0 | 76.4 | 81.1 | 69.3 | 66.2 | 57.2 | 80.2 | – | 66.9 |
| Gemma-2-9B-it | 64.9 | 91.6 | 78.3 | 73.0 | 61.5 | 58.8 | 59.3 | 51.2 | 63.6 | 46.8 | 70.4 | 80.4 | 66.0 | 66.9 | 51.9 | 76.0 | – | 62.1 |
| Mistral-7B-it | 54.7 | 79.6 | 63.9 | 60.6 | 60.1 | 40.5 | 54.0 | 43.6 | 56.0 | 42.4 | 64.1 | 67.0 | 56.0 | 45.8 | 43.9 | 65.1 | 43.9 | 54.5 |
| MoZi-qwen | 64.9 | 93.8 | 83.3 | 77.0 | 66.1 | 58.2 | 64.2 | 50.6 | 58.0 | 41.8 | 67.8 | 76.4 | 68.0 | 64.3 | 56.1 | 79.0 | 43.9 | 57.1 |
| DISC-LawLLM | 52.8 | 79.0 | 65.3 | 67.6 | 60.1 | 54.5 | 52.0 | 40.8 | 60.4 | 31.3 | 60.1 | 64.8 | 53.7 | 45.1 | 28.2 | 71.2 | – | 35.3 |
| Hanfei | 40.1 | 63.0 | 46.4 | 51.8 | 45.4 | 39.8 | 47.3 | 30.8 | 45.6 | 33.9 | 40.9 | 49.2 | 42.7 | 28.6 | 18.9 | 48.8 | – | 29.5 |
| DeepSeek-R1 | 73.9 | 96.0 | 92.0 | 87.6 | 80.8 | 64.9 | 71.7 | 53.6 | 64.6 | 71.8 | 78.1 | 85.4 | 63.3 | 78.2 | 67.2 | 82.0 | 47.5 | 74.3 |
| DS-Qwen-7B | 57.0 | 77.8 | 59.0 | 53.8 | 57.1 | 49.8 | 50.7 | 43.8 | 51.2 | 46.2 | 67.1 | 65.5 | 54.0 | 62.0 | 63.7 | 63.7 | 43.6 | 54.9 |
| QwQ-32B | 73.5 | 95.2 | 91.0 | 81.8 | 77.8 | 65.1 | 71.5 | 57.4 | 66.6 | 70.6 | 80.1 | 85.4 | 69.7 | 82.1 | 67.3 | 77.0 | 47.1 | 69.7 |

**Experimental Settings.** Inspired by previous benchmarks Team et al. (2025), we adopt five distinct evaluation settings for chat models: zero-shot, 1-shot, 2-shot, 3-shot, and Chain-of-Thought (CoT). For reasoning models, we use only the zero-shot setting to ensure a fair comparison given their limited prompt-handling flexibility. In few-shot settings, we randomly sample one to three in-context examples (excluding the current test instance) using a fixed seed to ensure reproducibility. To ensure consistency and reproducibility, we set the temperature to 0.0 across all experiments. The maximum input token limit is capped at 32k for reasoning models and 8k for chat models; for models with shorter context windows, we use the maximum supported length. All the prompts used are provided in Appendix G.

**Metrics.** We use accuracy as the primary evaluation metric for the most tasks. For IPC and CPC classification tasks, we follow the evaluation strategy of HELM (Liang et al., 2022), using Exact Match at **different granularity levels**: *Section*, *Class*, and *Subclass*. For generative tasks such as abstract and claim generation, we evaluate model outputs using the F1 score of metrics: BLEU (Papineni et al., 2002), ROUGE-L (Lin, 2004), and BERTScore (Zhang et al., 2019). Additionally, inspired by the fine-grained error taxonomy in PatentEval, we propose *LLMScore*, a multi-dimensional, automatic evaluation metric aligned with the LLM-as-a-judge paradigm (Liu et al., 2023; Li et al., 2025a). LLMScore is used to assess the semantic and structural quality of generated outputs, and we validate its consistency against human judgment. Details of metrics are provided in Appendix H.

### 4.2 MAIN RESULTS

As shown in Table 3, 4, and 5, we present the main results under the zero-shot setting, while results for the few-shot and CoT setting are provided in Figure 4a. More comprehensive results of IPBench can be found in Appendix K.

### 4.3 ANALYSIS

**Disparity between IP-oriented and general-purpose models.** Surprisingly, general-purpose models consistently outperform both law-oriented and IP-oriented models on IPBench. Although MoZi-qwen, an IP-oriented model, outperforms the 2 law-oriented models DISC-LawLLM and Hanfei, it still trails Qwen2.5-7B-it, by 3.1%. These results underscore a recurring issue in vertical domain models: despite being optimized for specific applications, they tend to underperform on domain-specific evaluations (Wang et al., 2024b; Hou et al., 2024; Li et al., 2024). This suggests that domain-specific models must adopt improved strategies for learning domain knowledge without sacrificing general-purpose capabilities.

**Model performance across different languages.** Model performance correlates strongly with the primary training language of the model. Results across the Chinese and English subsets of IP-

Table 4: Main results of IPC/CPC Classification tasks. The best-performing model is in darker purple , and the second best is in lighter purple .

| Model | IPC Classification (1-5-1) | | | | CPC Classification (1-5-2) | | | |
|---|---|---|---|---|---|---|---|---|
| | Exact-Match | Section | Class | Subclass | Exact-Match | Section | Class | Subclass |
| GPT-4o | 4.8 | 81.6 | 71.3 | 55.1 | 3.3 | 82.7 | 69.7 | 62.0 |
| GPT-4o-mini | 1.0 | 80.5 | 66.8 | 50.1 | 0.5 | 79.0 | 64.5 | 52.7 |
| DeepSeek-V3 | 10.6 | 83.7 | 73.3 | 58.3 | 9.5 | 84.0 | 73.3 | 65.2 |
| Qwen3 | 2.8 | 80.6 | 64.8 | 48.0 | 0.5 | 62.7 | 48.3 | 38.7 |
| Qwen2.5-72B-it | 4.9 | 82.4 | 70.4 | 55.2 | 2.5 | 81.5 | 69.5 | 60.7 |
| Qwen2.5-7B-it | 1.9 | 76.8 | 63.0 | 46.6 | 0.2 | 65.5 | 44.8 | 34.8 |
| Llama3.1-70B-it | 3.5 | 80.4 | 65.6 | 50.0 | 1.0 | 79.5 | 64.3 | 52.7 |
| Llama3.1-8B-it | 0.9 | 71.8 | 56.2 | 35.8 | 0.0 | 63.8 | 45.0 | 30.7 |
| Gemma-2-27B-it | 1.2 | 72.9 | 57.4 | 41.5 | 0.2 | 70.5 | 56.7 | 44.3 |
| Gemma-2-9B-it | 0.3 | 73.7 | 55.6 | 37.2 | 0.2 | 56.2 | 39.0 | 26.7 |
| Mistral-7B-it | 0.1 | 67.2 | 42.8 | 26.8 | 0.0 | 39.0 | 21.5 | 10.3 |
| MoZi-qwen | 0.6 | 38.8 | 29.6 | 20.3 | 0.0 | 8.5 | 3.1 | 1.8 |
| DISC-LawLLM | 0.0 | 68.2 | 47.2 | 28.3 | 0.0 | 31.0 | 23.4 | 11.5 |
| Hanfei | 0.0 | 11.7 | 2.0 | 0.1 | 0.0 | 0.8 | 0.0 | 0.0 |
| DeepSeek-R1 | 10.8 | 85.8 | 74.7 | 59.3 | 8.5 | 82.5 | 71.2 | 63.2 |
| DS-Qwen-7B | 0.0 | 20.5 | 6.9 | 1.4 | 0.0 | 5.1 | 0.5 | 0.2 |
| QwQ-32B | 2.9 | 83.8 | 70.4 | 53.8 | 0.5 | 76.0 | 62.3 | 51.3 |

Table 5: Main results of generation tasks. The best-performing model is in darker blue , and the second best is in lighter blue . R-L refers to ROUGE-L, BS refers to BERTScore, Tokens # denotes the average number of tokens in the generated text, and DC # indicates the average number of generated dependent claims.

| Model | Abstract Generation (4-1) | | | | | Dependent Claim Generation (4-2) | | | | | |
|---|---|---|---|---|---|---|---|---|---|---|---|
| | BLEU | R-L | BS | LLMScore (1-10) | Tokens # (148.5) | BLEU | R-L | BS | LLMScore (1-10) | Tokens # (437.6) | DC # (5.2) |
| GPT-4o | 17.7 | 31.1 | 89.3 | 8.42 | 271.4 | 18.9 | 26.5 | 88.8 | 6.63 | 647.8 | 6.5 |
| GPT-4o-mini | 23.4 | 31.9 | 89.6 | 8.05 | 218.1 | 20.3 | 28.3 | 88.4 | 6.37 | 478.1 | 6.5 |
| DeepSeek-V3 | 19.6 | 28.3 | 89.0 | 8.38 | 246.1 | 19.1 | 26.8 | 89.0 | 7.45 | 691.7 | 14.9 |
| Qwen2.5-72B-it | 21.0 | 30.6 | 89.5 | 8.33 | 326.0 | 10.0 | 17.1 | 89.2 | 6.30 | 3790.9 | 69.1 |
| Qwen2.5-7B-it | 27.3 | 35.7 | 90.2 | 8.18 | 209.2 | 15.1 | 22.3 | 89.2 | 5.67 | 3511.3 | 45.7 |
| Llama3.1-70B-it | 31.0 | 38.2 | 90.4 | 7.98 | 226.5 | 16.0 | 23.8 | 88.1 | 5.67 | 2294.4 | 28.3 |
| Llama3.1-8B-it | 20.1 | 28.4 | 89.2 | 7.47 | 457.3 | 8.1 | 13.9 | 88.4 | 3.86 | 6287.9 | 90.8 |
| Gemma-2-27B-it | 19.7 | 27.5 | 88.9 | 7.64 | 193.3 | 15.2 | 22.6 | 87.3 | 5.98 | 582.3 | 3.3 |
| Gemma-2-9B-it | 21.6 | 29.4 | 89.0 | 7.91 | 219.3 | 14.7 | 23.2 | 87.1 | 5.55 | 511.9 | 6.4 |
| Mistral-7B-it | 20.2 | 27.4 | 89.4 | 7.49 | 361.7 | 7.2 | 11.7 | 88.0 | 3.42 | 6543.1 | 96.3 |
| MoZi-qwen | 31.2 | 51.0 | 90.4 | 7.73 | 316.4 | 16.3 | 34.4 | 89.0 | 4.81 | 5121.5 | 47.7 |
| DeepSeek-R1 | 13.8 | 27.8 | 87.5 | 7.72 | 642.3 | 16.6 | 29.3 | 71.4 | 7.18 | 1302.9 | 19.1 |
| DS-Qwen-7B | 9.7 | 22.9 | 83.6 | 7.58 | 802.5 | 11.7 | 32.4 | 69.0 | 4.16 | 6096.9 | 54.1 |
| QwQ-32B | 16.6 | 32.0 | 87.9 | 8.51 | 1126.6 | 12.6 | 25.8 | 71.9 | 7.10 | 4997.7 | 41.8 |

Bench are provided in Figure 3a. DeepSeek-V3 achieves the highest accuracy on the Chinese subset (78.7%), while GPT-4o leads on the English subset (73.2%). These findings highlight the impact of legal system discrepancies across jurisdictions and the need for language models to recognize and adapt to structural and contextual differences during inference, consistent with the observations reported in IPEval (Wang et al., 2024b).

**Disparity between Chat Model and Reasoning Model.** In addition to chat models, we evaluate 3 reasoning-focused models, notably DeepSeek-R1. While these models do not achieve the highest overall scores, they demonstrate superior performance on logically intensive tasks. For example, in Task 2-3 (compensation calculation), DeepSeek-R1 surpasses the best-performing chat model, DeepSeek-V3, by 5.7%. This task requires not only domain knowledge but also strong arithmetic and logical reasoning skills. These findings highlight the need for future models to integrate both intuitive (**System 1**) and analytical (**System 2**) capabilities, particularly in high-stakes, knowledge-intensive domains such as IP.

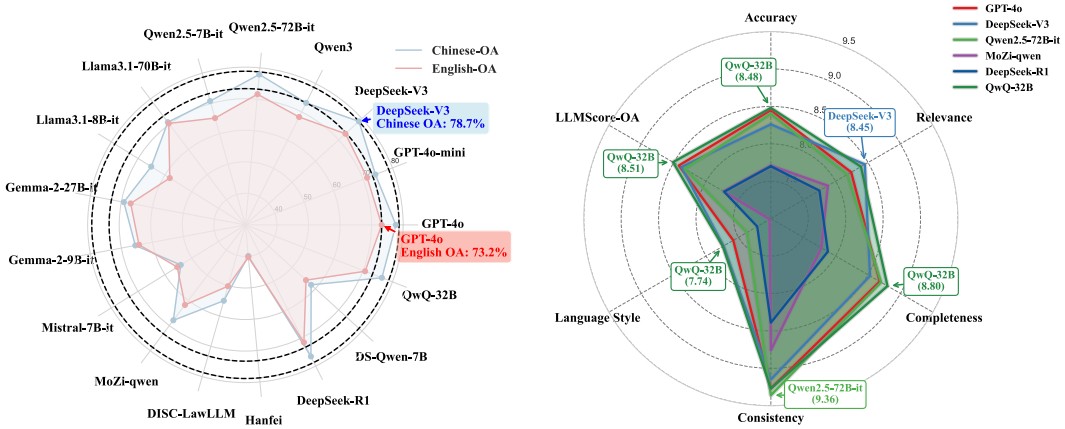

(a) Performance across different languages.

(b) LLM-as-a-judge evaluation across five dimensions.

Figure 3: Comparative results: (a) performance across languages; (b) evaluation across fine-grained dimensions.

**Disaster in IPC/CPC exact match performance.** Performance on IPC/CPC classification tasks remains particularly weak. DeepSeek-R1 achieves the highest Exact Match score at 10.8%, followed by DeepSeek-V3 at 9.5%, while several models score as low as 0.0%. As the classification granularity increases -from Section to Class to Subclass to Exact Match -the difficulty also rises, given the increasingly specific technical distinctions required. These results reveal substantial limitations in current models' abilities to perform fine-grained classification and highlight the complexity of capturing structured taxonomies in patent law. Since IPC/CPC classification underpins many foundational applications in patent management, this represents a critical area for model improvement.

**Lack of fine-grained, interpretable automatic evaluation for IP-related generative tasks.** For these two generative tasks, there is a lack of fine-grained, interpretable automatic evaluation methods to provide more reliable results. Traditional metrics such as BLEU, ROUGE-L, and BERTScore are limited in their effectiveness and exhibit low consistency. To address this issue, we adopt an LLM-as-a-judge approach with five fine-grained dimensions, inspired by PatentEval's error taxonomy, and introduce *LLMScore* for more reliable evaluation. As shown in Table 6, LLMScore demonstrates significantly higher consistency with human judgments than other metrics, which is reflected in its higher Kendall, Spearman, and Pearson correlation coefficients, and lower *p*-values. We present detailed LLM-as-a-judge evaluations of generative tasks across five dimensions: *Accuracy*, *Relevance*, *Completeness*, *Consistency*, and *Language Style*, as illustrated in Figure 3b. Detailed LLMScore results are provided in Appendix K.4.

Table 6: Correlation of LLMScore with human judgments on Task 4-1 and Task 4-2 (*p*-value in parentheses). ↑ Correlation coefficients, ↓ *p*-value.

| Metric | Task 4-1 | | | Task 4-2 | | |
|---|---|---|---|---|---|---|
| | **Kendall** | **Pearson** | **Spearman** | **Kendall** | **Pearson** | **Spearman** |
| LLMScore | **0.22 (0.0005)** | **0.29 (0.0011)** | **0.32 (0.0003)** | **0.40 (0.0000)** | **0.65 (0.0000)** | **0.58 (0.0000)** |
| BLEU | 0.17 (0.0042) | 0.22 (0.0068) | 0.23 (0.0046) | **0.40 (0.0000)** | 0.47 (0.0000) | 0.54 (0.0000) |
| ROUGE-L | 0.15 (0.0123) | 0.18 (0.0317) | 0.20 (0.0154) | 0.37 (0.0000) | 0.51 (0.0000) | 0.50 (0.0000) |
| BERTScore | 0.10 (0.0746) | 0.16 (0.0519) | 0.14 (0.0847) | 0.05 (0.3680) | 0.09 (0.2950) | 0.08 (0.3494) |

**Results and analysis of few-shot prompting.** As shown in Figure 4a, the performance of models on IPBench generally improves as the number of shots increases, reflecting a positive correlation between in-context learning and task performance -except for Llama3.1-8B, which does not exhibit this trend. This observation is consistent with prior studies (Li et al., 2024; Wang et al., 2024b), which show that the effectiveness of few-shot prompting varies significantly across model architec-

tures. These findings suggest that few-shot learning may not be a universally effective strategy for injecting domain-specific knowledge for complex domains.

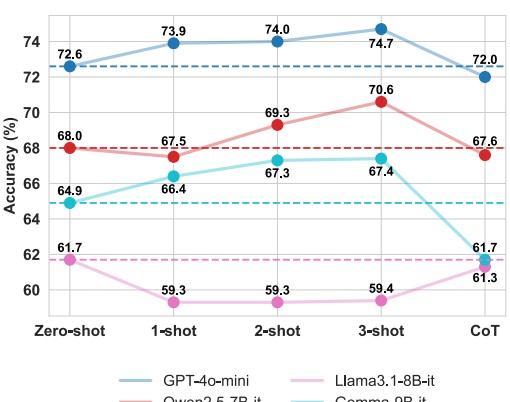

(a) Performance under different prompt settings.

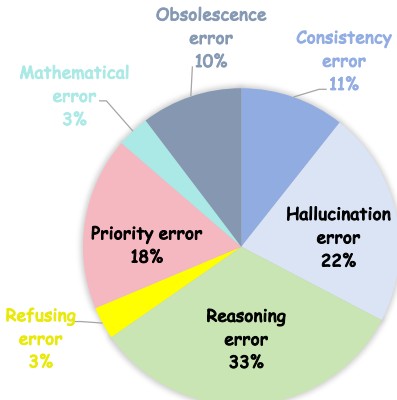

(b) Error distribution of GPT-4o-mini's responses.

Figure 4: Few-shot performance and error distribution.

**Results and analysis of CoT prompting.** As shown in Figure 4a, all models experience a slight decline in performance-ranging from 0.4% to 0.6%-when using CoT prompting. Upon deeper analysis of the error cases, we observe that models generate not only the final answer but also a reasoning trajectory. This additional reasoning, while intended to aid logical flow, often introduces new sources of error or distracts from more intuitive solutions. These results align with recent findings (Zheng et al., 2025; Fan et al., 2025), which suggest that CoT prompting may conflict with the natural inferential preferences of language models -especially in tasks relying more on memorization or domain recall than on abstract reasoning. This is further reflected in the observation that reasoning models do not outperform chat models on IPBench, despite conducting longer reasoning sequences during inference.

## 4.4 ERROR ANALYSIS

To gain deeper insight into model limitations, we perform a qualitative error analysis. We randomly selected 300 incorrect responses generated by GPT-4o-mini under the CoT setting across all IPBench tasks. These samples were manually reviewed and annotated by expert evaluators. As shown in Figure 4b, the errors are categorized into seven types: *Consistency Error*, *Hallucination Error*, *Reasoning Error*, *Refusing Error*, *Priority Error*, *Mathematical Error*, and *Obsolescence Error*. Among these, *Reasoning Error* is the most frequent, accounting for 33% of the total. This error analysis is crucial for gaining deeper insights into the model's capabilities in the IP domain and for revealing potential directions for future research. More details of error analysis and case study are in Appendix L,M and N. We also provide more discuss and limitations in Appendix J, B.

## 5 CONCLUSION

We introduce the first comprehensive IP task taxonomy and present IPBench, a bilingual benchmark comprising 20 tasks and 10,374 test instances, covering both technical-legal and generation-comprehension evaluations. Our experiments show that even the best-performing model, DeepSeek-V3, achieves only a 75.8% score. We observe that current models, including IP-oriented ones, still lag significantly behind powerful closed-source models, highlighting the need for improved domain-specific learning approaches. Our extensive performance analysis, error analysis and case study provide a comprehensive insight in models' IP knowledge and capabilities. We are committed to continuously expanding IPBench to foster advancements in both the IP domain and NLP research, providing meaningful guidance for the integration of LLMs into specialized vertical fields.

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

# APPENDIX

## A   THE USE OF LARGE LANGUAGE MODELS (LLMS)

In accordance with the policy on the use of Large Language Models, we clarify that in this work LLMs were employed exclusively for improving the presentation of the manuscript, such as correcting grammatical errors, enhancing clarity, and refining writing. The research design, conceptual development, and analytical contributions were made solely by the authors.

## B   LIMITATIONS

While IPBench represents a significant step forward in evaluating large language models for intellectual property tasks, several limitations remain.

First, due to the jurisdiction-specific nature of intellectual property law, the current version of IP-Bench focuses primarily on the legal frameworks of the United States and mainland China. This restricts its global applicability, as key differences in legal definitions, procedural structures, and enforcement standards exist across countries. Expanding the benchmark to include legal systems from jurisdictions such as the European Union, Japan, and Korea would enhance its cross-cultural robustness and relevance.

Second, resource constraints limited our evaluation to four reasoning models. While these include some of the most advanced publicly available systems, we were unable to include proprietary models such as OpenAI's o1 (Jaech et al., 2024) and o3 series due to prohibitive costs. As pricing structures evolve and research access improves, future iterations of IPBench will aim to incorporate a wider array of state-of-the-art reasoning models.

Third, intellectual property remains an underexplored vertical domain in large language model research. Currently, MoZi is the only publicly available IP-specific model, and thus the only one benchmarked in this study. The development and release of more open-source IP-oriented models will be essential for driving progress in this field and enabling more comprehensive comparisons in future studies.

Lastly, although we propose *LLMScore*, a fine-grained, interpretable, and high-consistency evaluation metric grounded in the LLM-as-a-judge paradigm, there is still room for improvement. Future work should focus on minimizing bias and improving the robustness of automatic evaluation methods across diverse model architectures, task types, and cultural contexts.

We view these limitations not only as constraints, but also as valuable directions for extending the scope, depth, and impact of IPBench in future work.

## C    DATA USAGE STATEMENT

In developing IPBench, all data are collected exclusively from open and publicly available sources. We strictly adhered to all relevant copyright and licensing regulations. Any data originating from websites or platforms that prohibit copying, redistribution, or automated crawling are explicitly excluded from use. Furthermore, we confirm that all data are used solely for academic and research purposes, and not for any commercial applications. We are committed to upholding responsible data usage and transparency in our research practices. Future updates of IPBench will continue to follow the same principles and remain fully open to academic scrutiny and community feedback.

## D    TAXONOMY AND TASK DETAILS

### D.1    TAXONOMY DETAILS

**Information Processing.**    In the first level of the taxonomy, we replace Recall and Reproduction with Information Processing, which encompasses the legal concepts, clauses, evolution, and typical case knowledge of various IP mechanisms. It also includes real-world applications such as patent classification, IP element identification, and process guidance, requiring models to memorize different concepts, along with the procedures executed in real-world scenarios. Our expert-annotated memory-type tasks are inspired by those in LexEval (Li et al., 2024) but differ significantly from it across various IP mechanisms, using accuracy as the evaluation metric. The IP element identification task focuses on identifying key elements in a case, such as claim coverage in patent infringement. Previous work has made significant progress in patent classification (Li et al., 2018; Lee & Hsiang, 2020; Fall et al., 2003), which has been adopted by IP offices in many countries. However, these models are task-specific and lack the strong generalization ability of LLMs. Our patent classification task consists of two types: International Patent Classification (IPC) and Cooperative Patent Classification (CPC). We aim to accomplish these tasks within a single model, enabling it to distinguish both differences within the same classification system and across different classification systems. We adopt the top-prediction scheme, following Fall et al. (Fall et al., 2003), to compare the top predicted category with the label for an Exact Match (Liang et al., 2022) in the main IPC symbol, and CPC. This setup increases the task difficulty for LLMs, requiring models to be familiar with classification rules.

**Logical Reasoning.**    At the second level of the taxonomy, we focus on examining a model's ability to apply memorized concepts and utilize logical reasoning to provide insights into both text analysis and mathematical calculations. One of the important roles of IP is to protect inventors' rights from infringement. Therefore, we define the tasks of Infringement Behavior Determination and Compensation Calculation. To complete these two tasks, models need to analyze the background of cases to identify infringement behavior and apply relevant laws to determine the appropriate penalties. Apart from the legal aspect, we introduce Patent Technology Forecasting, Patent Valuation, and Fact Checking to evaluate models' ability in information mining and conditional reasoning. As we mentioned, our IPBench consists of different IP mechanisms. We specifically introduce a novel task called Trade Secret Requirements, which differs from Infringement Behavior Determination. This task focuses on trade secret rights, requiring the model to determine whether a situation meets the confidentiality requirements of trade secrets.

**Discriminant Evaluation.**    At this level, we focus on evaluating models' understanding of IP in-domain texts, particularly patent documents, as well as their ability to perform discriminative tasks such as rights attribution. AAs an important part of IP management activities, as mentioned before, IP offices face a massive volume of patent applications. Determining the quality of an application requires assessing its patentability based on four aspects outlined in the Manual of Patent Examining Procedure (MPEP) (USTPO, 2024b; EPO, 1994): utility, non-obviousness, statutory subject matter, and novelty. We aim to evaluate whether current LLMs can assist patent examiners in reducing time costs within a single model. To achieve this, we introduce three tasks: Patent Document Proofreading, Patent Validity Identification, and Patent Match. LLMs' output mechanisms are not well-suited for retrieval-based approaches, and Li et al. (2025b) introduced a novel perspective on matching documents through a generative approach. Based on this insight, the Patent Match task draws in-

spiration from the corresponding task in MoZIP (Ni et al., 2024). We sample 1000 datapoints from MoZIP in both English and Chinese and require expert annotation for detailed examination.

Apart from the evaluation of in-domain text, we introduce one real-world common task for evaluating models' discrimination ability: Rights Attribution Analysis. The Rights Attribution Analysis task requires the model to infer the legal rights holder of a specific intellectual property based on the context of IP creation, legal agreements such as contract terms and confidentiality agreements, and judicial precedents within the legal framework. At last, we extend HUPD's (Suzgun et al., 2023) Patent Acceptance Prediction task into a more comprehensive Patent Application Examination task, leveraging the USPTO Office Action Dataset (Lu et al., 2017). In this task, the model is required to determine whether a given patent application should be accepted or rejected. Additionally, we provide stepwise examination actions for an interpretable examination process, which can be used in future work to construct a reliable examination system.

**Creative Generation.** At the final level of our IPBench, we focus on evaluating the models' ability to extract critical information, convert between different linguistic styles, and generate new content. Previous works such as BigPatent (Sharma et al., 2019), Patent-CR (Jiang et al., 2024), and PatentEval (Zuo et al., 2024) focus on specific types of content for patent generation. We draw inspiration from some of their tasks and extend their scope to include both Chinese and English. All the data used in Abstract Generation, Claim Generation, Sequential Claim Generation are sourced from the latest patents, ensuring no data leakage and distinguishing our dataset from existing ones. At last, we introduce a novel task called Design-Around Solution Generation, which evaluates whether models can generate innovative solutions that avoid duplication of existing ones. This capability is crucial in strategic patent planning. Given the distinct characteristics of the tasks at this level, we use accuracy as the metric for Language Simplification and Design-Around Solution Generation. For the other three generative tasks, we note that PatentEval (Zuo et al., 2024) provides an LLM-based evaluation method for claim generation. However, this approach relies on the assumption that the employed LLMs are sufficiently capable. Moreover, for other types of content, no superior evaluation method currently exists. We adopt a combination of automated evaluation and human assessment. For automated evaluation, we use n-gram-based metrics such as BLEU (Papineni et al., 2002) and ROUGE (Lin, 2004), along with the semantic metric BERTScore (Zhang et al., 2019), and analyze their consistency with human evaluation to enhance result interpretation. We will explore better evaluation methods in future work, especially for patent generation, which involves complex technical and legal content.

It is important to note that the abstract generation evaluation in BigPatent (Sharma et al., 2019) is based on converting only the first 400 words of a patent's description into an abstract, a limitation imposed by the context length of language models at the time. In our IPBench, we evaluate models on their ability to generate abstracts from the entire description, assessing their long-context understanding and summarization capabilities for complex patent documents.

## D.2 TASK DEFINITION

### D.2.1 INFORMATION PROCESSING

**Task 1-1: Legal Concept Memory** Legal Concept Memory refers to the ability to precisely memorize and recall foundational definitions within the intellectual property domain. These definitions, such as those of patents, copyrights, trademarks, and trade secrets, are grounded in authoritative legal frameworks and scholarly interpretations that constitute the foundation of intellectual property law. When given a concept name or contextual description, LLMs must retrieve the precise legal definition, scope, and jurisdictional boundaries as codified in statutes such as China's Patent Law and Copyright Law, as well as relevant international agreements, purely from their intrinsic knowledge without relying on external databases or tools.

**Task 1-2: Legal Clause Memory** Legal Clause Memory requires the precise memorization and retrieval of specific legal provisions, including their exact article numbers and textual content. These clauses, drawn from authoritative legal codes such as China's Criminal Law, Civil Code, and Intellectual Property Law, define rights, obligations, penalties, or procedural rules within statutory frameworks. When provided with an article number (e.g., Article 217 of China's Copyright Law) or

a contextual description of a legal scenario, LLMs must accurately recall the verbatim wording and scope of the corresponding clause.

**Task 1-3: Legal Evolution**  Legal Evolution refers to the ability to accurately memorize and analyze the revision history of legal texts, including the tracking of changes in specific clauses across different versions of statutes, regulations, or international treaties. This capability requires models to retain knowledge of amendments, such as updates to China's Patent Law, and to systematically compare the wording, scope, and intent of clauses before and after revisions.

**Task 1-4: Typical Case Memory**  Typical Case Memory requires the memorization of landmark intellectual property cases, including their judicial outcomes, factual details, and legal reasoning. These cases, such as high-profile patent disputes, copyright infringement rulings, or trademark opposition decisions, establish precedents that shape the interpretation and enforcement of IP law. When provided with a case name, jurisdiction, or factual scenario, models must accurately recall the judgment summary, key legal arguments, cited statutes, and contextual factors, without using an external database or retrieval tool.

**Task 1-5: Patent Classification**  Patent Classification involves the capability to automatically assign International Patent Classification (IPC) or Cooperative Patent Classification (CPC) codes based on the technical content of patent documents. This task requires models to analyze patent texts, including titles and abstractsto identify the core inventions, technological domains, and functional features, then map them to hierarchical classification codes.This task evaluates the model's capabilities across three hierarchical levels: Section, Class, and Subclass. A distribution table for the section level as shown in Table 7.

Table 7: International Patent Classification (IPC) Sections

| Section | Content |
|---------|---------|
| A | Human Necessities |
| B | Performing Operations; Transporting |
| C | Chemistry; Metallurgy |
| D | Textiles; Paper |
| E | Fixed Constructions |
| F | Mechanical Engineering; Lighting; Heating; Weapons; Blasting |
| G | Physics |
| H | Electricity |

Table 8: Cooperative Patent Classification (CPC) Sections

| Section | Content |
|---------|---------|
| A | Human Necessities |
| B | Operations and Transport |
| C | Chemistry and Metallurgy |
| D | Textiles and Paper |
| E | Fixed Constructions |
| F | Mechanical Engineering and Lighting |
| G | Physics |
| H | Electricity |
| Y | Emerging Technologies |

**Task 1-6: IP Element Identification**  IP Element Identification entails detecting and categorizing intellectual property componentssuch as patent claims, trademark-protected assets, copyrighted material, or trade secret identifierswithin legal disputes, technical specifications, or commercial contracts. This task requires models to analyze textual data to identify legally protected innovations, distinctive brand assets, and ownership claims, while ensuring alignment with statutory definitions.

**Task 1-7: Process Guidance**  Process Guidance focuses on delivering structured knowledge of intellectual property application procedures, covering legal requirements, technical documentation

standards, and jurisdictional workflows. This task requires models to provide step-by-step guidance on processes such as conducting patent or trademark searches, drafting application materials, navigating submission procedures, and ensuring compliance with examination regulations.

### D.2.2 LOGICAL REASONING

The Logical Reasoning level is designed to evaluate the capability of large language models (LLMs) to perform multi-dimensional legal and technical reasoning within the complex framework of intellectual property (IP) law and textual analysis. This layer tests the model's ability to analyze, interpret, and apply intersecting legal rules. It focuses on assessing whether models can synthesize statutory provisions, case law precedents, and technical domain knowledge to reach legally sound conclusions such as identifying infringement risks, resolving conflicts between overlapping rights, or predicting litigation outcomes based on factual scenarios.

**Task 2-1: Patent Technology Forecasting** Patent Technology Forecasting involves analyzing the technical features of patents such as claims, innovation summaries, and domain-specific terminology to predict future technological trajectories and potential application areas. This task requires models to identify emerging trends, interconnected technical fields, and latent innovation pathways within patent datasets, enabling the projection of how core inventions might evolve or intersect with adjacent industries.

**Task 2-2: Infringement Behavior Determination** Infringement Behavior Determination focuses on identifying acts that constitute violations of intellectual property rights. It involves analyzing the legally protected scope of patents, copyrights, trademarks, or other IP types, and comparing them with suspected infringing products, services, or content to determine whether an intellectual property infringement has occurred. This task requires models to evaluate technical equivalence, trademark similarity, or substantial similarity in copyrighted works, while accurately applying the relevant statutory criteria to determine whether an intellectual property infringement has occurred.

**Task 2-3: Compensation Calculation** Compensation Calculation focuses on determining statutory damages for intellectual property infringement by analyzing the severity, scope, and economic impact of the violation. This task requires models to perform mathematical reasoning and calculation, taking into account factors such as the rights holder's actual losses, reasonable licensing fees, and statutory limits. Additionally, models must incorporate contextual elements such as the duration of infringement, geographic scope, and the presence of malicious intent to arrive at a legally grounded and quantitatively sound compensation estimate.

**Task 2-4: Patent Valuation** Patent Valuation entails evaluating the value trajectory of a patent by synthesizing its technical merit, market viability, and legal robustness. This task requires models to analyze technical claims, market analysis reports, and legal histories to project trends such as value appreciation, obsolescence risks, or licensing potential.

**Task 2-5: Trade Secret Requirements** Trade Secret Requirements assesses whether a given scenario satisfies the legal criteria for trade secret protection under statutory frameworks such as China's Anti-Unfair Competition Law and the U.S. Defend Trade Secrets Act (DTSA). This task requires models to verify three core elements: the existence of secrecy, the presence of commercial value, and the implementation of reasonable confidentiality measures.

### D.2.3 DISCRIMINANT EVALUATION

**Task 3-1: Patent Document Proofreading** Patent Document Proofreading involves identifying formatting deviations and logical inconsistencies within patent specifications, claims, and technical descriptions to ensure compliance with statutory drafting standards. This task requires models to detect issues such as mismatched section numbering, non-compliant claim dependencies, contradictory technical descriptions, and deviations from jurisdiction-specific filing guidelines.

**Task 3-2: Patent Validity Identification** Patent Validity Identification involves assessing whether a patent satisfies the statutory criteria of novelty, inventiveness (non-obviousness), and practical applicability (utility) by analyzing its technical disclosures in light of relevant prior art. This task

requires models to evaluate patent texts, including claims and specifications, against existing technologies to determine if the invention is new, involves an inventive step, and has industrial applicability.

**Task 3-3: Patent Match**  Patent Match involves identifying the most relevant patents from a candidate pool based on technical, legal, and contextual alignment with a query patent. This task requires models to analyze technical features and semantic similarity to rank patents by relevance. This task is inspired by MoZIP (Ni et al., 2024).

**Task 3-4: Rights Attribution Analysis**  Rights Attribution Analysis involves determining the legitimate rights holder in intellectual property ownership disputes by analyzing legal documents, contractual agreements, and contextual evidence. This task requires models to evaluate factors such as invention ownership under employment relationships, joint authorship claims in copyright cases, or trademark transfer agreements, while reconciling conflicting claims based on statutory frameworks.

**Task 3-5: Patent Application Examination**  Patent Application Examination involves conducting compliance reviews of patent documents to ensure adherence to statutory and administrative requirements. This task requires models to verify the accuracy, completeness, and legal sufficiency of patent applications, including claims, specifications, and drawings, against jurisdictional standards. Key checks include clarity of technical disclosure, consistency between claims and descriptions, proper support for embodiments, and alignment with formalities. The data for this task is sourced from the USPTO Office Action Dataset (Lu et al., 2017).

### D.2.4 Creative Generation

**Task 4-1: Abstract Generation**  Abstract Generation assesses a model's ability to automatically extract core elements from intellectual property (IP) texts, such as patent claims, and synthesize them into concise, structured, and legally compliant summaries. This task requires models to distill technical innovations, legal scopes, and critical details while adhering to jurisdictional formatting rules and avoiding oversimplification that misrepresents legal or technical nuances.

**Task 4-2: Dependent Claim Generation**  Dependent Claim Generation involves automatically drafting legally compliant and technically precise dependent claims based on the core inventions described in patent disclosures. This task requires models to analyze technical descriptions and generate claims that refine or limit the scope of independent claims by incorporating additional technical features, while ensuring logical dependency and alignment with jurisdictional formalities. This task is inspired by PatentEval (Zuo et al., 2024).

**Task 4-3: Design-Around Solution Generation**  Design-Around Solution Generation focuses on creating non-infringing technical alternatives by analyzing existing patent claims and identifying opportunities to circumvent key protected elements. This task requires models to deconstruct patent claims and propose modifications that avoid literal or equivalent infringement, while maintaining technical feasibility.

Table 9: Data language distribution of IPBench.

| Language | 1-1 | 1-2 | 1-3 | 1-4 | 1-5-1 | 1-5-2 | 1-6 | 1-7 | 2-1 | 2-2 | 2-3 | 2-4 | 2-5 | 3-1 | 3-2 | 3-3 | 3-4 | 3-5 | 4-1 | 4-2 | 4-3 | Sum |
|---|---|---|---|---|---|---|---|---|---|---|---|---|---|---|---|---|---|---|---|---|---|---|
| Chinese | 259 | 276 | 294 | 252 | 525 | 0 | 338 | 308 | 250 | 228 | 156 | 139 | 301 | 160 | 159 | 500 | 217 | 0 | 200 | 200 | 328 | **5090** |
| English | 241 | 226 | 206 | 252 | 600 | 600 | 219 | 240 | 250 | 272 | 160 | 162 | 0 | 140 | 149 | 500 | 183 | 314 | 200 | 199 | 171 | **5284** |
| Total | 500 | 502 | 500 | 504 | 1125 | 600 | 557 | 548 | 500 | 500 | 316 | 301 | 301 | 300 | 308 | 1000 | 400 | 314 | 400 | 399 | 499 | **10374** |

## E  Data Annotation and Examination Protocol

### E.1  Data Collection

We list the primary websites from which we collected the raw data as follows:

- USTPO's Open Data Portal: https://data.uspto.gov/home

- CNIPA's Official Website: https://www.cnipa.gov.cn/

- China Judgements Online: https://wenshu.court.gov.cn/

**Ethical considerations.** The data we collected come from open and public sources, and we confirm that they are not used for any commercial purposes. We strictly comply with all copyright and licensing regulations. Data originating from sources that do not allow copying or redistribution are deliberately excluded.

### E.2 ANNOTATION AND EXAMINATION GUIDELINES

We provide detailed data annotation guidelines to ensure the quality, correctness, and difficulty of our benchmark. Notably, most of our human expert annotators, who come from backgrounds in intellectual property and public management, range from senior undergraduates to Ph.D. candidates. They are included as co-authors of this paper as a non-monetary form of acknowledgment for their efforts. They possess deep knowledge of intellectual property.

**Preparation before annotation.** We divide our 21 human expert annotators into four groups and assign them to different tasks, including data annotation and annotation review. Each group is required to thoroughly understand their assigned task and formulate a comprehensive annotation plan accordingly. This involves understanding the task definition, relevant legal concepts, and technical terminologies related to intellectual property.

**General principles and process of annotation.** Firstly, all raw data or information must be collected from official websites that are publicly accessible. For websites that prohibit copying, annotators are instructed not to use them. Secondly, all annotators are required to ensure the accuracy of their annotated questions and to ensure that the difficulty level is appropriate. For data containing mathematical equations or special notations, we ask annotators to convert them into LaTeX format. For other typographical errors, human expert annotators will correct them manually. Thirdly, all data will be examined by switching roles between annotation teams to verify and ensure their quality. For each datapoint, after the quality check, human expert annotators are required to label the language, the type of IP mechanism, and the data source.

**Specific principles of examination.** To ensure data quality, we assign a different annotation team to double-check and cross-validate the results. In cases where errors, inconsistencies, or misunderstandings are identified, human examiners must provide detailed explanations and determine whether the data can be corrected and preserved. After the annotator corrects the question, the examiner will re-evaluate the data until it passes the review with mutual agreement. This strict process ensures the reliability of our data, with each datapoint undergoing an average of three rounds of review to form IPBench.

## F   MORE DETAILS ABOUT DATA STATICS

In this section, we provide additional details about the data. Further statistical information can be found in Section F.

Our IPBench comprises 10,374 datapoints spanning 20 tasks, including multiple-choice questions, classification tasks, and generation tasks. In this section, we provide additional data statistics, covering language distribution, IP mechanism distribution, IPC/CPC classification distribution, text length distribution, and the distribution of option counts in multiple-choice questions.

### F.1   DATA LANGUAGE DISTRIBUTION

Our IPBench is constrained to the legal frameworks of the United States and mainland China; therefore, the dataset includes both English and Chinese languages. We present the language distribution for each task, as well as for the entire dataset, in Table 9.

Table 10: Intellectual property mechanisms distribution of IPBench. TD: Trademark, SC: Software Copyright, TS: Trade Secret, PV: Plant Variety, CR: Copyright, IC: Integrated Circuit, GM: Geographical Mark.

| Task | Patent | TD | SC | TS | PV | CR | IC | GM | Total |
|------|--------|-----|-----|-----|-----|-----|----|----|-------|
| 1-1 | 225 | 157 | 13 | 25 | 13 | 34 | 24 | 9 | 500 |
| 1-2 | 221 | 95 | 21 | 0 | 0 | 141 | 6 | 18 | 502 |
| 1-3 | 237 | 116 | 1 | 1 | 1 | 143 | 0 | 1 | 500 |
| 1-4 | 325 | 37 | 12 | 33 | 29 | 58 | 8 | 2 | 504 |
| 1-5-1 | 525 | 600 | 0 | 0 | 0 | 0 | 0 | 0 | 1125 |
| 1-5-2 | 600 | 0 | 0 | 0 | 0 | 0 | 0 | 0 | 600 |
| 1-6 | 159 | 103 | 22 | 107 | 1 | 157 | 1 | 7 | 557 |
| 1-7 | 190 | 358 | 0 | 0 | 0 | 0 | 0 | 0 | 548 |
| 2-1 | 320 | 21 | 9 | 77 | 39 | 24 | 10 | 0 | 500 |
| 2-2 | 183 | 105 | 16 | 49 | 3 | 144 | 0 | 0 | 500 |
| 2-3 | 101 | 94 | 11 | 10 | 0 | 100 | 0 | 0 | 316 |
| 2-4 | 301 | 0 | 0 | 0 | 0 | 0 | 0 | 0 | 301 |
| 2-5 | 0 | 0 | 0 | 301 | 0 | 0 | 0 | 0 | 301 |
| 3-1 | 300 | 0 | 0 | 0 | 0 | 0 | 0 | 0 | 300 |
| 3-2 | 308 | 0 | 0 | 0 | 0 | 0 | 0 | 0 | 308 |
| 3-3 | 1000 | 0 | 0 | 0 | 0 | 0 | 0 | 0 | 1000 |
| 3-4 | 353 | 0 | 8 | 13 | 18 | 5 | 3 | 0 | 400 |
| 3-5 | 314 | 0 | 0 | 0 | 0 | 0 | 0 | 0 | 314 |
| 4-1 | 400 | 0 | 0 | 0 | 0 | 0 | 0 | 0 | 400 |
| 4-2 | 399 | 0 | 0 | 0 | 0 | 0 | 0 | 0 | 399 |
| 4-3 | 497 | 1 | 0 | 1 | 0 | 0 | 0 | 0 | 499 |
| **Total** | **6958** | **1687** | **113** | **617** | **104** | **806** | **52** | **37** | **10374** |

Table 11: Distribution of intellectual property mechanisms in the English portion of IPBench.

| Task | Patent | TD | SC | TS | PV | CR | IC | GM | Total |
|------|--------|-----|-----|-----|-----|-----|----|----|-------|
| 1-1 | 150 | 57 | 0 | 1 | 10 | 3 | 20 | 0 | 241 |
| 1-2 | 92 | 52 | 0 | 0 | 0 | 82 | 0 | 0 | 226 |
| 1-3 | 53 | 64 | 0 | 1 | 1 | 86 | 0 | 1 | 206 |
| 1-4 | 202 | 13 | 4 | 4 | 0 | 29 | 0 | 0 | 252 |
| 1-5-1 | 0 | 600 | 0 | 0 | 0 | 0 | 0 | 0 | 600 |
| 1-5-2 | 600 | 0 | 0 | 0 | 0 | 0 | 0 | 0 | 600 |
| 1-6 | 65 | 54 | 8 | 26 | 0 | 66 | 0 | 0 | 219 |
| 1-7 | 58 | 182 | 0 | 0 | 0 | 0 | 0 | 0 | 240 |
| 2-1 | 170 | 21 | 4 | 22 | 9 | 24 | 0 | 0 | 250 |
| 2-2 | 101 | 58 | 4 | 29 | 0 | 80 | 0 | 0 | 272 |
| 2-3 | 52 | 45 | 6 | 2 | 0 | 55 | 0 | 0 | 160 |
| 2-4 | 162 | 0 | 0 | 0 | 0 | 0 | 0 | 0 | 162 |
| 2-5 | 0 | 0 | 0 | 0 | 0 | 0 | 0 | 0 | 0 |
| 3-1 | 140 | 0 | 0 | 0 | 0 | 0 | 0 | 0 | 140 |
| 3-2 | 149 | 0 | 0 | 0 | 0 | 0 | 0 | 0 | 149 |
| 3-3 | 500 | 0 | 0 | 0 | 0 | 0 | 0 | 0 | 500 |
| 3-4 | 175 | 0 | 0 | 4 | 0 | 4 | 0 | 0 | 183 |
| 3-5 | 314 | 0 | 0 | 0 | 0 | 0 | 0 | 0 | 314 |
| 4-1 | 200 | 0 | 0 | 0 | 0 | 0 | 0 | 0 | 200 |
| 4-2 | 199 | 0 | 0 | 0 | 0 | 0 | 0 | 0 | 199 |
| 4-3 | 169 | 1 | 0 | 1 | 0 | 0 | 0 | 0 | 171 |
| **Total** | **3551** | **1147** | **26** | **90** | **20** | **429** | **20** | **1** | **5284** |

## F.2 INTELLECTUAL PROPERTY MECHANISMS DISTRIBUTION

Our IPBench covers eight intellectual property mechanisms, including Patent, Trademark, Software Copyright, Trade Secret, New Plant Variety, Copyright, Integrated Circuit Layout Design, and Geographical Indication. We present a detailed distribution of these intellectual property mechanisms in our benchmark, as shown in Table 10, Table 11 (English section), and Table 12 (Chinese section).

Table 12: Distribution of Intellectual Property Mechanisms in the Chinese Portion of IPBench.

| Task | Patent | TD | SC | TS | PV | CR | IC | GM | Total |
|------|--------|-----|-----|-----|-----|-----|----|----|-------|
| 1-1 | 75 | 100 | 13 | 24 | 3 | 31 | 4 | 9 | 259 |
| 1-2 | 129 | 43 | 21 | 0 | 0 | 59 | 6 | 18 | 276 |
| 1-3 | 184 | 52 | 1 | 0 | 0 | 57 | 0 | 0 | 294 |
| 1-4 | 123 | 24 | 8 | 29 | 29 | 29 | 8 | 2 | 252 |
| 1-5-1 | 525 | 0 | 0 | 0 | 0 | 0 | 0 | 0 | 525 |
| 1-5-2 | 0 | 0 | 0 | 0 | 0 | 0 | 0 | 0 | 0 |
| 1-6 | 94 | 49 | 14 | 81 | 1 | 91 | 1 | 7 | 338 |
| 1-7 | 132 | 176 | 0 | 0 | 0 | 0 | 0 | 0 | 308 |
| 2-1 | 150 | 0 | 5 | 55 | 30 | 0 | 10 | 0 | 250 |
| 2-2 | 82 | 47 | 12 | 20 | 3 | 64 | 0 | 0 | 228 |
| 2-3 | 49 | 49 | 5 | 8 | 0 | 45 | 0 | 0 | 156 |
| 2-4 | 139 | 0 | 0 | 0 | 0 | 0 | 0 | 0 | 139 |
| 2-5 | 0 | 0 | 0 | 301 | 0 | 0 | 0 | 0 | 301 |
| 3-1 | 160 | 0 | 0 | 0 | 0 | 0 | 0 | 0 | 160 |
| 3-2 | 159 | 0 | 0 | 0 | 0 | 0 | 0 | 0 | 159 |
| 3-3 | 500 | 0 | 0 | 0 | 0 | 0 | 0 | 0 | 500 |
| 3-4 | 178 | 0 | 8 | 9 | 18 | 1 | 3 | 0 | 217 |
| 3-5 | 0 | 0 | 0 | 0 | 0 | 0 | 0 | 0 | 0 |
| 4-1 | 200 | 0 | 0 | 0 | 0 | 0 | 0 | 0 | 200 |
| 4-2 | 200 | 0 | 0 | 0 | 0 | 0 | 0 | 0 | 200 |
| 4-3 | 328 | 0 | 0 | 0 | 0 | 0 | 0 | 0 | 328 |
| **Total** | **3407** | **540** | **87** | **527** | **84** | **377** | **32** | **36** | **5090** |

## F.3 IPC AND CPC CLASSIFICATION DISTRIBUTION

We present the IPC Section classification distribution in Table 13a and the CPC Section classification distribution in Table 13b.

Table 13: Distribution of IPC and CPC sections.

| Section | Count | Percentage (%) |
|---|---|---|
| A | 72 | 6.5 |
| B | 249 | 22.1 |
| C | 29 | 2.6 |
| D | 6 | 0.5 |
| E | 63 | 5.6 |
| F | 64 | 5.7 |
| G | 113 | 10.0 |
| H | 529 | 47.0 |
| **All** | **1125** | **100** |

(a) Distribution of IPC sections

| Section | Count | Percentage (%) |
|---|---|---|
| A | 90 | 15.0 |
| B | 90 | 15.0 |
| C | 48 | 8.0 |
| D | 17 | 2.8 |
| E | 71 | 11.9 |
| F | 189 | 31.5 |
| G | 44 | 7.3 |
| H | 51 | 8.5 |
| **All** | **600** | **100** |

(b) Distribution of CPC sections

Table 14: Text length statistics (in tokens) for each task across three dimensions: average, minimum, and maximum length; each further split by language (EN/CH). Missing values are denoted by "-".

| Task | Avg-All | Avg-EN | Avg-CH | Min-All | Min-EN | Min-CH | Max-All | Max-EN | Max-CH |
|---|---|---|---|---|---|---|---|---|---|
| 1-1 | 83.9 | 68.9 | 97.8 | 46 | 46 | 66 | 258 | 112 | 258 |
| 1-2 | 81.2 | 71.3 | 89.3 | 47 | 47 | 66 | 135 | 105 | 135 |
| 1-3 | 102.2 | 80.3 | 117.6 | 55 | 55 | 76 | 208 | 129 | 208 |
| 1-4 | 116.2 | 112.8 | 119.6 | 61 | 61 | 73 | 195 | 151 | 195 |
| 1-5-1 | 216.7 | 163.6 | 277.4 | 49 | 49 | 110 | 305 | 305 | 455 |
| 1-5-2 | 165.1 | 165.1 | - | 50 | 50 | - | 337 | 337 | - |
| 1-6 | 89.4 | 77.0 | 97.5 | 55 | 55 | 70 | 146 | 146 | 128 |
| 1-7 | 41.7 | - | 74.2 | 40 | 40 | 53 | 107 | 101 | 107 |
| 2-1 | 161.6 | 103.1 | 220.1 | 66 | 66 | 177 | 310 | 140 | 310 |
| 2-2 | 109.4 | 97.4 | 123.6 | 59 | 59 | 71 | 211 | 189 | 211 |
| 2-3 | 122.8 | 107.9 | 138.1 | 70 | 70 | 80 | 263 | 171 | 263 |
| 2-4 | 99.8 | 88.1 | 113.5 | 66 | 66 | 87 | 144 | 125 | 144 |
| 2-5 | 112.1 | - | 112.1 | 51 | - | 51 | 302 | - | 302 |
| 3-1 | 158.2 | 145.8 | 169.1 | - | - | - | - | - | - |
| 3-2 | 91.4 | 76.0 | 105.9 | 53 | 53 | 78 | 150 | 121 | 150 |
| 3-3 | 1239.5 | 1231.8 | 1247.3 | 575 | 581 | 575 | 1956 | 1845 | 1956 |
| 3-4 | 166.1 | 169.8 | 163.0 | 60 | 92 | 60 | 297 | 297 | 327 |
| 3-5 | 7460.4 | 7460.4 | 60.5 | 1428 | 1428 | - | 10219 | 10219 | - |
| 4-1 | 1636.7 | 2199.3 | 1074.0 | - | - | 285 | - | 8064 | 5675 |
| 4-2 | 448.5 | 534.1 | 363.3 | 68 | 89 | 68 | 1861 | 1485 | 1861 |
| 4-3 | 121.2 | 111.8 | 126.1 | 56 | 56 | 84 | 218 | 183 | 218 |

Table 15: Aggregated text length statistics (in tokens) by task type. PE refers to Patent Examination (Task 3-5), MCQA refers to Multiple-choice Question Answering.

| Type | Avg-All | Avg-EN | Avg-CH | Min-All | Min-EN | Min-CH | Max-All | Max-EN | Max-CH |
|---|---|---|---|---|---|---|---|---|---|
| MCQA | 181.0 | 181.6 | 194.7 | 90.7 | 96.2 | 111.1 | 326.7 | 272.5 | 327.5 |
| PE | 7460.4 | 7460.4 | 60.5 | 1428.0 | 1428.0 | - | 10219.0 | 10219.0 | - |
| Classification | 190.9 | 164.4 | 277.4 | 49.5 | 49.5 | 110.0 | 321.0 | 321.0 | 455.0 |
| Generation | 1042.6 | 1366.7 | 718.7 | 68.0 | 89.0 | 176.5 | 1861.0 | 4774.5 | 3768.0 |

## F.4 TEXT LENGTH DISTRIBUTION

We provide detailed statistics on the text length distribution for each task, across the three question types, in both Chinese and English. In all text length computations presented in this paper, we

Table 16: Distribution of answer choices by task.

| Task | A | B | C | D | Total |
|------|------|------|------|------|------|
| 1-1 | 125 | 129 | 126 | 120 | 500 |
| 1-2 | 117 | 162 | 117 | 106 | 502 |
| 1-3 | 125 | 126 | 126 | 123 | 500 |
| 1-4 | 126 | 126 | 127 | 125 | 504 |
| 1-6 | 151 | 137 | 142 | 127 | 557 |
| 1-7 | 132 | 194 | 124 | 98 | 548 |
| 2-1 | 155 | 130 | 124 | 91 | 500 |
| 2-2 | 101 | 177 | 154 | 68 | 500 |
| 2-3 | 74 | 108 | 85 | 49 | 316 |
| 2-4 | 74 | 83 | 75 | 69 | 301 |
| 2-5 | 57 | 165 | 63 | 16 | 301 |
| 3-1 | 114 | 82 | 59 | 45 | 300 |
| 3-2 | 72 | 76 | 76 | 84 | 308 |
| 3-3 | 240 | 256 | 230 | 278 | 1000 |
| 3-4 | 76 | 141 | 128 | 55 | 400 |
| 4-3 | 170 | 144 | 111 | 74 | 499 |
| **Total** | **1909** | **2236** | **1867** | **1528** | **7536** |

Table 17: Distribution of English questions' answers by task.

| Task | A | B | C | D | Total |
|------|------|------|------|------|------|
| 1-1 | 64 | 53 | 65 | 59 | 241 |
| 1-2 | 65 | 69 | 48 | 44 | 226 |
| 1-3 | 44 | 47 | 58 | 57 | 206 |
| 1-4 | 70 | 71 | 57 | 54 | 252 |
| 1-6 | 39 | 70 | 40 | 70 | 219 |
| 1-7 | 41 | 70 | 64 | 65 | 240 |
| 2-1 | 92 | 65 | 64 | 29 | 250 |
| 2-2 | 70 | 68 | 77 | 57 | 272 |
| 2-3 | 38 | 68 | 26 | 28 | 160 |
| 2-4 | 39 | 45 | 40 | 38 | 162 |
| 3-1 | 54 | 38 | 33 | 15 | 140 |
| 3-2 | 38 | 35 | 39 | 37 | 149 |
| 3-3 | 120 | 126 | 115 | 139 | 500 |
| 3-4 | 44 | 64 | 50 | 25 | 183 |
| 4-3 | 44 | 65 | 42 | 20 | 171 |
| **Total** | **862** | **954** | **818** | **737** | **3371** |

adopt the tokenizer of GPT-4o for consistency and comparability. Table 14 and Table 15 present the distribution of text lengths from different perspectives: the former provides statistics by task, while the latter summarizes the data by question type.

## F.5 MULTI-CHOICE QUESTION OPTION COUNT DISTRIBUTION.

In this section, we present the distribution of multiple-choice question option counts, as shown in Table 16, Table 17, and Table 18, along with the examination option distribution for Task 3-5, as shown in Table 19. For multiple-choice questions, each question has four options: A, B, C, and D. In contrast, for Task 3-5, each question has two options: allowed and rejected.

Table 18: Distribution of Chinese questions' answers by task.

| Task | A | B | C | D | Total |
|------|------|------|------|------|------|
| 1-1 | 61 | 76 | 61 | 61 | 259 |
| 1-2 | 52 | 93 | 69 | 62 | 276 |
| 1-3 | 81 | 79 | 68 | 66 | 294 |
| 1-4 | 56 | 55 | 70 | 71 | 252 |
| 1-6 | 112 | 67 | 102 | 57 | 338 |
| 1-7 | 91 | 124 | 60 | 33 | 308 |
| 2-1 | 63 | 65 | 60 | 62 | 250 |
| 2-2 | 31 | 109 | 77 | 11 | 228 |
| 2-3 | 36 | 40 | 59 | 21 | 156 |
| 2-4 | 35 | 38 | 35 | 31 | 139 |
| 2-5 | 57 | 165 | 63 | 16 | 301 |
| 3-1 | 60 | 44 | 26 | 30 | 160 |
| 3-2 | 34 | 41 | 37 | 47 | 159 |
| 3-3 | 120 | 126 | 115 | 139 | 500 |
| 3-4 | 32 | 77 | 78 | 30 | 217 |
| 4-3 | 126 | 79 | 69 | 54 | 328 |
| **Total** | **1047** | **1278** | **1049** | **791** | **4165** |

Table 19: Examination outcome distribution for Task 3-5.

| Examination Outcome | Count | Percentage (%) |
|---|---|---|
| Allowed | 138 | 43.95 |
| Rejected | 176 | 56.05 |
| **Total** | **314** | **100** |

# G PROMPTS

## G.1 ZERO-SHOT AND FEW-SHOT PROMPT

We adapt four types of zero-shot prompts and few-shot prompts for our experiment, corresponding to different task types: choice questions, classification, examination, and generation, across both English and Chinese languages. The Chinese version uses the same content as the English version.

---

**Zero-shot Prompt for Choice Question Task**

Please answer the following question thoughtfully and provide your final answer at the end in the format 'Answer: **option**'

{ Question }

---

**Zero-shot Prompt for IPC/CPC Classification Task (1-5)**

Please answer the following question thoughtfully and provide your final answer at the end in the format 'Answer: **corresponding IPC number**'

{Question}

---

**Zero-shot Prompt for Generation Task (4-1, 4-2)**

**Abstract Generation based on Claims (4-1):**
# Claims
{Claims Text}
Please generate the abstract of the patent based on the given claims.

**Dependent Claim Generation (1-5-2):**
# Independent Claim
{Claim Text}
Please generate all dependent claims corresponding to the given independent claim.

---

**Zero-shot Prompt for Patent Application Examination Task (3-5)**

Please examine the patents in # Patent Applications Awaiting Examination. Determine whether each patent application should be allowed or rejected.
Return your decision in the following format:

Answer: allowed / rejected

---

---

**Few-shot Prompt for Choice Question Task**

\# There are k examples
\#\# Example {1}
Question: {1-shot-question}
Answer:{1-shot-answer}

...
\#\# Example {k}
Question: {k-shot-question}
Answer:{k-shot-answer}

Please answer the following question thoughtfully and provide your final answer at the end in the format 'Answer: **option**'

{ Question }

---

**Few-shot Prompt for IPC/CPC Classification Task**

\# There are k examples
\#\# Example {1}
Question: {1-shot-question}
Answer:{1-shot-answer}

...
\#\# Example {k}
Question: {k-shot-question}
Answer:{k-shot-answer}

Please answer the following question thoughtfully and provide your final answer at the end in the format 'Answer: **corresponding IPC/CPC number**'

{Question}

---

## G.2 CHAIN-OF-THOUGHT PROMPT

---

**Chain-of-Thought Prompt for Choice Question Task**

Please answer the following question thoughtfully and provide your final answer at the end in the format 'Answer: **option**'

{ Question }

**Let's think step by step.**

---

**Chain-of-Thought Prompt for IPC/CPC Classification Task**

Please answer the following question thoughtfully and provide your final answer at the end in the format 'Answer: **corresponding IPC/CPC number**'

{Question}

**Let's think step by step.**

---

# H METRICS

In this section, we provide the details of the metrics used in our IPBench. The details of the multiple-choice question metric are in Section H.1, the details of the classification task metric are in Section H.2, and the details of the generation task are in Section H.3.

Table 20: The overview of evaluated models. Max Context refers to the maximum context length of the model without length extrapolation for all models.

| Model | Size | Max Context | Type | Orientation | Access |
|---|---|---|---|---|---|
| GPT-4o (Hurst et al., 2024) | – | 128k | Chat Model | General | OpenAI API |
| GPT-4o-mini (Hurst et al., 2024) | – | 128k | Chat Model | General | OpenAI API |
| DeepSeek-V3 (DeepSeek-AI et al., 2024) | 671B | 128k | Chat Model | General | DeepSeek API |
| Qwen3 (Team, 2025a) | 8B | 32k | Chat Model | General | Weights |
| Qwen2.5-Instruct (Yang et al., 2024) | 7/72B | 32k | Chat Model | General | Weights |
| Llama3.1-Instruct (Dubey et al., 2024) | 8/70B | 32k | Chat Model | General | Weights |
| Gemma-2-Instruct (Riviere et al., 2024) | 9/27B | 8k | Chat Model | General | Weights |
| Mistral-7B-Instruct (Jiang, 2024) | 7B | 32k | Chat Model | General | Weights |
| MoZi-qwen (Ni et al., 2024) | 7B | 32k | Chat Model | IP | Weights |
| DISC-LawLLM (Yue et al., 2023; 2024) | 6B | 2048 | Chat Model | Law | Weights |
| HanFei (He et al., 2023) | 7B | 2048 | Chat Model | Law | Weights |
| DeepSeek-R1 (DeepSeek-AI et al., 2025) | 671B | 128k | Reasoning Model | General | DeepSeek API |
| Deepseek-R1-Distill-Qwen (DeepSeek-AI et al., 2025) | 7B | 32k | Reasoning Model | General | Weights |
| QwQ (Team, 2025b) | 32B | 32k | Reasoning Model | General | Weights |

## H.1 MULTI-CHOICE QUESTION METRIC

For multiple-choice questions, we use accurac'y as the metric due to the straightforward nature of the judgment process. Each multiple-choice question has four options: A, B, C, and D. We use the same extraction method for each model's response, compare the selected answer with the ground-truth option, and then compute the average accuracy. The average score ranges from 0 to 100, and is computed as shown in the Equation 1.

$$\text{Accuracy} = \frac{\text{Number of Correct Answers \#}}{\text{Total Number of Questions \#}} \tag{1}$$

## H.2 IPC/CPC CLASSIFICATION TASK METRIC

For IPC/CPC classification task, we use exact-match as the metric. For example, in the IPC code A01B00/66, 'A' represents the Section, '01' the Class, and 'B' the Subclass. If the model predicts 'A', it earns one point for the Section; if it predicts 'A01', it earns one point for the Class; and if it predicts 'A01B', it earns one point for the Subclass. If the entire code is predicted correctly, one point is awarded for the Exact Match. We evaluate all the test data to calculate the average exact-match score across these four levels. The difficulty increases as the model is required to make correct predictions at more levels.

## H.3 GENERATION TASK METRIC

In this section, we provide the details of the LLM-as-a-judge approach used for LLMScore and analyze its consistency with human evaluation.

We design five evaluation dimensions for LLM-as-a-judge: Accuracy, Relevance, Completeness, Consistency, and Language-Style. The detailed definitions are provided in the prompts below. Each dimension is scored on a scale from 1 to 10 points. We use DeepSeek-V3 as the judge model because it achieves relatively better performance on the multiple-choice tasks, indicating solid knowledge in the intellectual property domain. In addition to the LLM-as-a-judge evaluation, we further sample 50 responses each from GPT-4o, DeepSeek-V3 and LLaMA3.1-8B-Instruct for the two tasks. These responses are assessed by three human experts using the same criteria as the LLM-as-a-judge framework. The results and the corresponding consistency between the LLM and human analysis are presented in Table 6.

We provide a consistency analysis between different metrics and human evaluations, including Kendall, Pearson, and Spearman coefficients. The higher the consistency coefficient, the better, indicating stronger consistency; the smaller the p-value, the better, indicating statistical significance. A smaller $p$-value, typically less than 0.05, indicates that the observed correlation is statistically significant.

**LLMScore for Generation Task.** For Task 4-1 and 4-2, we draw inspiration from the error taxonomy for abstract generation and dependent claims generation proposed in PatentEval (Zuo et al., 2024), and used five dimensions to evaluate the quality of the generated abstract. The specific prompt we use for LLM-as-a-judge in evaluating generation task are provided in code.

## I DETAILS ABOUT EVALUATED MODELS

We provide details of the evaluated models, including their size, context length, type, and access method, as shown in Table 20.

## J MORE DISCUSSION

The growing integration of LLMs into high-stakes domains demands rigorous, domain-specific evaluation frameworks. Among these domains, IP presents unique challenges that remain largely unaddressed in existing NLP benchmarks. IP tasks operate at the intersection of technical innovation and legal regulation, requiring precise reasoning over structured taxonomies (e.g., IPC/CPC classifications), formal legal constructs (e.g., claim scope and infringement logic), and high-stakes decisions (e.g., patentability, damages, licensing). Yet most LLM benchmarks either omit this domain or reduce it to surface-level tasks like summarization or basic classification.

This oversight poses real risks. As LLMs begin to influence decision-making pipelines in patent examination, IP analytics, or IP litigation support, the lack of tailored evaluation may lead to misleading conclusions about model capabilities. Moreover, the complexity of IP, spanning multiple jurisdictions, languages, legal doctrines, and technical fields, makes it an ideal stress test for measuring LLMs' reasoning, memory, and generation under constraint.

Our work addressed this critical gap by introducing IPBench, a bilingual, multi-dimensional benchmark grounded in real-world IP tasks. The benchmark is built on a four-level task taxonomy adapted from Webb's DoK theory, ranging from low-level recall to high-level creative synthesis. These levels are aligned not just with educational psychology but with actual workflows in patent offices, IP law firms, and technology transfer environments. Unlike prior benchmarks such as PatentEval (Zuo et al., 2024), which focus narrowly on a few patent tasks, IPBench spans 20 tasks across 8 IP mechanisms and includes both comprehension-based and generative formats.

Our empirical findings revealed several important trends and limitations in current LLMs. First, general-purpose models such as GPT-4o and DeepSeek-V3 consistently outperform law- and IP-specific models. This may seem counterintuitive, as vertical models like MoZi-qwen are explicitly trained on legal corpora. However, this underperformance likely results from a combination of overfitting, insufficient general reasoning capabilities, and inadequate coverage of the procedural and generative aspects of IP workflows. Vertical fine-tuning strategies may inadvertently narrow the model's inferential space or induce catastrophic forgetting, degrading performance on multi-step reasoning tasks.

Second, our evaluation of reasoning-oriented models such as DeepSeek-R1 and QwQ-32B revealed that while they do not top the overall leaderboard, they outperform chat-based models on specific tasks requiring arithmetic logic, legal thresholds, or rule-based evaluation (e.g., compensation estimation or damages calculation). This supports the hypothesis that architecture matters: models with symbolic reasoning capabilities have a structural advantage in tasks where correctness hinges on numerical precision or multi-condition rule satisfaction.

Third, our analysis of prompting techniques showed mixed outcomes. Few-shot prompting improves performance on some models and tasks, particularly in instruction-following or retrieval-based scenarios. However, models like Llama3.1-8B-it show no consistent improvement, suggesting sensitivity to prompt design or training data mismatches. The CoT prompting, often touted as a reasoning enhancer, surprisingly leads to performance drops (0.40.6%) across models. Our error analysis attributes this to the injection of spurious reasoning paths and overthinkinga phenomenon also observed in prior work (Zheng et al., 2025; Fan et al., 2025). In domains like IP, where many tasks hinge on memorized definitions or hierarchical rule structures, CoT may actually degrade performance by introducing incorrect logic.

Fourth, the performance on IPC/CPC classification is alarmingly low. Even the best model, DeepSeek-R1, achieves only a 10.8% Exact Match rate. These classification systems are essential for patent analytics, prior art search, and innovation tracking, and failure to resolve them accurately reflects fundamental limitations in LLMs' ability to represent domain hierarchies, align semantic cues with technical structure, and disambiguate overlapping categories. These failures underscore a broader issue in LLM design: current architectures are not optimized for structured symbolic taxonomies or discrete label hierarchies that are common in regulatory domains.

We also introduced LLMScore, an automatic evaluation metric tailored to generative tasks in IP. Unlike traditional metrics such as BLEU and ROUGE, which are inadequate for legal text due to their lack of semantic granularity, LLMScore is based on the LLM-as-a-judge paradigm and evaluates responses across four human-aligned dimensions. Empirical results demonstrate that LLMScore correlates more strongly with human judgments and supports nuanced evaluation of claim and abstract generation, which are central to both patent drafting and retrieval.

Collectively, these results highlight not only current limitations in model generalization and prompting strategies but also the inherent complexity of the IP domain. This complexity arises from its hybrid nature: legal and technical, deterministic and interpretive, global and jurisdiction-specific. Benchmarks like IPBench are thus essential not only for evaluation but for guiding the next phase of model development.

Looking forward, we envision several extensions to IPBench and its applications. The current version focuses on U.S. and Chinese legal frameworks; future iterations will incorporate additional jurisdictions such as the EU and Japan, enabling cross-legal evaluation and comparative reasoning. Moreover, as more IP-specific models become available, IPBench can serve as a testbed for fine-tuning strategies, prompt engineering, and hybrid symbolicneural architectures. More broadly, IPBench offers a blueprint for evaluating LLMs in other complex verticalssuch as medicine, finance, or regulatory compliance, where task diversity, interpretability, and factual correctness are non-negotiable. By operationalizing cognitive depth and legal realism in benchmark design, we hope to catalyze the development of trustworthy, capable, and domain-aligned LLMs.

## K MORE RESULTS

In Section Section K, we present additional results under various experimental settings, covering both Chinese and English. Specifically, Section K.1 reports the overall results on IPBench, Section K.2 presents the results for Chinese questions, Section K.3 covers the results for English questions, and Section K.4 provides detailed results of the LLM-as-a-judge evaluation along with its consistency with human judgments.

### K.1 OVERALL RESULTS

We provide the results of overall performance under the few-shot setting (1-shot, 2-shot, and 3-shot) in Section K.1.1, and the results under the chain-of-thought setting in Section K.1.2.We provide a model performance heatmap as shown in Figure 5, where models are sorted by their overall performance. A redder color indicates that the model on the x-axis outperforms the corresponding model on the y-axis.

#### K.1.1 FEW-SHOT RESULTS

The 1-shot results of IPBench are presented in Table 21 and Table 22, the 2-shot results in Table 23 and Table 24, and the 3-shot results in Table 25 and Table 26.

#### K.1.2 CHAIN-OF-THOUGHT RESULTS

The chain-of-thought results of IPBench are presented in Table 27 and Table 28.

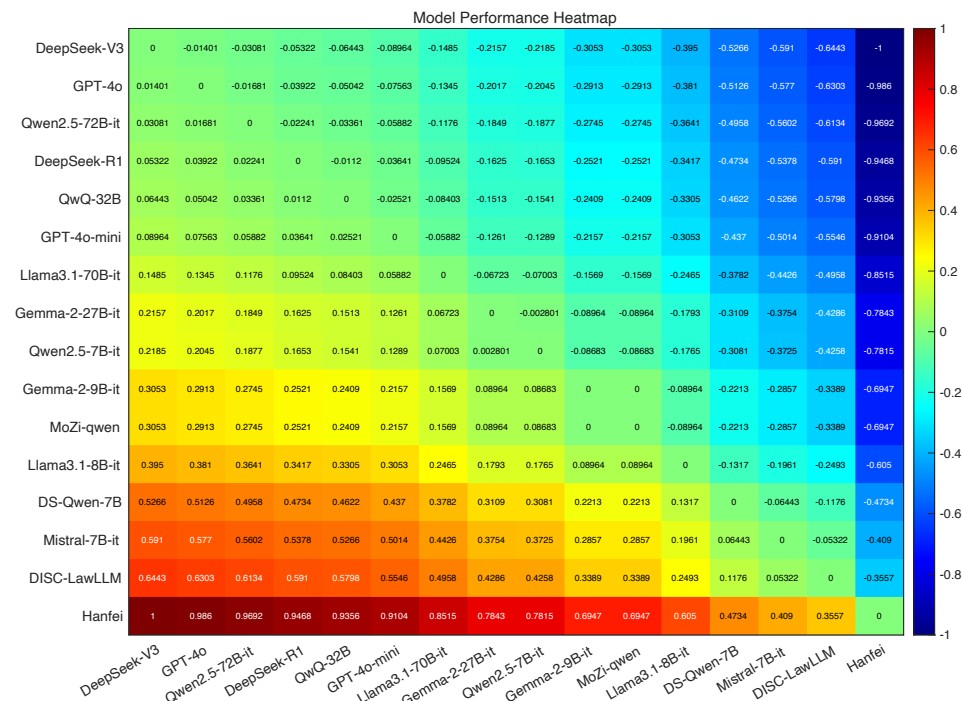

Figure 5: Model performance heatmap.

Table 21: Results of IPBench with 1-shot setting. The best-performing model in each task is in **darker red**, and the second best is in lighter red.

| Model | OA | 1-1 | 1-2 | 1-3 | 1-4 | 1-6 | 1-7 | 2-1 | 2-2 | 2-3 | 2-4 | 2-5 | 3-1 | 3-2 | 3-3 | 3-4 | 4-3 |
|---|---|---|---|---|---|---|---|---|---|---|---|---|---|---|---|---|---|
| GPT-4o-mini | **73.9** | 93.8 | **86.3** | **78.6** | **79.8** | 61.1 | **66.6** | **51.6** | 62.6 | **61.4** | **77.1** | 81.4 | 69.3 | **76.9** | **81.1** | **79.8** | **70.3** |
| Qwen2.5-7B-it | 67.5 | 94.2 | 82.5 | 76.4 | 72.8 | 60.8 | 63.5 | 48.8 | 62.0 | 52.2 | 70.4 | 76.1 | 69.0 | 69.5 | 55.6 | 79.5 | 62.1 |
| Llama3.1-8B-it | 59.3 | 87.0 | 69.7 | 67.4 | 72.0 | 50.7 | 64.6 | 46.0 | 57.6 | 43.7 | 69.4 | 45.9 | 60.0 | 69.8 | 36.8 | 73.2 | 57.1 |
| Gemma-2-9B-it | 66.4 | 89.2 | 74.3 | 71.0 | 73.2 | 55.6 | 60.8 | 50.8 | **65.0** | 50.0 | 72.8 | **82.7** | **70.0** | 70.8 | 53.9 | 75.7 | 69.5 |
| Mistral-7B-it | 54.8 | 79.0 | 61.6 | 63.4 | 59.9 | 43.4 | 52.9 | 44.6 | 57.4 | 36.4 | 61.8 | 62.1 | 60.3 | 48.4 | 39.4 | 67.2 | 57.1 |
| MoZi-qwen | 63.0 | **95.2** | 81.5 | 76.2 | 66.9 | 59.9 | 64.4 | 47.2 | 62.6 | 40.2 | 72.1 | 77.4 | 69.0 | 58.8 | 36.6 | 71.8 | 57.9 |
| DISC-LawLLM | 47.7 | 78.8 | 66.9 | 66.4 | 65.5 | 45.0 | 52.2 | 40.0 | 54.6 | 32.0 | 51.8 | 70.8 | 58.7 | 39.6 | – | 67.3 | 30.7 |
| Hanfei | 28.0 | 37.0 | 29.3 | 28.8 | 31.2 | 33.9 | 43.6 | 26.6 | 37.2 | 35.8 | 29.9 | 24.6 | 37.0 | 22.4 | – | 31.0 | 30.1 |

Table 22: Results of Patent IPC/CPC Classification tasks (1-5-1 and 1-5-2) with 1-shot setting. The best-performing model in each task is in **darker purple**, and the second best is in lighter purple.

| Model | IPC Classification (1-5-1) | | | | CPC Classification (1-5-2) | | | |
|---|---|---|---|---|---|---|---|---|
| | Exact-Match | Section | Class | Subclass | Exact-Match | Section | Class | Subclass |
| GPT-4o-mini | 2.2 | 81.8 | 67.0 | 50.8 | 0.5 | 74.3 | 59.1 | 49.1 |
| DeepSeek-V3 | **15.1** | **86.3** | **75.5** | **60.5** | **7.3** | **86.2** | **74.3** | **65.0** |
| Qwen2.5-7B-it | 2.2 | 73.8 | 57.9 | 42.2 | 0.3 | 67.5 | 48.8 | 37.2 |
| Llama3.1-8B-it | 0.7 | 64.1 | 49.7 | 33.7 | 0.0 | 45.2 | 35.2 | 22.2 |
| MoZi-qwen | 0.4 | 47.0 | 34.6 | 21.9 | 0.0 | 16.5 | 7.8 | 4.3 |

## K.2 CHINESE QUESTIONS RESULTS

In Section Section K.2, we focus on the IPBench results for Chinese questions. We provide the zero-shot results for the Chinese portion of IPBench in SectionK.2.1, the few-shot results in SectionK.2.2, and the chain-of-thought results in Section K.2.3.

Table 23: Results of IPBench with 2-shot setting. The best-performing model in each task is in darker red , and the second best is in lighter red .

| Model | OA | 1-1 | 1-2 | 1-3 | 1-4 | 1-6 | 1-7 | 2-1 | 2-2 | 2-3 | 2-4 | 2-5 | 3-1 | 3-2 | 3-3 | 3-4 | 4-3 |
|---|---|---|---|---|---|---|---|---|---|---|---|---|---|---|---|---|---|
| GPT-4o-mini | **74.0** | 94.2 | **87.9** | 77.0 | **80.2** | 60.6 | **66.6** | **52.0** | 59.6 | **63.3** | **79.4** | 83.4 | 68.0 | **78.6** | 80.7 | 76.7 | **72.5** |
| Qwen2.5-7B-it | 69.3 | 94.8 | 83.3 | 76.0 | 77.0 | 59.9 | 65.0 | 47.8 | 60.4 | 56.0 | 71.8 | 83.7 | 67.7 | 69.5 | 65.9 | **81.5** | 57.7 |
| Llama3.1-8B-it | 59.3 | 85.8 | 67.3 | 68.4 | 69.6 | 52.3 | 61.7 | 48.0 | 56.6 | 45.6 | 71.4 | 66.8 | 60.0 | 61.4 | 37.8 | 69.8 | 53.9 |
| Gemma-2-9B-it | 67.3 | 89.0 | 76.3 | 71.4 | 72.2 | 56.3 | 61.1 | 51.4 | **62.8** | 52.2 | 74.7 | **84.7** | 66.7 | 72.4 | 57.4 | 78.0 | 70.3 |
| Mistral-7B-it | 57.2 | 79.8 | 64.1 | 64.6 | 61.5 | 44.4 | 53.8 | 47.0 | 56.6 | 38.0 | 65.1 | 73.1 | 61.8 | 48.7 | 47.1 | 67.5 | 56.5 |
| MoZi-qwen | 66.8 | **96.0** | 84.1 | **77.2** | 75.0 | **63.6** | 64.2 | 48.4 | 61.2 | 43.0 | 75.1 | 82.4 | **72.0** | 64.6 | 49.4 | 80.8 | 55.1 |
| DISC-LawLLM | 56.7 | 79.2 | 67.1 | 66.6 | 65.3 | 50.2 | 51.3 | 39.6 | 53.8 | 31.3 | 58.1 | 78.1 | 57.7 | 42.2 | – | 71.5 | 37.1 |
| Hanfei | 32.5 | 36.0 | 24.7 | 35.8 | 33.7 | 35.3 | 33.0 | 25.4 | 37.0 | 32.6 | 26.6 | 49.2 | 42.3 | 22.1 | – | 32.2 | 25.7 |

Table 24: Results of Patent IPC/CPC Classification tasks (1-5-1 and 1-5-2) with 2-shot setting. The best-performing model in each task is in **darker purple** , and the second best is in lighter purple .

| Model | IPC Classification (1-5-1) | | | | CPC Classification (1-5-2) | | | |
|---|---|---|---|---|---|---|---|---|
| | Exact-Match | Section | Class | Subclass | Exact-Match | Section | Class | Subclass |
| GPT-4o-mini | 2.3 | 82.2 | 68.0 | 51.5 | 0.2 | 76.3 | 61.6 | 51.6 |
| DeepSeek-V3 | **15.1** | **86.7** | **76.1** | **60.6** | **7.2** | **86.5** | **73.3** | **65.7** |
| Qwen2.5-7B-it | 2.5 | 78.2 | 62.4 | 46.2 | 0.3 | 68.5 | 51.2 | 38.8 |
| Llama3.1-8B-it | 1.1 | 59.7 | 44.7 | 29.2 | 0.0 | 63.3 | 45.7 | 26.7 |
| MoZi-qwen | 0.6 | 56.6 | 41.7 | 26.8 | 0.2 | 32.3 | 17.3 | 9.3 |

Table 25: Results of IPBench with 3-shot setting. The best-performing model in each task is in darker red , and the second best is in lighter red .

| Model | OA | 1-1 | 1-2 | 1-3 | 1-4 | 1-6 | 1-7 | 2-1 | 2-2 | 2-3 | 2-4 | 2-5 | 3-1 | 3-2 | 3-3 | 3-4 | 4-3 |
|---|---|---|---|---|---|---|---|---|---|---|---|---|---|---|---|---|---|
| GPT-4o-mini | **74.7** | 94.4 | **87.5** | 79.6 | 80.0 | 63.3 | 68.8 | 52.4 | 58.6 | **63.6** | 80.1 | 82.7 | 70.3 | **77.9** | 80.0 | 80.0 | **75.0** |
| Qwen2.5-7B-it | 70.6 | 94.2 | 83.3 | 73.6 | 76.0 | 62.2 | 68.2 | 50.8 | **62.2** | 51.3 | 74.8 | 84.7 | 70.7 | 72.4 | 68.5 | **82.2** | 60.3 |
| Llama3.1-8B-it | 59.4 | 87.4 | 67.0 | 66.4 | 69.8 | 52.7 | 63.7 | 45.6 | 55.4 | 43.0 | 66.4 | 75.4 | 61.0 | 62.7 | 36.7 | 70.5 | 56.3 |
| Gemma-2-9B-it | 67.4 | 89.4 | 76.1 | 70.6 | 70.6 | 56.6 | 62.5 | 51.2 | **62.2** | 51.6 | 76.4 | **85.1** | 68.3 | 69.5 | 58.7 | 76.3 | 70.7 |
| Mistral-7B-it | 56.5 | 80.6 | 63.9 | 62.8 | 61.7 | 45.9 | 54.4 | 47.4 | 56.0 | 36.1 | 64.1 | 73.8 | 63.3 | 50.3 | 40.1 | 67.8 | 58.1 |
| MoZi-qwen | 65.3 | **96.2** | 83.5 | 77.2 | 76.0 | 62.4 | 65.1 | 49.4 | 61.8 | 40.5 | 75.8 | 80.4 | **72.7** | 62.0 | 38.3 | 79.0 | 57.3 |
| DISC-LawLLM | 57.4 | 83.8 | 67.3 | 64.6 | 66.7 | 53.6 | 52.0 | 41.2 | 54.8 | 29.4 | 62.1 | 74.4 | 60.0 | 41.2 | – | 67.5 | 38.5 |
| Hanfei | 29.9 | 32.0 | 28.9 | 26.2 | 28.4 | 31.7 | 31.6 | 23.6 | 36.2 | 29.1 | 22.6 | 30.6 | 41.0 | 24.4 | – | 26.3 | 34.5 |

Table 26: Results of Patent IPC/CPC Classification tasks (1-5-1 and 1-5-2) with 3-shot setting. The best-performing model in each task is in **darker purple** , and the second best is in lighter purple .

| Model | IPC Classification (1-5-1) | | | | CPC Classification (1-5-2) | | | |
|---|---|---|---|---|---|---|---|---|
| | Exact-Match | Section | Class | Subclass | Exact-Match | Section | Class | Subclass |
| GPT-4o-mini | 2.0 | 82.5 | 67.8 | 50.9 | 0.3 | 80.1 | 65.3 | 54.1 |
| DeepSeek-V3 | **15.6** | **87.1** | **76.2** | **61.2** | **7.7** | **85.7** | **73.3** | **64.7** |
| Qwen2.5-7B-it | 2.3 | 78.8 | 62.7 | 46.8 | 0.5 | 68.3 | 50.8 | 38.8 |
| Llama3.1-8B-it | 1.2 | 65.6 | 48.9 | 32.8 | 0.0 | 64.8 | 45.8 | 29.8 |
| MoZi-qwen | 1.0 | 70.6 | 51.3 | 34.2 | 0.0 | 24.2 | 12.8 | 7.7 |

### K.2.1 ZERO-SHOT RESULTS

The zero-shot results for the Chinese portion of IPBench are shown in Table 29, Table 30 and Table 31. Since the Patent CPC Classification task (1-5-2) only includes English questions, Table 30 does not include it.

Table 27: Results of IPBench with chain-of-thought setting. The best-performing model in each task is in  darker red , and the second best is in  lighter red .

| Model | OA | 1-1 | 1-2 | 1-3 | 1-4 | 1-6 | 1-7 | 2-1 | 2-2 | 2-3 | 2-4 | 2-5 | 3-1 | 3-2 | 3-3 | 3-4 | 3-5 | 4-3 |
|---|---|---|---|---|---|---|---|---|---|---|---|---|---|---|---|---|---|---|
| GPT-4o-mini | 72.0 | 94.4 | 85.9 | 78.0 | 80.4 | 59.9 | 67.3 | 51.4 | 62.6 | 62.0 | 74.8 | 80.4 | 65.7 | 71.1 | 81.1 | 78.8 | 44.9 | 66.9 |
| Qwen2.5-7B-it | 67.6 | 89.0 | 82.9 | 75.2 | 76.2 | 57.3 | 63.0 | 48.0 | 64.2 | 58.2 | 73.8 | 79.7 | 66.7 | 70.1 | 65.1 | 78.5 | 44.3 | 58.9 |
| Llama3.1-8B-it | 61.3 | 84.1 | 69.5 | 67.6 | 70.2 | 53.6 | 59.5 | 49.4 | 60.6 | 45.9 | 66.4 | 71.8 | 62.3 | 59.4 | 54.8 | 73.0 | 43.6 | 54.1 |
| Gemma-2-9B-it | 61.7 | 87.0 | 72.3 | 65.4 | 66.3 | 54.1 | 55.7 | 51.0 | 64.8 | 47.8 | 71.1 | 76.4 | 66.3 | 67.9 | 57.9 | 73.5 | – | 65.1 |
| Mistral-7B-it | 54.3 | 80.6 | 63.3 | 63.6 | 62.5 | 43.6 | 54.0 | 42.4 | 54.6 | 44.3 | 64.1 | 65.8 | 56.7 | 51.0 | 41.0 | 66.5 | 36.3 | 47.9 |
| MoZi-qwen | 60.2 | 93.0 | 79.9 | 72.0 | 65.3 | 50.2 | 61.9 | 45.2 | 52.4 | 45.2 | 66.8 | 72.4 | 58.3 | 62.7 | 49.2 | 71.0 | 43.9 | 44.1 |
| DISC-LawLLM | 37.3 | 65.4 | 57.4 | 48.6 | 39.3 | 42.8 | 41.4 | 25.4 | 34.8 | 25.9 | 32.2 | 62.8 | 26.7 | 25.7 | 17.5 | 39.0 | – | 30.5 |
| Hanfei | 29.9 | 42.0 | 28.5 | 32.4 | 34.1 | 30.6 | 26.8 | 28.8 | 29.6 | 21.5 | 25.9 | 24.3 | 34.3 | 26.6 | 24.6 | 31.7 | – | 35.3 |

Table 28: Results of Patent IPC/CPC Classification tasks (1-5-1 and 1-5-2) with chain-of-thought setting. The best-performing model in each task is in  darker purple , and the second best is in  lighter purple .

| Model | IPC Classification (1-5-1) | | | | CPC Classification (1-5-2) | | | |
|---|---|---|---|---|---|---|---|---|
| | Exact-Match | Section | Class | Subclass | Exact-Match | Section | Class | Subclass |
| GPT-4o-mini | 0.2 | 80.4 | 67.4 | 51.6 | 0.0 | 76.8 | 63.0 | 52.5 |
| DeepSeek-V3 | 1.3 | 82.3 | 72.0 | 57.4 | 1.0 | 83.3 | 70.7 | 63.0 |
| Qwen2.5-7B-it | 1.8 | 74.3 | 60.4 | 42.0 | 0.5 | 60.2 | 46.0 | 35.7 |
| Llama3.1-8B-it | 0.9 | 67.0 | 50.5 | 32.4 | 0.2 | 64.0 | 44.8 | 29.5 |
| MoZi-qwen | 0.3 | 22.4 | 17.2 | 12.6 | 0.0 | 7.7 | 2.8 | 1.8 |

Table 29: Chinese questions results of IPBench. The best-performing model in each task is in  darker red , and the second best is in  lighter red . The model DS-Qwen refers to DeepSeek-R1-Distill-Qwen, while the suffix *it* indicates the Instruct version of the model. OA denotes the overall average accuracy on the choice tasks.

| Model | OA | 1-1 | 1-2 | 1-3 | 1-4 | 1-6 | 1-7 | 2-1 | 2-2 | 2-3 | 2-4 | 2-5 | 3-1 | 3-2 | 3-3 | 3-4 | 4-3 |
|---|---|---|---|---|---|---|---|---|---|---|---|---|---|---|---|---|---|
| GPT-4o | 77.7 | 95.0 | 92.4 | 82.0 | 80.6 | 73.1 | 73.7 | 50.8 | 73.7 | 67.3 | 70.0 | 84.1 | 66.9 | 63.5 | 80.2 | 82.9 | 79.0 |
| GPT-4o-mini | 74.2 | 91.9 | 86.2 | 76.9 | 80.6 | 65.1 | 66.6 | 44.4 | 66.2 | 57.7 | 66.9 | 83.4 | 58.8 | 80.5 | 82.2 | 77.9 | 72.6 |
| DeepSeek-V3 | 78.7 | 97.3 | 90.2 | 87.1 | 83.0 | 76.3 | 72.1 | 48.0 | 74.1 | 65.4 | 70.5 | 84.1 | 67.5 | 71.1 | 78.2 | 83.9 | 84.8 |
| Qwen3 | 73.2 | 95.4 | 85.5 | 75.9 | 79.8 | 68.3 | 68.2 | 43.6 | 74.1 | 55.8 | 65.5 | 82.7 | 62.5 | 73.0 | 67.4 | 78.3 | 77.4 |
| Qwen2.5-72B-it | 77.9 | 96.5 | 92.4 | 82.0 | 81.8 | 69.8 | 70.5 | 48.0 | 81.1 | 62.2 | 74.8 | 82.1 | 67.5 | 68.6 | 81.8 | 79.3 | 82.6 |
| Qwen2.5-7B-it | 70.8 | 93.8 | 85.9 | 72.4 | 77.1 | 64.5 | 63.6 | 47.6 | 76.3 | 51.9 | 66.9 | 77.1 | 60.6 | 66.7 | 67.6 | 78.3 | 68.0 |
| Llama3.1-70B-it | 70.8 | 91.5 | 81.9 | 71.4 | 76.2 | 66.6 | 66.9 | 47.2 | 71.9 | 48.7 | 66.9 | 81.1 | 66.3 | 66.7 | 63.0 | 78.3 | 77.4 |
| Llama3.1-8B-it | 65.1 | 88.4 | 73.6 | 65.7 | 78.6 | 63.6 | 58.4 | 50.4 | 69.3 | 41.7 | 62.6 | 75.7 | 58.8 | 57.2 | 55.8 | 79.3 | 55.8 |
| Gemma-2-27B-it | 69.2 | 88.8 | 78.3 | 66.0 | 75.4 | 60.4 | 60.1 | 47.2 | 77.2 | 53.2 | 66.2 | 81.1 | 66.3 | 61.6 | 62.8 | 80.2 | 76.8 |
| Gemma-2-9B-it | 65.5 | 91.5 | 75.4 | 68.4 | 63.5 | 67.5 | 57.5 | 45.2 | 73.7 | 45.5 | 59.7 | 80.4 | 62.5 | 57.9 | 51.4 | 73.7 | 68.9 |
| Mistral-7B-it | 54.1 | 74.1 | 58.3 | 55.4 | 58.7 | 44.1 | 51.6 | 45.6 | 60.5 | 40.4 | 48.2 | 67.0 | 53.1 | 29.6 | 47.4 | 60.1 | 59.5 |
| MoZi-qwen | 67.9 | 93.1 | 86.2 | 73.1 | 59.1 | 69.2 | 64.6 | 48.4 | 64.0 | 41.0 | 56.8 | 76.4 | 60.0 | 57.9 | 64.0 | 82.0 | 65.5 |
| DISC-LawLLM | 55.0 | 86.9 | 69.2 | 64.6 | 63.5 | 64.5 | 49.7 | 38.0 | 66.7 | 40.4 | 46.8 | 64.8 | 50.6 | 37.7 | 28.4 | 72.8 | 41.5 |
| Hanfei | 39.9 | 65.3 | 46.7 | 50.3 | 53.2 | 45.9 | 51.3 | 28.4 | 43.4 | 26.9 | 36.7 | 49.2 | 41.9 | 20.1 | 10.0 | 45.2 | 35.1 |
| DeepSeek-R1 | 76.6 | 95.8 | 91.3 | 84.4 | 79.4 | 74.0 | 73.1 | 44.8 | 76.8 | 66.0 | 67.6 | 85.4 | 59.4 | 78.6 | 64.8 | 84.3 | 82.9 |
| DS-Qwen-7B | 58.2 | 79.2 | 59.1 | 50.7 | 63.9 | 51.2 | 49.4 | 44.0 | 53.5 | 38.5 | 49.6 | 65.5 | 50.6 | 64.8 | 63.6 | 63.6 | 63.7 |
| QwQ-32B | 76.4 | 94.6 | 93.1 | 79.3 | 79.8 | 76.9 | 73.7 | 50.0 | 77.2 | 66.7 | 74.8 | 85.4 | 64.4 | 80.5 | 65.6 | 77.4 | 75.0 |

### K.2.2 Few-shot Results

The 1-shot results for the Chinese portion of IPBench are shown in Table 32 and Table 33, the 2-shot results in Table 34 and Table 35 and the 3-shot results in Table 36 and Table 37.

Table 30: Results of Chinese Patent IPC Classification task (1-5-1). The best-performing model in each task is in darker purple , and the second best is in lighter purple .

| Model | IPC Classification (1-5-1) | | | |
|---|---|---|---|---|
| | Exact-Match | Section | Class | Subclass |
| GPT-4o | 8.0 | 76.4 | 70.5 | 62.1 |
| GPT-4o-mini | 1.0 | 76.0 | 66.1 | 53.7 |
| DeepSeek-V3 | **20.2** | 80.4 | 72.8 | 66.3 |
| Qwen3 | 3.4 | 76.8 | 61.0 | 47.6 |
| Qwen2.5-72B-it | 8.0 | 79.8 | 71.0 | 62.3 |
| Qwen2.5-7B-it | 2.3 | 68.6 | 57.9 | 46.3 |
| Llama3.1-70B-it | 5.0 | 77.9 | 64.8 | 55.2 |
| Llama3.1-8B-it | 0.4 | 65.1 | 51.8 | 34.1 |
| Gemma-2-27B-it | 1.0 | 72.4 | 56.4 | 45.9 |
| Gemma-2-9B-it | 0.0 | 66.3 | 51.0 | 34.9 |
| Mistral-7B-it | 0.0 | 49.9 | 26.1 | 16.8 |
| MoZi-qwen | 0.4 | 34.3 | 25.5 | 16.8 |
| DISC-LawLLM | 0.0 | 51.2 | 30.9 | 15.5 |
| Hanfei | 0.0 | 17.2 | 4.2 | 0.2 |
| DeepSeek-R1 | 19.6 | **83.2** | **75.4** | **67.6** |
| DS-Qwen-7B | 0.0 | 29.3 | 8.5 | 1.2 |
| QwQ-32B | 3.9 | 80.9 | 71.5 | 60.7 |

Table 31: Results of Chinese generation tasks (4-1 and 4-2). The best-performing model in each task is in darker blue , and the second best is in lighter blue . R-L refers to ROUGE-L, BS refers to BERTScore, LLMScore refers to GPT-4 judge score (1-10), Avg Tokens # denotes the average number of tokens in the generated text, and Avg DC # indicates the average number of generated dependent claims.

| Model | Abstract Generation (4-1) | | | | | Dependent Claim Generation (4-2) | | | | | |
|---|---|---|---|---|---|---|---|---|---|---|---|
| | BLEU | R-L | BS | LLMScore (1-10) | Tokens # (167.9) | BLEU | R-L | BS | LLMScore (1-10) | Tokens # (457.8) | DC # (4.1) |
| GPT-4o | 17.7 | 34.8 | 91.0 | 8.77 | 278.7 | 12.7 | 25.0 | 90.3 | 6.30 | 658.3 | 6.8 |
| GPT-4o-mini | 17.9 | 35.2 | 90.9 | 8.51 | 224.9 | **15.0** | 28.6 | 90.3 | 6.09 | 497.9 | 11.8 |
| DeepSeek-V3 | 12.4 | 29.9 | 90.5 | **8.92** | 273.8 | 10.8 | 23.4 | 90.0 | **7.36** | 799.7 | 14.9 |
| Qwen2.5-72B-it | 13.7 | 30.8 | 90.8 | 8.50 | 379.6 | 9.6 | 20.3 | **90.6** | 6.60 | 1374.2 | 17.4 |
| Qwen2.5-7B-it | 20.5 | 36.8 | 91.1 | 8.29 | 190.4 | 11.0 | 21.5 | **90.6** | 5.64 | 3453.3 | 43.5 |
| Llama3.1-70B-it | 24.9 | 40.3 | 91.1 | 7.89 | 261.2 | 7.3 | 19.8 | 89.7 | 4.99 | 4045.2 | 43.0 |
| Llama3.1-8B-it | 11.9 | 26.7 | 90.2 | 7.16 | 554.9 | 4.7 | 13.1 | 90.1 | 3.09 | 4932.2 | 36.1 |
| Gemma-2-27B-it | 14.7 | 30.6 | 90.2 | 7.74 | 215.5 | 6.5 | 19.1 | 88.6 | 5.46 | 678.5 | 3.1 |
| Gemma-2-9B-it | 17.7 | 33.9 | 90.6 | 8.07 | 247.8 | 5.7 | 20.8 | 88.3 | 5.15 | 577.1 | 5.8 |
| Mistral-7B-it | 11.8 | 26.2 | 90.4 | 7.24 | 479.5 | 3.5 | 10.7 | 88.8 | 2.13 | 4968.3 | 44.0 |
| MoZi-qwen | **31.3** | **53.4** | **91.6** | 7.91 | 335.9 | 7.7 | 28.6 | 90.3 | 4.28 | 8306.6 | 59.8 |
| DeepSeek-R1 | 8.9 | 28.0 | 89.3 | 7.89 | 671.0 | 9.3 | 26.8 | 81.8 | 7.17 | 1374.2 | 15.8 |
| DS-Qwen-7B | 12.4 | 36.1 | 90.3 | 7.80 | 918.2 | 5.4 | **32.5** | 81.8 | 3.69 | 9878.2 | 89.8 |
| QwQ-32B | 11.1 | 33.6 | 90.2 | 8.84 | 1403.8 | 5.4 | 22.6 | 80.8 | 7.05 | 5360.0 | 37.8 |

### K.2.3 CHAIN-OF-THOUGHT RESULTS

The chain-of-thought results for the Chinese portion of IPBench are presented in Table 38 and Table 39.

Table 32: Chinese questions results of IPBench with 1-shot setting. The best-performing model in each task is in darker red , and the second best is in lighter red .

| Model | OA | 1-1 | 1-2 | 1-3 | 1-4 | 1-6 | 1-7 | 2-1 | 2-2 | 2-3 | 2-4 | 2-5 | 3-1 | 3-2 | 3-3 | 3-4 | 4-3 |
|---|---|---|---|---|---|---|---|---|---|---|---|---|---|---|---|---|---|
| GPT-4o-mini | 73.1 | 91.5 | 85.9 | 72.5 | 76.6 | 67.2 | 65.9 | 43.6 | 68.9 | 53.2 | 66.9 | 81.4 | 63.1 | 74.2 | 79.2 | 77.0 | 76.5 |
| Qwen2.5-7B-it | 69.9 | 94.6 | 86.2 | 72.5 | 73.8 | 71.0 | 66.6 | 43.6 | 71.5 | 50.0 | 59.7 | 76.1 | 58.1 | 64.2 | 60.0 | 82.5 | 70.1 |
| Llama3.1-8B-it | 58.9 | 81.5 | 64.1 | 60.9 | 73.0 | 59.2 | 61.7 | 43.6 | 69.3 | 38.4 | 61.2 | 45.9 | 56.3 | 64.8 | 38.0 | 73.2 | 65.6 |
| Gemma-2-9B-it | 68.1 | 86.9 | 74.3 | 67.4 | 69.4 | 64.8 | 60.7 | 46.8 | 79.4 | 45.5 | 67.6 | 82.7 | 64.4 | 64.8 | 56.8 | 74.7 | 79.0 |
| Mistral-7B-it | 53.7 | 71.0 | 56.2 | 58.5 | 59.5 | 47.0 | 52.3 | 38.8 | 66.2 | 42.3 | 43.2 | 62.1 | 53.8 | 35.2 | 41.4 | 65.4 | 61.6 |
| MoZi-qwen | 67.5 | 95.0 | 87.3 | 73.8 | 61.9 | 71.0 | 63.0 | 44.8 | 71.9 | 39.4 | 64.0 | 77.4 | 63.1 | 60.4 | 48.2 | 80.7 | 71.7 |
| DISC-LawLLM | 55.7 | 79.2 | 68.8 | 62.9 | 69.1 | 51.5 | 44.8 | 38.8 | 56.6 | 35.9 | 46.0 | 70.8 | 55.6 | 38.4 | – | 75.1 | 31.1 |
| Hanfei | 31.6 | 34.0 | 23.9 | 23.1 | 33.7 | 33.1 | 49.0 | 24.0 | 49.1 | 35.9 | 29.5 | 24.6 | 38.1 | 18.9 | – | 33.6 | 27.4 |

Table 33: Results of Chinese Patent IPC Classification task (1-5-1) with 1-shot setting. The best-performing model in each task is in darker purple , and the second best is in lighter purple .

| Model | IPC Classification (1-5-1) | | | |
|---|---|---|---|---|
| | Exact-Match | Section | Class | Subclass |
| GPT-4o-mini | 3.6 | 76.8 | 67.1 | 54.9 |
| DeepSeek-V3 | 30.2 | 82.4 | 75.4 | 68.9 |
| Qwen2.5-7B-it | 3.8 | 59.6 | 50.9 | 39.6 |
| Llama3.1-8B-it | 0.8 | 45.9 | 36.4 | 25.5 |
| MoZi-qwen | 0.0 | 11.6 | 8.2 | 3.6 |

Table 34: Chinese questions results of IPBench with 2-shot setting. The best-performing model in each task is in darker red , and the second best is in lighter red .

| Model | OA | 1-1 | 1-2 | 1-3 | 1-4 | 1-6 | 1-7 | 2-1 | 2-2 | 2-3 | 2-4 | 2-5 | 3-1 | 3-2 | 3-3 | 3-4 | 4-3 |
|---|---|---|---|---|---|---|---|---|---|---|---|---|---|---|---|---|---|
| GPT-4o-mini | 73.5 | 91.9 | 87.3 | 69.1 | 76.2 | 66.3 | 67.2 | 44.8 | 64.5 | 58.3 | 69.8 | 83.4 | 60.0 | 73.0 | 81.0 | 73.7 | 78.7 |
| Qwen2.5-7B-it | 70.8 | 95.0 | 86.2 | 70.8 | 76.2 | 69.5 | 68.8 | 39.6 | 68.9 | 51.9 | 63.3 | 83.7 | 54.4 | 60.4 | 67.2 | 83.0 | 65.6 |
| Llama3.1-8B-it | 58.9 | 79.5 | 63.4 | 63.6 | 69.8 | 58.3 | 59.1 | 46.8 | 64.0 | 36.5 | 65.5 | 66.8 | 53.1 | 54.7 | 38.0 | 66.4 | 61.3 |
| Gemma-2-9B-it | 68.7 | 86.5 | 76.1 | 66.7 | 70.6 | 65.4 | 60.4 | 46.8 | 75.9 | 46.8 | 68.4 | 84.7 | 61.3 | 63.5 | 58.6 | 77.9 | 79.6 |
| Mistral-7B-it | 56.6 | 73.0 | 58.0 | 58.5 | 63.1 | 49.1 | 52.6 | 44.4 | 63.2 | 42.3 | 51.1 | 73.1 | 55.6 | 37.1 | 48.6 | 66.4 | 61.0 |
| MoZi-qwen | 70.3 | 95.8 | 87.3 | 75.2 | 76.6 | 74.9 | 65.3 | 47.6 | 71.1 | 43.6 | 67.6 | 82.4 | 68.1 | 61.0 | 52.2 | 83.4 | 70.1 |
| DISC-LawLLM | 57.6 | 76.5 | 66.3 | 62.9 | 68.3 | 56.8 | 45.8 | 38.0 | 56.1 | 36.5 | 51.1 | 78.1 | 54.4 | 34.6 | – | 74.7 | 43.9 |
| Hanfei | 31.7 | 30.5 | 19.6 | 40.1 | 30.2 | 35.2 | 34.4 | 25.6 | 35.5 | 22.4 | 25.9 | 49.2 | 44.4 | 12.0 | – | 32.7 | 32.3 |

Table 35: Results of Chinese Patent IPC Classification task (1-5-1) with 2-shot setting. The best-performing model in each task is in darker purple , and the second best is in lighter purple .

| Model | IPC Classification (1-5-1) | | | |
|---|---|---|---|---|
| | Exact-Match | Section | Class | Subclass |
| GPT-4o-mini | 2.5 | 75.4 | 65.9 | 54.5 |
| DeepSeek-V3 | 29.4 | 82.8 | 76.0 | 69.1 |
| Qwen2.5-7B-it | 3.6 | 70.5 | 60.0 | 47.6 |
| Llama3.1-8B-it | 1.0 | 34.9 | 28.4 | 18.9 |
| MoZi-qwen | 0.4 | 29.9 | 23.1 | 15.4 |

Table 36: Chinese questions results of IPBench with 3-shot setting. The best-performing model in each task is in darker red , and the second best is in lighter red .

| Model | OA | 1-1 | 1-2 | 1-3 | 1-4 | 1-6 | 1-7 | 2-1 | 2-2 | 2-3 | 2-4 | 2-5 | 3-1 | 3-2 | 3-3 | 3-4 | 4-3 |
|---|---|---|---|---|---|---|---|---|---|---|---|---|---|---|---|---|---|
| GPT-4o-mini | 74.7 | 91.5 | 87.3 | 73.5 | 77.0 | 70.1 | 67.9 | 47.2 | 64.0 | 59.6 | 71.9 | 82.7 | 62.5 | 73.6 | 80.6 | 79.3 | 79.3 |
| Qwen2.5-7B-it | 72.1 | 95.0 | 84.4 | 69.4 | 77.0 | 71.6 | 69.8 | 45.6 | 73.7 | 43.6 | 65.5 | 84.7 | 58.1 | 65.4 | 69.6 | 83.4 | 68.9 |
| Llama3.1-8B-it | 59.3 | 81.1 | 60.9 | 59.5 | 68.7 | 60.9 | 64.0 | 47.6 | 57.5 | 32.1 | 58.3 | 75.4 | 53.1 | 55.3 | 37.8 | 67.7 | 64.6 |
| Gemma-2-9B-it | 68.7 | 87.6 | 75.0 | 66.0 | 69.0 | 64.5 | 63.3 | 47.6 | 75.0 | 48.1 | 69.8 | 85.1 | 63.8 | 61.6 | 60.0 | 76.5 | 77.4 |
| Mistral-7B-it | 56.3 | 72.2 | 59.8 | 56.8 | 65.1 | 51.2 | 53.6 | 45.6 | 62.3 | 41.0 | 52.5 | 73.8 | 58.1 | 39.0 | 43.4 | 65.4 | 60.7 |
| MoZi-qwen | 69.2 | 95.8 | 87.0 | 74.5 | 75.8 | 72.5 | 65.3 | 51.2 | 70.6 | 37.8 | 68.4 | 80.4 | 67.5 | 55.4 | 47.4 | 81.6 | 72.9 |
| DISC-LawLLM | 58.0 | 81.9 | 69.2 | 60.9 | 70.6 | 61.0 | 47.4 | 40.4 | 53.5 | 35.9 | 53.2 | 74.4 | 58.1 | 34.0 | – | 67.7 | 43.9 |
| Hanfei | 30.1 | 27.8 | 22.8 | 28.9 | 25.8 | 34.3 | 34.7 | 27.2 | 41.7 | 24.4 | 20.9 | 30.6 | 43.1 | 22.0 | – | 27.2 | 39.9 |

Table 37: Results of Chinese Patent IPC Classification task (1-5-1) with 3-shot setting. The best-performing model in each task is in darker purple , and the second best is in lighter purple .

| Model | IPC Classification (1-5-1) | | | |
|---|---|---|---|---|
| | Exact-Match | Section | Class | Subclass |
| GPT-4o-mini | 2.3 | 76.4 | 66.3 | 55.6 |
| DeepSeek-V3 | 29.4 | 83.8 | 76.9 | 70.8 |
| Qwen2.5-7B-it | 3.6 | 71.1 | 60.2 | 48.0 |
| Llama3.1-8B-it | 0.4 | 46.1 | 33.9 | 23.2 |
| MoZi-qwen | 0.6 | 59.4 | 42.1 | 31.4 |

Table 38: Chinese questions results of IPBench with chain-of-thought setting. The best-performing model in each task is in darker red , and the second best is in lighter red .

| Model | OA | 1-1 | 1-2 | 1-3 | 1-4 | 1-6 | 1-7 | 2-1 | 2-2 | 2-3 | 2-4 | 2-5 | 3-1 | 3-2 | 3-3 | 3-4 | 4-3 |
|---|---|---|---|---|---|---|---|---|---|---|---|---|---|---|---|---|---|
| GPT-4o-mini | 72.4 | 92.3 | 84.1 | 70.8 | 77.8 | 65.4 | 66.9 | 45.2 | 68.9 | 59.6 | 60.4 | 80.4 | 57.5 | 66.0 | 81.00 | 75.1 | 71.3 |
| Qwen2.5-7B-it | 70.9 | 93.4 | 86.2 | 72.8 | 79.4 | 63.0 | 65.3 | 46.0 | 74.1 | 55.1 | 61.9 | 79.7 | 59.4 | 69.8 | 67.00 | 78.3 | 67.1 |
| Llama3.1-8B-it | 62.3 | 83.4 | 63.8 | 62.9 | 66.7 | 58.3 | 55.8 | 52.4 | 66.7 | 39.1 | 51.8 | 71.8 | 58.8 | 51.6 | 62.20 | 71.4 | 61.0 |
| Gemma-2-9B-it | 63.1 | 85.3 | 69.9 | 57.8 | 63.1 | 59.8 | 52.3 | 46.8 | 75.0 | 49.4 | 59.7 | 76.4 | 59.4 | 61.0 | 53.80 | 67.3 | 70.7 |
| Mistral-7B-it | 54.5 | 76.1 | 55.8 | 57.8 | 60.3 | 48.8 | 50.0 | 45.6 | 59.7 | 45.5 | 51.1 | 65.8 | 56.9 | 39.6 | 44.80 | 61.8 | 54.3 |
| MoZi-qwen | 60.5 | 91.5 | 80.4 | 64.6 | 57.9 | 55.3 | 60.7 | 39.2 | 54.8 | 44.9 | 54.0 | 72.4 | 45.0 | 55.3 | 58.00 | 65.9 | 45.4 |
| DISC-LawLLM | 52.5 | 83.8 | 66.3 | 62.9 | 63.5 | 61.0 | 49.0 | 37.2 | 58.8 | 25.6 | 48.2 | 62.8 | 43.8 | 37.7 | 30.40 | 60.4 | 41.5 |
| Hanfei | 29.1 | 44.0 | 25.0 | 35.0 | 37.7 | 34.9 | 26.3 | 24.8 | 25.9 | 22.4 | 25.2 | 24.3 | 35.0 | 17.6 | 23.00 | 24.4 | 38.4 |

Table 39: Results of Chinese Patent IPC Classification task (1-5-1) with chain-of-thought setting. The best-performing model in each task is in darker purple , and the second best is in lighter purple .

| Model | IPC Classification (1-5-1) | | | |
|---|---|---|---|---|
| | Exact-Match | Section | Class | Subclass |
| GPT-4o-mini | 0.4 | 75.4 | 67.6 | 56.0 |
| DeepSeek-V3 | 2.5 | 77.1 | 70.9 | 64.2 |
| Qwen2.5-7B-it | 1.7 | 58.7 | 50.1 | 36.4 |
| Llama3.1-8B-it | 0.2 | 56.6 | 41.3 | 26.7 |
| MoZi-qwen | 0.0 | 3.6 | 2.9 | 2.1 |

## K.3 ENGLISH QUESTIONS RESULTS

In Section Section K.3, we focus on the IPBench results for English questions. We provide the zero-shot results for the English portion of IPBench in SectionK.3.1, the few-shot results in SectionK.3.2, and the chain-of-thought results in Section K.3.3.

### K.3.1 ZERO-SHOT RESULTS

The zero-shot results for the English portion of IPBench are shown in Table 40, Table 41 and Table 42.

Table 40: English questions results of IPBench. The best-performing model in each task is in darker red , and the second best is in lighter red . The model DS-Qwen refers to DeepSeek-R1-Distill-Qwen, while the suffix *it* indicates the Instruct version of the model. OA denotes the overall average accuracy on the choice tasks.

| Model | OA | 1-1 | 1-2 | 1-3 | 1-4 | 1-6 | 1-7 | 2-1 | 2-2 | 2-3 | 2-4 | 3-1 | 3-2 | 3-3 | 3-4 | 3-5 | 4-3 |
|---|---|---|---|---|---|---|---|---|---|---|---|---|---|---|---|---|---|
| GPT-4o | 73.2 | 97.1 | 91.6 | 82.5 | 86.9 | 50.7 | 69.6 | 58.8 | 53.3 | 60.6 | 85.8 | 75.7 | 77.2 | 82.4 | 84.2 | 50.0 | 68.4 |
| GPT-4o-mini | 71.4 | 97.1 | 88.9 | 85.0 | 83.7 | 49.3 | 68.8 | 56.0 | 62.1 | 61.3 | 85.2 | 77.1 | 69.2 | 81.0 | 79.2 | 44.0 | 54.4 |
| DeepSeek-V3 | 72.9 | 95.9 | 90.3 | 90.3 | 82.5 | 50.7 | 67.1 | 65.6 | 55.9 | 66.9 | 82.1 | 77.1 | 79.2 | 79.6 | 83.1 | 44.6 | 67.3 |
| Qwen3 | 68.2 | 93.4 | 80.1 | 73.8 | 73.4 | 49.8 | 65.0 | 59.2 | 60.7 | 65.0 | 83.3 | 77.9 | 75.8 | 73.6 | 77.6 | 44.0 | 49.7 |
| Qwen2.5-72B-it | 71.6 | 95.4 | 88.1 | 87.4 | 85.3 | 48.4 | 67.5 | 60.8 | 54.4 | 63.8 | 85.2 | 76.4 | 78.5 | 78.0 | 82.5 | 43.3 | 61.4 |
| Qwen2.5-7B-it | 65.2 | 90.9 | 80.1 | 84.0 | 77.4 | 49.3 | 60.0 | 51.2 | 54.4 | 62.5 | 80.9 | 75.7 | 75.8 | 64.0 | 78.1 | 38.9 | 41.5 |
| Llama3.1-70B-it | 70.3 | 96.3 | 89.4 | 86.4 | 83.3 | 48.4 | 67.1 | 58.8 | 58.8 | 58.1 | 81.5 | 75.0 | 82.6 | 71.2 | 77.6 | 45.2 | 59.7 |
| Llama3.1-8B-it | 58.2 | 92.5 | 78.8 | 71.8 | 64.0 | 37.0 | 62.9 | 44.8 | 47.6 | 47.5 | 79.0 | 61.4 | 66.4 | 45.4 | 74.9 | 41.7 | 45.6 |
| Gemma-2-27B-it | 67.0 | 92.5 | 83.2 | 83.5 | 79.8 | 45.7 | 62.9 | 59.6 | 54.8 | 58.8 | 85.2 | 72.9 | 71.1 | 51.6 | 80.3 | – | 48.0 |
| Gemma-2-9B-it | 64.3 | 91.7 | 81.9 | 79.6 | 59.5 | 45.7 | 61.7 | 57.2 | 55.1 | 48.1 | 79.6 | 70.0 | 76.5 | 52.4 | 78.7 | – | 49.1 |
| Mistral-7B-it | 55.4 | 85.5 | 70.8 | 68.0 | 61.5 | 35.2 | 57.1 | 41.6 | 52.2 | 44.4 | 77.8 | 59.3 | 63.1 | 40.4 | 71.0 | 43.9 | 45.0 |
| MoZi-qwen | 61.8 | 94.6 | 79.6 | 82.5 | 73.0 | 41.6 | 63.8 | 52.8 | 52.9 | 42.5 | 77.2 | 77.1 | 71.1 | 48.2 | 75.4 | 43.9 | 40.9 |
| DISC-LawLLM | 50.2 | 70.5 | 60.6 | 71.8 | 56.8 | 39.3 | 55.0 | 43.6 | 55.2 | 22.5 | 71.6 | 57.1 | 53.0 | 28.0 | 69.4 | – | 23.4 |
| Hanfei | 40.3 | 60.6 | 46.0 | 53.9 | 37.7 | 30.6 | 42.1 | 33.2 | 47.4 | 40.6 | 44.4 | 43.6 | 37.6 | 27.8 | 53.0 | – | 18.7 |
| DeepSeek-R1 | 71.6 | 96.3 | 92.9 | 92.2 | 82.1 | 51.1 | 70.0 | 62.4 | 54.4 | 77.5 | 87.0 | 67.9 | 77.9 | 69.6 | 79.2 | 47.5 | 57.9 |
| DS-Qwen-7B | 56.0 | 76.4 | 58.9 | 58.3 | 50.4 | 48.0 | 52.5 | 43.6 | 49.3 | 53.8 | 82.1 | 57.9 | 59.1 | 63.8 | 63.8 | 43.6 | 38.0 |
| QwQ-32B | 70.7 | 95.9 | 88.5 | 85.4 | 75.8 | 47.0 | 68.8 | 64.8 | 57.7 | 74.4 | 84.6 | 75.7 | 83.9 | 69.0 | 76.5 | 47.1 | 59.7 |

Table 41: Results of English Patent IPC/CPC Classification tasks (1-5-1 and 1-5-2). The best-performing model in each task is in darker purple , and the second best is in lighter purple .

| Model | IPC Classification (1-5-1) | | | | CPC Classification (1-5-2) | | | |
|---|---|---|---|---|---|---|---|---|
| | Exact-Match | Section | Class | Subclass | Exact-Match | Section | Class | Subclass |
| GPT-4o | 2.0 | 86.2 | 72.0 | 49.0 | 3.3 | 82.7 | 69.7 | 62.0 |
| GPT-4o-mini | 1.0 | 84.5 | 67.5 | 47.0 | 0.5 | 79.0 | 64.5 | 52.7 |
| DeepSeek-V3 | 2.3 | 86.7 | 73.8 | 51.3 | 9.5 | 84.0 | 73.3 | 65.2 |
| Qwen3 | 2.2 | 84.0 | 68.2 | 48.3 | 0.5 | 62.7 | 48.3 | 38.7 |
| Qwen2.5-72B-it | 2.2 | 84.7 | 69.8 | 49.0 | 2.5 | 81.5 | 69.5 | 60.7 |
| Qwen2.5-7B-it | 1.5 | 84.0 | 67.5 | 46.8 | 0.2 | 65.5 | 44.8 | 34.8 |
| Llama3.1-70B-it | 2.2 | 82.7 | 66.3 | 45.3 | 1.0 | 79.5 | 64.3 | 52.7 |
| Llama3.1-8B-it | 1.3 | 77.7 | 60.0 | 37.3 | 0.0 | 63.8 | 45.0 | 30.7 |
| Gemma-2-27B-it | 1.3 | 73.3 | 58.3 | 37.7 | 0.2 | 70.5 | 56.7 | 44.3 |
| Gemma-2-9B-it | 0.5 | 80.2 | 59.7 | 39.2 | 0.2 | 56.2 | 39.0 | 26.7 |
| Mistral-7B-it | 0.2 | 82.3 | 57.3 | 35.5 | 0.0 | 39.0 | 21.5 | 10.3 |
| MoZi-qwen | 0.8 | 42.8 | 33.2 | 23.3 | 0.0 | 8.5 | 3.1 | 1.8 |
| DISC-LawLLM | 0.0 | 83.0 | 61.3 | 39.5 | 0.0 | 31.0 | 23.4 | 11.5 |
| Hanfei | 0.0 | 6.9 | 0.0 | 0.0 | 0.0 | 0.9 | 0.0 | 0.0 |
| DeepSeek-R1 | 3.2 | 88.0 | 74.0 | 52.0 | 8.5 | 82.5 | 71.2 | 63.2 |
| DS-Qwen-7B | 0.0 | 12.9 | 5.5 | 1.7 | 0.0 | 5.1 | 0.5 | 0.2 |
| QwQ-32B | 2.0 | 86.3 | 69.5 | 47.7 | 0.5 | 76.0 | 62.3 | 51.3 |

### K.3.2 FEW-SHOT RESULTS

The 1-shot results for the English portion of IPBench are shown in Table 43 and Table 44, the 2-shot results in Table 45 and Table 46 and the 3-shot results in Table 47 and Table 48.

Table 42: Results of English generation tasks (4-1 and 4-2). The best-performing model in each task is in **darker blue** , and the second best is in lighter blue . R-L refers to ROUGE-L, BS refers to BERTScore, LLMScore refers to GPT-4 judge score (1-10), Avg Tokens # denotes the average number of generated tokens, and Avg DC # denotes the average number of generated dependent claims.

| Model | Abstract Generation (4-1) | | | | | Dependent Claim Generation (4-2) | | | | | |
|---|---|---|---|---|---|---|---|---|---|---|---|
| | BLEU | R-L | BS | LLMScore (1-10) | Tokens # (129.0) | BLEU | R-L | BS | LLMScore (1-10) | Tokens # (417.4) | DC # (13.1) |
| GPT-4o | 17.7 | 27.3 | 87.7 | 8.07 | 264.2 | 25.2 | 28.0 | 87.4 | 6.97 | 637.4 | 6.2 |
| GPT-4o-mini | 28.8 | 28.6 | 88.4 | 7.59 | 211.3 | 25.5 | 28.0 | 86.5 | 6.66 | 458.4 | 1.1 |
| DeepSeek-V3 | 26.7 | 26.8 | 87.4 | 7.84 | 218.3 | **27.4** | 30.1 | **88.0** | **7.54** | 583.7 | 14.8 |
| Qwen2.5-72B-it | 28.3 | 30.4 | 88.3 | **8.17** | 272.5 | 10.3 | 13.9 | 87.8 | 6.01 | 6207.6 | 120.8 |
| Qwen2.5-7B-it | 34.2 | 34.6 | 89.3 | 8.07 | 227.9 | 19.2 | 23.1 | 87.8 | 5.71 | 3569.3 | 48.0 |
| Llama3.1-70B-it | **37.1** | 36.1 | **89.7** | 8.07 | 191.8 | 24.8 | 27.8 | 86.5 | 6.36 | 543.6 | 13.6 |
| Llama3.1-8B-it | 28.4 | 30.1 | 88.1 | 7.79 | 359.7 | 11.4 | 14.6 | 86.7 | 4.64 | 7643.6 | 145.4 |
| Gemma-2-27B-it | 24.7 | 24.4 | 87.6 | 7.54 | 171.2 | 23.8 | 26.1 | 86.0 | 6.49 | 486.1 | 3.4 |
| Gemma-2-9B-it | 25.5 | 24.9 | 87.4 | 7.76 | 190.7 | 23.8 | 25.6 | 86.0 | 5.95 | 446.7 | 7.0 |
| Mistral-7B-it | 28.6 | 28.6 | 88.4 | 7.75 | 243.9 | 10.9 | 12.8 | 87.3 | 4.72 | 8117.8 | 148.5 |
| MoZi-qwen | 31.1 | **48.6** | 89.1 | 7.56 | 296.8 | 24.8 | **40.3** | 87.7 | 5.34 | 1936.5 | 35.5 |
| DeepSeek-R1 | 18.7 | 27.6 | 85.7 | 7.55 | 613.6 | 23.8 | 31.8 | 61.1 | 7.19 | 1231.6 | 22.4 |
| DS-Qwen-7B | 7.0 | 9.7 | 76.8 | 7.36 | 686.9 | 17.9 | 32.2 | 56.2 | 4.62 | 2315.7 | 18.4 |
| QwQ-32B | 22.2 | 30.4 | 85.6 | **8.17** | 849.5 | 19.8 | 29.1 | 63.1 | 7.14 | 4635.4 | 45.7 |

Table 43: English questions results of IPBench with 1-shot setting. The best-performing model in each task is in **darker red** , and the second best is in lighter red .

| Model | OA | 1-1 | 1-2 | 1-3 | 1-4 | 1-6 | 1-7 | 2-1 | 2-2 | 2-3 | 2-4 | 3-1 | 3-2 | 3-3 | 3-4 | 4-3 |
|---|---|---|---|---|---|---|---|---|---|---|---|---|---|---|---|---|
| GPT-4o-mini | **75.4** | **96.3** | **86.7** | **87.4** | **82.9** | **52.1** | 67.5 | **59.6** | **57.4** | **69.4** | **85.8** | 63.1 | **79.9** | **83.0** | **83.1** | **58.5** |
| Qwen2.5-7B-it | 65.1 | 93.8 | 77.9 | 82.0 | 71.8 | 45.2 | 59.6 | 54.0 | 54.0 | 54.4 | 79.6 | 58.1 | 75.2 | 51.2 | 76.0 | 46.8 |
| Llama3.1-8B-it | 59.9 | 93.0 | 76.5 | 76.7 | 71.0 | 37.9 | **68.3** | 48.4 | 47.8 | 48.8 | 76.5 | 56.3 | 75.2 | 35.6 | 73.2 | 40.9 |
| Gemma-2-9B-it | 64.6 | 91.7 | 74.3 | 76.2 | **77.0** | 41.6 | 60.8 | 54.8 | 52.9 | 54.4 | 77.2 | **64.4** | 77.2 | 51.0 | 77.0 | 51.5 |
| Mistral-7B-it | 56.2 | 87.6 | 68.1 | 70.4 | 60.3 | 37.9 | 53.8 | 50.4 | 50.0 | 30.6 | 77.8 | 53.8 | 62.4 | 37.4 | 69.4 | 48.5 |
| MoZi-qwen | 57.6 | 95.4 | 74.4 | 79.6 | 71.8 | 42.9 | 66.3 | 49.6 | 54.8 | 39.4 | 79.0 | 63.1 | 57.1 | 25.0 | 61.2 | 31.6 |
| DISC-LawLLM | 54.1 | 78.4 | 64.6 | 71.4 | 61.9 | 35.2 | 61.7 | 41.2 | 52.9 | 28.1 | 56.8 | 55.6 | 40.9 | – | 57.9 | 29.8 |
| Hanfei | 32.9 | 40.3 | 35.8 | 36.9 | 28.6 | 35.2 | 36.7 | 29.2 | 27.2 | 35.6 | 30.3 | 38.1 | 26.2 | – | 27.9 | 35.1 |

Table 44: Results of English Patent IPC/CPC Classification tasks (1-5-1 and 1-5-2) with 1-shot setting. The best-performing model in each task is in **darker purple** , and the second best is in lighter purple .

| Model | IPC Classification (1-5-1) | | | | CPC Classification (1-5-2) | | | |
|---|---|---|---|---|---|---|---|---|
| | Exact-Match | Section | Class | Subclass | Exact-Match | Section | Class | Subclass |
| GPT-4o-mini | 1.0 | 86.2 | 67.0 | 47.3 | 0.5 | 74.3 | 59.1 | 49.1 |
| DeepSeek-V3 | **2.0** | **89.7** | **75.7** | **53.2** | **7.3** | **86.2** | **74.3** | **65.0** |
| Qwen2.5-7B-it | 0.8 | 86.2 | 64.0 | 44.5 | 0.3 | 67.5 | 48.8 | 37.2 |
| Llama3.1-8B-it | 0.6 | 80.0 | 61.3 | 40.8 | 0.0 | 45.2 | 35.2 | 22.2 |
| MoZi-qwen | 0.7 | 78.0 | 57.7 | 37.8 | 0.0 | 16.5 | 7.8 | 4.3 |

### K.3.3 CHAIN-OF-THOUGHT RESULTS

The chain-of-thought results for the English portion of IPBench are presented in Table 49 and Table 50.

Table 45: English questions results of IPBench with 2-shot setting. The best-performing model in each task is in darker red , and the second best is in lighter red .

| Model | OA | 1-1 | 1-2 | 1-3 | 1-4 | 1-6 | 1-7 | 2-1 | 2-2 | 2-3 | 2-4 | 3-1 | 3-2 | 3-3 | 3-4 | 4-3 |
|---|---|---|---|---|---|---|---|---|---|---|---|---|---|---|---|---|
| GPT-4o-mini | 75.2 | 96.7 | 88.5 | 88.4 | 84.1 | 52.1 | 65.8 | 59.2 | 55.5 | 68.1 | 87.7 | 77.1 | 84.6 | 80.4 | 80.3 | 60.8 |
| Qwen2.5-7B-it | 68.3 | 94.6 | 79.7 | 83.5 | 77.8 | 45.2 | 60.0 | 56.0 | 53.3 | 60.0 | 79.0 | 82.9 | 79.2 | 64.6 | 79.8 | 42.7 |
| Llama3.1-8B-it | 60.1 | 92.5 | 72.1 | 75.2 | 69.4 | 43.4 | 65.0 | 49.2 | 50.4 | 54.4 | 76.5 | 67.9 | 68.5 | 37.6 | 73.8 | 39.8 |
| Gemma-2-9B-it | 66.0 | 91.7 | 76.5 | 78.2 | 73.8 | 42.5 | 62.1 | 56.0 | 51.8 | 57.5 | 80.2 | 72.9 | 81.9 | 56.2 | 78.1 | 52.6 |
| Mistral-7B-it | 58.0 | 87.1 | 71.7 | 73.3 | 59.9 | 37.4 | 55.4 | 49.6 | 51.1 | 33.8 | 77.2 | 68.8 | 61.1 | 45.6 | 68.9 | 48.0 |
| MoZi-qwen | 62.7 | 96.3 | 80.1 | 80.1 | 73.4 | 46.6 | 62.9 | 49.2 | 52.9 | 42.5 | 81.5 | 76.4 | 68.5 | 46.6 | 77.6 | 26.3 |
| DISC-LawLLM | 55.8 | 82.2 | 68.1 | 71.8 | 62.3 | 40.2 | 58.3 | 41.2 | 51.8 | 26.3 | 64.2 | 61.4 | 50.3 | – | 67.8 | 24.0 |
| Hanfei | 32.8 | 41.9 | 31.0 | 29.6 | 37.3 | 35.6 | 31.3 | 25.2 | 38.2 | 42.5 | 27.2 | 40.0 | 32.9 | – | 31.7 | 12.9 |

Table 46: Results of English Patent IPC/CPC Classification tasks (1-5-1 and 1-5-2) with 2-shot setting. The best-performing model in each task is in darker purple , and the second best is in lighter purple .

| Model | IPC Classification (1-5-1) | | | | CPC Classification (1-5-2) | | | |
|---|---|---|---|---|---|---|---|---|
| | Exact-Match | Section | Class | Subclass | Exact-Match | Section | Class | Subclass |
| GPT-4o-mini | 2.17 | 88.2 | 69.8 | 48.8 | 0.2 | 76.3 | 61.6 | 51.6 |
| DeepSeek-V3 | 2.67 | 90.2 | 76.2 | 53.2 | 7.2 | 86.5 | 73.3 | 65.7 |
| Qwen2.5-7B-it | 1.50 | 85.0 | 64.5 | 45.0 | 0.3 | 68.5 | 51.2 | 38.8 |
| Llama3.1-8B-it | 1.30 | 81.5 | 59.0 | 38.2 | 0.0 | 63.3 | 45.7 | 26.7 |
| MoZi-qwen | 0.83 | 80.0 | 58.0 | 36.7 | 0.2 | 32.3 | 17.3 | 9.3 |

Table 47: English questions results of IPBench with 3-shot setting. The best-performing model in each task is in darker red , and the second best is in lighter red .

| Model | OA | 1-1 | 1-2 | 1-3 | 1-4 | 1-6 | 1-7 | 2-1 | 2-2 | 2-3 | 2-4 | 3-1 | 3-2 | 3-3 | 3-4 | 4-3 |
|---|---|---|---|---|---|---|---|---|---|---|---|---|---|---|---|---|
| GPT-4o-mini | 74.7 | 97.5 | 87.6 | 88.4 | 82.9 | 53.0 | 70.0 | 57.6 | 54.0 | 67.5 | 87.0 | 79.3 | 82.6 | 79.4 | 80.9 | 66.7 |
| Qwen2.5-7B-it | 70.6 | 93.4 | 81.9 | 79.6 | 75.0 | 48.0 | 66.3 | 56.0 | 52.6 | 58.8 | 82.7 | 85.0 | 79.9 | 67.4 | 80.9 | 43.9 |
| Llama3.1-8B-it | 59.4 | 94.2 | 74.4 | 76.3 | 71.0 | 40.2 | 63.3 | 43.6 | 53.7 | 53.8 | 73.5 | 70.0 | 70.5 | 35.6 | 73.8 | 40.4 |
| Gemma-2-9B-it | 67.4 | 91.3 | 77.4 | 77.2 | 72.2 | 44.7 | 61.4 | 54.8 | 51.5 | 55.0 | 82.1 | 73.6 | 77.9 | 57.4 | 76.0 | 57.9 |
| Mistral-7B-it | 56.5 | 89.6 | 69.0 | 71.4 | 58.3 | 37.9 | 55.4 | 49.2 | 50.7 | 31.3 | 74.1 | 69.3 | 62.4 | 36.8 | 70.5 | 53.2 |
| MoZi-qwen | 65.3 | 96.7 | 79.2 | 81.1 | 76.2 | 47.0 | 65.0 | 47.6 | 54.4 | 43.1 | 82.1 | 78.6 | 69.1 | 29.1 | 76.0 | 27.5 |
| DISC-LawLLM | 56.6 | 85.9 | 65.0 | 69.9 | 62.7 | 42.5 | 57.9 | 42.0 | 55.9 | 23.1 | 69.8 | 62.1 | 49.0 | – | 67.2 | 28.1 |
| Hanfei | 28.9 | 36.5 | 36.3 | 22.3 | 31.0 | 27.9 | 27.5 | 20.0 | 31.6 | 33.8 | 24.1 | 38.6 | 26.9 | – | 25.1 | 24.0 |

Table 48: Results of English Patent IPC/CPC Classification tasks (1-5-1 and 1-5-2) with 3-shot setting. The best-performing model in each task is in darker purple , and the second best is in lighter purple .

| Model | IPC Classification (1-5-1) | | | | CPC Classification (1-5-2) | | | |
|---|---|---|---|---|---|---|---|---|
| | Exact-Match | Section | Class | Subclass | Exact-Match | Section | Class | Subclass |
| GPT-4o-mini | 1.8 | 87.8 | 69.2 | 46.8 | 0.3 | 80.1 | 65.3 | 54.1 |
| DeepSeek-V3 | 3.5 | 90.0 | 75.7 | 52.8 | 7.7 | 85.7 | 73.3 | 64.7 |
| Qwen2.5-7B-it | 1.2 | 85.7 | 64.8 | 45.7 | 0.5 | 68.3 | 50.8 | 38.8 |
| Llama3.1-8B-it | 1.8 | 82.7 | 62.0 | 41.2 | 0.0 | 64.8 | 45.8 | 29.8 |
| MoZi-qwen | 1.3 | 80.3 | 59.3 | 36.7 | 0.0 | 24.2 | 12.8 | 7.7 |

Table 49: English questions results of IPBench with chain-of-thought setting. The best-performing model in each task is in darker red , and the second best is in lighter red .

| Model | OA | 1-1 | 1-2 | 1-3 | 1-4 | 1-6 | 1-7 | 2-1 | 2-2 | 2-3 | 2-4 | 3-1 | 3-2 | 3-3 | 3-4 | 3-5 | 4-3 |
|---|---|---|---|---|---|---|---|---|---|---|---|---|---|---|---|---|---|
| GPT-4o-mini | 72.2 | 96.7 | 88.1 | 88.4 | 82.9 | 51.6 | 67.9 | 57.6 | 57.4 | 64.4 | 87.0 | 75.0 | 76.5 | 81.2 | 83.1 | 44.9 | 58.5 |
| Qwen2.5-7B-it | 64.4 | 84.2 | 78.8 | 78.6 | 73.0 | 48.9 | 60.0 | 50.0 | 55.9 | 61.3 | 84.0 | 75.0 | 70.5 | 63.2 | 78.8 | 44.3 | 43.3 |
| Llama3.1-8B-it | 60.4 | 84.9 | 76.6 | 74.3 | 73.8 | 46.6 | 64.2 | 46.4 | 55.5 | 52.5 | 79.0 | 66.4 | 67.8 | 47.4 | 74.9 | 43.6 | 40.9 |
| Gemma-2-9B-it | 60.3 | 88.8 | 75.2 | 76.2 | 69.4 | 45.7 | 60.0 | 55.2 | 56.3 | 46.3 | 80.9 | 74.3 | 75.2 | 62.0 | 80.9 | – | 54.4 |
| Mistral-7B-it | 54.1 | 85.5 | 72.6 | 71.8 | 64.7 | 35.6 | 59.2 | 39.2 | 50.4 | 43.1 | 75.3 | 56.4 | 63.1 | 37.2 | 72.1 | 36.3 | 35.7 |
| MoZi-qwen | 60.5 | 94.6 | 79.2 | 82.5 | 72.6 | 42.5 | 63.3 | 51.2 | 50.4 | 45.6 | 77.8 | 73.6 | 70.5 | 40.4 | 77.1 | 43.9 | 41.5 |
| DISC-LawLLM | 19.5 | 45.6 | 46.5 | 28.2 | 15.1 | 15.1 | 31.7 | 13.6 | 14.7 | 26.3 | 18.5 | 7.1 | 12.8 | 13.6 | 4.6 | – | 9.4 |
| Hanfei | 30.5 | 39.8 | 32.7 | 28.6 | 30.6 | 24.2 | 27.5 | 32.8 | 32.7 | 20.6 | 26.5 | 33.6 | 36.2 | 26.2 | 40.4 | – | 29.2 |

Table 50: Results of English Patent IPC/CPC Classification tasks (1-5-1 and 1-5-2) with chain-of-thought setting. The best-performing model in each task is in darker purple , and the second best is in lighter purple .

| Model | IPC Classification (1-5-1) | | | | CPC Classification (1-5-2) | | | |
|---|---|---|---|---|---|---|---|---|
| | Exact-Match | Section | Class | Subclass | Exact-Match | Section | Class | Subclass |
| GPT-4o-mini | 0.0 | 84.8 | 67.3 | 47.8 | 0.0 | 76.8 | 63.0 | 52.5 |
| DeepSeek-V3 | 0.3 | 86.8 | 73.0 | 51.5 | 1.0 | 83.3 | 70.7 | 63.0 |
| Qwen2.5-7B-it | 1.8 | 88.0 | 69.5 | 47.0 | 0.5 | 60.2 | 46.0 | 35.7 |
| Llama3.1-8B-it | 1.5 | 76.2 | 58.5 | 37.5 | 0.2 | 64.0 | 44.8 | 29.5 |
| MoZi-qwen | 0.5 | 38.8 | 29.8 | 21.8 | 0.0 | 7.7 | 2.8 | 1.8 |

### K.4  LLM-AS-A-JUDGE RESULTS

We provide detailed results of the LLM-as-a-judge evaluation for the overall, Chinese, and English parts. The evaluation includes four dimensions and an overall score, as shown in Table 51, Table 52, and Table 53. The definitions of these metrics are provided in Section H.3, with all scores ranging from 1 to 10.

Table 51: Multi-dimension results of generation tasks (4-1 and 4-2) in LLM-as-a-judge. The best-performing model in each task is in darker blue , and the second best is in lighter blue . Accuracy (Acc.), Relevance (Rel.), Completeness (Comp.), Consistency (Cons.), L-S and LLMScore are generation quality metrics rated by an LLM-as-a-judge.

| Model | Abstract Generation (4-1) | | | | | | Dependent Claim Generation (4-2) | | | | | |
|---|---|---|---|---|---|---|---|---|---|---|---|---|
| | Acc. | Rel. | Comp. | Cons. | L-S | LLMScore | Acc. | Rel. | Comp. | Cons. | L-S | LLMScore |
| GPT-4o | 8.45 | 8.24 | 8.68 | 9.27 | 7.58 | 8.42 | 7.45 | 6.28 | 6.22 | 6.58 | 7.17 | 6.63 |
| GPT-4o-mini | 7.99 | 8.02 | 8.13 | 8.94 | 7.47 | 8.05 | 7.17 | 5.92 | 6.06 | 6.30 | 7.02 | 6.37 |
| DeepSeek-V3 | 8.26 | 8.45 | 8.53 | 9.15 | 7.73 | 8.38 | 7.93 | 7.30 | 7.13 | 7.38 | 7.92 | 7.45 |
| Qwen2.5-72B-it | 8.40 | 8.18 | 8.70 | 9.36 | 7.37 | 8.33 | 7.13 | 5.77 | 6.00 | 6.35 | 6.72 | 6.30 |
| Qwen2.5-7B-it | 8.17 | 8.14 | 8.19 | 9.08 | 7.61 | 8.18 | 6.59 | 5.47 | 5.09 | 5.68 | 5.96 | 5.67 |
| Llama3.1-70B-it | 7.98 | 8.03 | 7.94 | 8.96 | 7.31 | 7.98 | 6.57 | 5.38 | 5.16 | 5.69 | 6.21 | 5.67 |
| Llama3.1-8B-it | 7.52 | 7.41 | 7.71 | 8.57 | 6.54 | 7.47 | 4.70 | 3.95 | 3.18 | 3.91 | 4.15 | 3.86 |
| Gemma-2-27B-it | 7.63 | 7.78 | 7.46 | 8.40 | 7.32 | 7.64 | 6.51 | 5.56 | 5.71 | 5.84 | 6.54 | 5.98 |
| Gemma-2-9B-it | 7.89 | 8.03 | 7.82 | 8.76 | 7.43 | 7.91 | 6.21 | 5.23 | 5.20 | 5.51 | 6.12 | 5.55 |
| Mistral-7B-it | 7.47 | 7.38 | 7.86 | 8.62 | 6.40 | 7.49 | 4.19 | 3.30 | 3.07 | 3.38 | 3.71 | 3.42 |
| MoZi-qwen | 7.71 | 7.88 | 7.78 | 8.76 | 7.02 | 7.73 | 5.82 | 4.70 | 4.00 | 4.83 | 5.17 | 4.81 |
| DeepSeek-R1 | 7.70 | 7.75 | 7.88 | 8.39 | 7.21 | 7.72 | 7.73 | 6.76 | 7.00 | 7.16 | 7.69 | 7.18 |
| DS-Qwen-7B | 7.58 | 7.50 | 7.78 | 8.43 | 6.90 | 7.58 | 4.67 | 4.02 | 3.97 | 4.01 | 4.60 | 4.16 |
| QwQ-32B | 8.48 | 8.39 | 8.80 | 9.27 | 7.74 | 8.51 | 7.63 | 6.61 | 6.97 | 7.13 | 7.61 | 7.10 |

Table 52: Multi-dimension results of Chinese generation tasks (4-1 and 4-2) in LLM-as-a-judge. The best-performing model in each task is in **darker blue**, and the second best is in lighter blue.

| Model | Abstract Generation (4-1) | | | | | | Dependent Claim Generation (4-2) | | | | | |
|---|---|---|---|---|---|---|---|---|---|---|---|---|
| | Acc. | Rel. | Comp. | Cons. | L-S | LLMScore | Acc. | Rel. | Comp. | Cons. | L-S | LLMScore |
| GPT-4o | 8.66 | 8.77 | 8.96 | 9.50 | 8.00 | 8.77 | 7.06 | 5.85 | 5.91 | 6.27 | 6.94 | 6.30 |
| GPT-4o-mini | 8.39 | 8.59 | 8.46 | 9.24 | 8.00 | 8.51 | 6.93 | 5.59 | 5.69 | 6.10 | 6.85 | 6.09 |
| DeepSeek-V3 | 8.80 | 9.05 | 9.08 | 9.54 | 8.27 | 8.92 | 7.90 | 6.98 | 7.09 | 7.40 | 7.93 | 7.36 |
| Qwen2.5-72B-it | 8.53 | 8.55 | 8.86 | 9.47 | 7.56 | 8.50 | 7.33 | 6.14 | 6.36 | 6.61 | 7.12 | 6.60 |
| Qwen2.5-7B-it | 8.21 | 8.50 | 8.09 | 9.05 | 8.03 | 8.29 | 6.56 | 5.45 | 5.11 | 5.58 | 5.89 | 5.64 |
| Llama3.1-70B-it | 7.85 | 8.22 | 7.76 | 8.88 | 7.26 | 7.89 | 6.04 | 4.80 | 4.37 | 5.00 | 5.45 | 4.99 |
| Llama3.1-8B-it | 7.12 | 7.30 | 7.47 | 8.31 | 6.11 | 7.16 | 3.55 | 3.48 | 2.47 | 3.03 | 3.49 | 3.09 |
| Gemma-2-27B-it | 7.65 | 7.98 | 7.52 | 8.45 | 7.49 | 7.74 | 5.89 | 5.14 | 5.33 | 5.19 | 5.92 | 5.46 |
| Gemma-2-9B-it | 7.94 | 8.30 | 7.98 | 8.86 | 7.52 | 8.07 | 5.82 | 4.87 | 4.88 | 5.14 | 5.68 | 5.15 |
| Mistral-7B-it | 7.20 | 7.35 | 7.73 | 8.50 | 6.03 | 7.24 | 2.50 | 2.22 | 1.96 | 2.05 | 2.36 | 2.13 |
| MoZi-qwen | 7.79 | 8.40 | 7.60 | 8.75 | 7.47 | 7.91 | 5.33 | 4.23 | 3.42 | 4.34 | 4.52 | 4.28 |
| DeepSeek-R1 | 7.76 | 8.01 | 8.05 | 8.30 | 7.42 | 7.89 | 7.74 | 6.74 | 6.98 | 7.16 | 7.69 | 7.17 |
| DS-Qwen-7B | 7.68 | 7.84 | 8.08 | 8.10 | 6.97 | 7.80 | 4.26 | 3.71 | 3.30 | 4.47 | 4.00 | 3.69 |
| QwQ-32B | 8.71 | 8.82 | 9.10 | 9.09 | 8.09 | 8.84 | 7.70 | 6.55 | 6.92 | 7.13 | 7.58 | 7.05 |

Table 53: Multi-dimension results of English generation tasks (4-1 and 4-2) in LLM-as-a-judge. The best-performing model in each task is in **darker blue**, and the second best is in lighter blue.

| Model | Abstract Generation (4-1) | | | | | | Dependent Claim Generation (4-2) | | | | | |
|---|---|---|---|---|---|---|---|---|---|---|---|---|
| | Acc. | Rel. | Comp. | Cons. | L-S | LLMScore | Acc. | Rel. | Comp. | Cons. | L-S | LLMScore |
| GPT-4o | 8.24 | 7.70 | 8.40 | 9.04 | 7.16 | 8.07 | 7.85 | 6.71 | 6.53 | 6.88 | 7.40 | 6.97 |
| GPT-4o-mini | 7.59 | 7.45 | 7.81 | 8.64 | 6.94 | 7.59 | 7.42 | 6.26 | 6.43 | 6.49 | 7.20 | 6.66 |
| DeepSeek-V3 | 7.72 | 7.86 | 7.99 | 8.76 | 7.20 | 7.84 | 7.96 | 7.63 | 7.17 | 7.36 | 7.92 | 7.54 |
| Qwen2.5-72B-it | 8.28 | 7.82 | 8.54 | 9.25 | 7.19 | 8.17 | 6.93 | 5.40 | 5.64 | 6.09 | 6.33 | 6.01 |
| Qwen2.5-7B-it | 8.12 | 7.78 | 8.30 | 9.10 | 7.20 | 8.07 | 6.63 | 5.48 | 5.07 | 5.78 | 6.03 | 5.71 |
| Llama3.1-70B-it | 8.12 | 7.84 | 8.12 | 9.03 | 7.36 | 8.07 | 7.10 | 5.96 | 5.95 | 6.38 | 6.97 | 6.36 |
| Llama3.1-8B-it | 7.93 | 7.51 | 7.96 | 8.82 | 6.98 | 7.79 | 5.84 | 4.43 | 3.89 | 4.79 | 4.80 | 4.64 |
| Gemma-2-27B-it | 7.61 | 7.58 | 7.40 | 8.35 | 7.15 | 7.54 | 7.13 | 5.97 | 6.10 | 6.48 | 7.17 | 6.49 |
| Gemma-2-9B-it | 7.85 | 7.75 | 7.67 | 8.65 | 7.34 | 7.76 | 6.60 | 5.59 | 5.53 | 5.88 | 6.56 | 5.95 |
| Mistral-7B-it | 7.75 | 7.42 | 7.99 | 8.73 | 6.76 | 7.75 | 5.89 | 4.39 | 4.18 | 4.71 | 5.07 | 4.72 |
| MoZi-qwen | 7.63 | 7.37 | 7.97 | 8.77 | 6.57 | 7.56 | 6.32 | 5.18 | 4.59 | 5.32 | 5.83 | 5.34 |
| DeepSeek-R1 | 7.63 | 7.48 | 7.70 | 8.47 | 6.99 | 7.55 | 7.71 | 6.77 | 7.02 | 7.16 | 7.68 | 7.19 |
| DS-Qwen-7B | 7.48 | 7.15 | 7.48 | 8.76 | 6.82 | 7.36 | 5.07 | 4.32 | 4.64 | 3.55 | 5.19 | 4.62 |
| QwQ-32B | 8.25 | 7.96 | 8.50 | 9.44 | 7.39 | 8.17 | 7.55 | 6.66 | 7.01 | 7.13 | 7.64 | 7.14 |

## L   MORE DETAILS ABOUT ERROR ANALYSIS

**Definition of Different Error Type.**   We classify the error into 7 types: Consistency error, Hallucination error, Reasoning error, Refusing error, Priority error, Mathematical error and Obsolescence error. The detailed definitions of each error type are as follows:

- **Consistency error**: The content in the model's response is inherently flawed or internally inconsistent, such as when the intermediate reasoning steps contradict the model's final answer.

- **Hallucination error**: The large language model's responses sometimes introduce fabricated legal information or include statements that sound plausible but are factually incorrectparticularly in Tasks 14, which require familiarity with typical legal cases.

- **Reasoning error**: This type refers to flaws in the logical process used by the model to arrive at its answer. These errors may include invalid deductions, misinterpretation of conditions, or incorrect application of domain-specific rules. In many cases, the model's intermediate reasoning steps fail to logically support its final conclusion, even if the answer appears superficially plausible. Such issues are particularly critical in the second-level tasks of IPBench, which demand accurate multi-step and conditional reasoning within legal and technical contexts.

- **Refusing error**: This error typically occurs in Tasks 14, which require the model to recall specific factual or legal cases. In these instances, some models respond by asking the user for additional information or by explicitly refusing to provide an answer. While such refusals may be more cautious or aligned with reliability principles, they still indicate a limitation in the model's ability to engage with the task as expected.

- **Priority error**: Priority Error refers to the model's failure to identify and prioritize the most critical factor(s) when multiple elements jointly influence the outcome. Instead of focusing on the decisive issue, the model may weigh secondary or irrelevant aspects equally, leading to incorrect or misleading conclusions.

- **Mathematical error**: This error type refers to issues related to a lack of precision in complex calculations, often resulting in incorrect outcomes. These errors can arise from miscalculations, rounding mistakes, or failure to properly apply mathematical operations, leading to significant discrepancies in the final result. This is particularly evident in Tasks 23, Compensation Calculation, where both IP law knowledge and an understanding of the case background are necessary to perform accurate calculations.

- **Obsolescence error**: Obsolescence Error refers to the model's failure to account for differences between current and outdated versions of legal documents or frameworks. This error occurs when the generated answer overlooks changes in the law, leading to outdated or inaccurate information. This is especially relevant in Tasks 13, Legal Evolution, where the model must retain knowledge of both current and past laws and understand the differences between them. However, some models do not update their memory, resulting in the use of obsolete information.

The most common error type is reasoning error, accounting for 33%. This is consistent with the performance decrease observed in models using the Chain-of-Thought setting. This highlights the importance of developing an IP-oriented model that balances both System 1 and System 2 capabilities.

**Case Study for Each Error Type.**   We provide two examples, one in Chinese and one in English, for each error type, as shown from Figure 6 to Figure 12. More extensive case studies for each task can be found in Appendix N.

## M   DATA EXAMPLES

We provide extensive data examples for each task in this section, as shown from Figure 13 to Figure 33. These examples include both English and Chinese datapoints, serving as representative samples for each corresponding task and helping to better illustrate the task definitions.

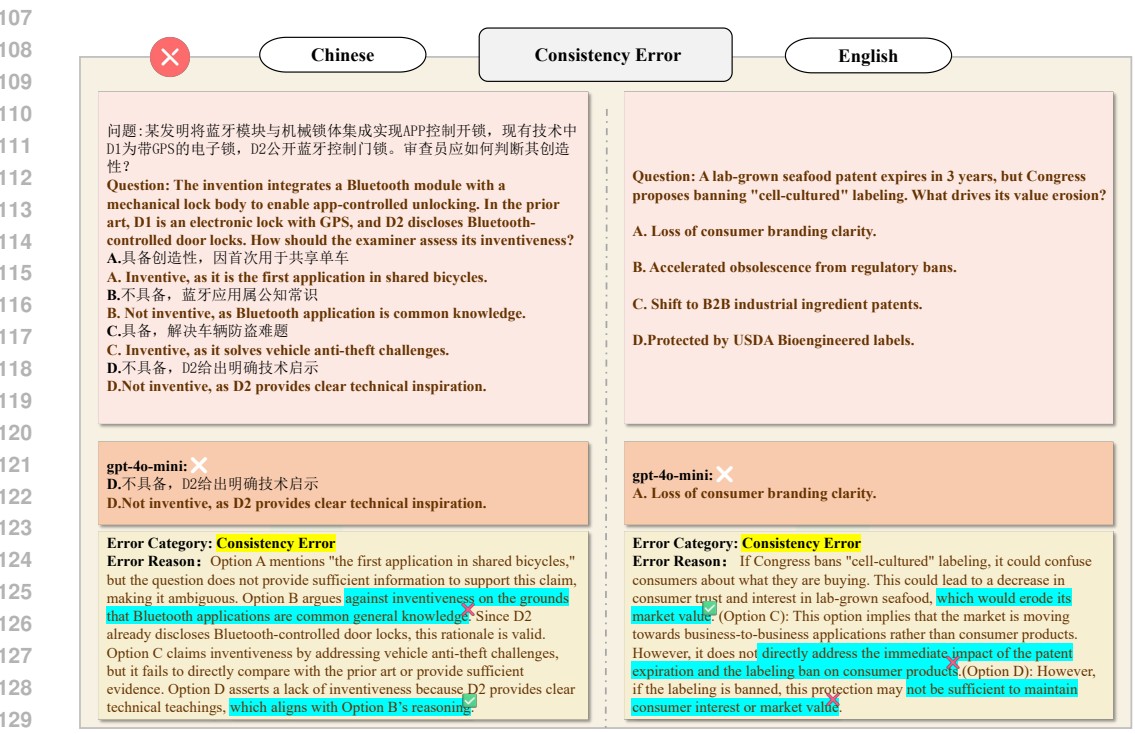

Figure 6: Consistency error case study.

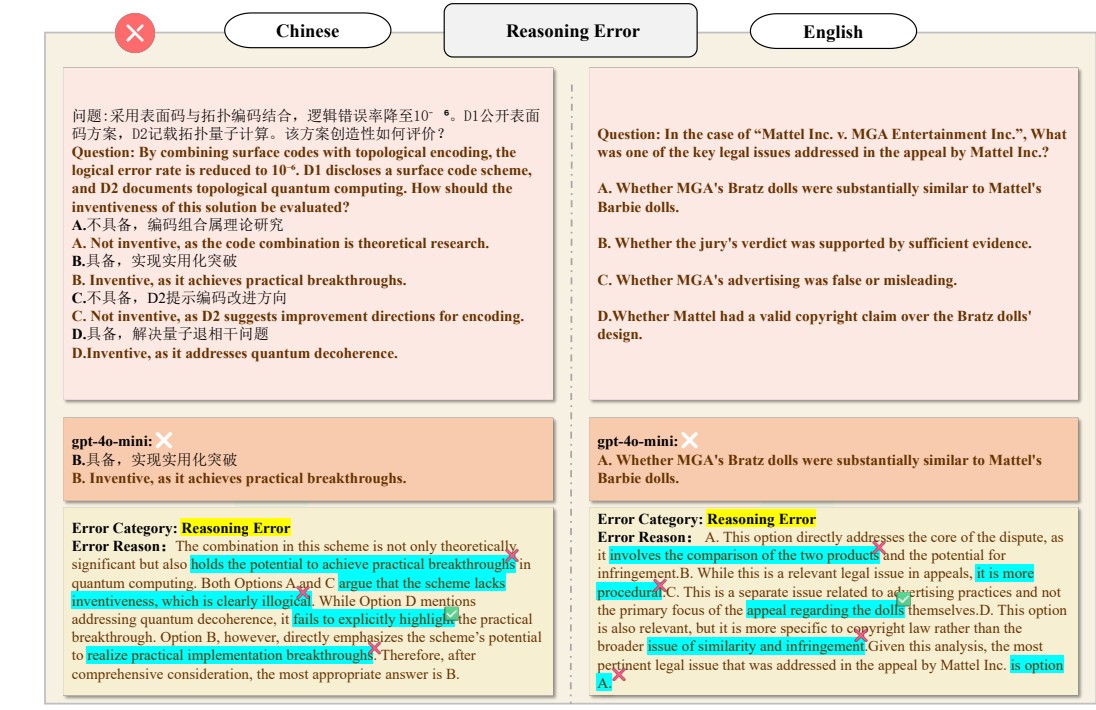

Figure 7: Reasoninig error case study.

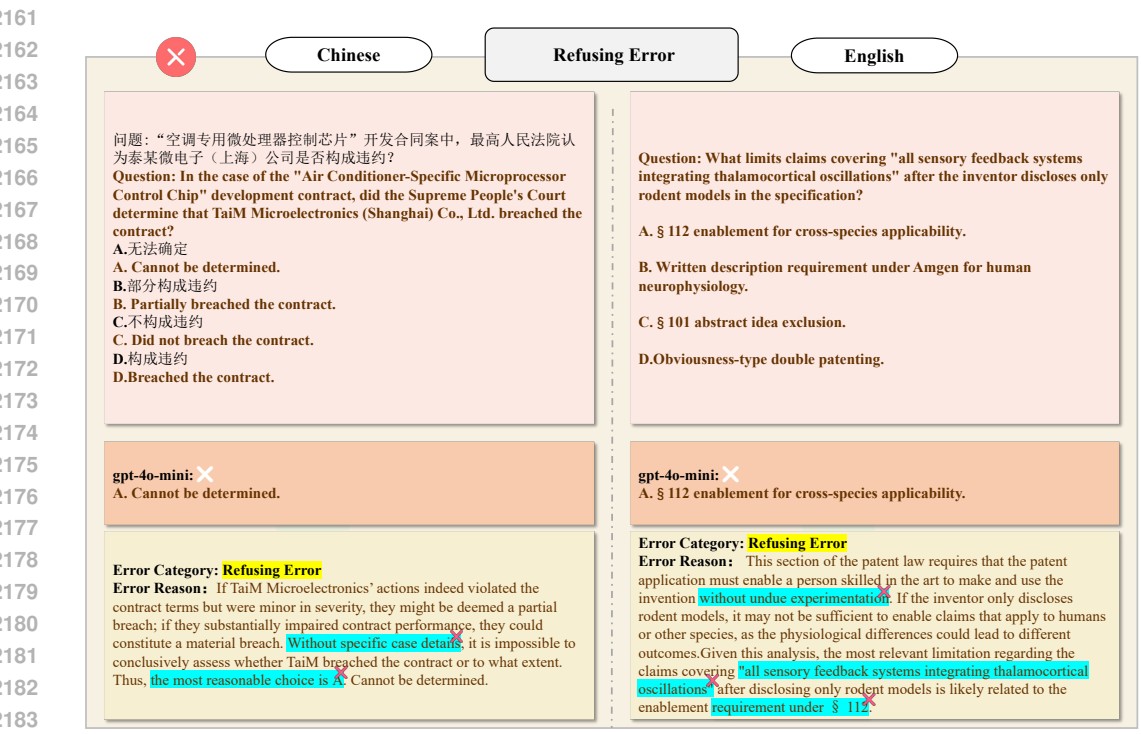

Figure 8: Refusing error case study.

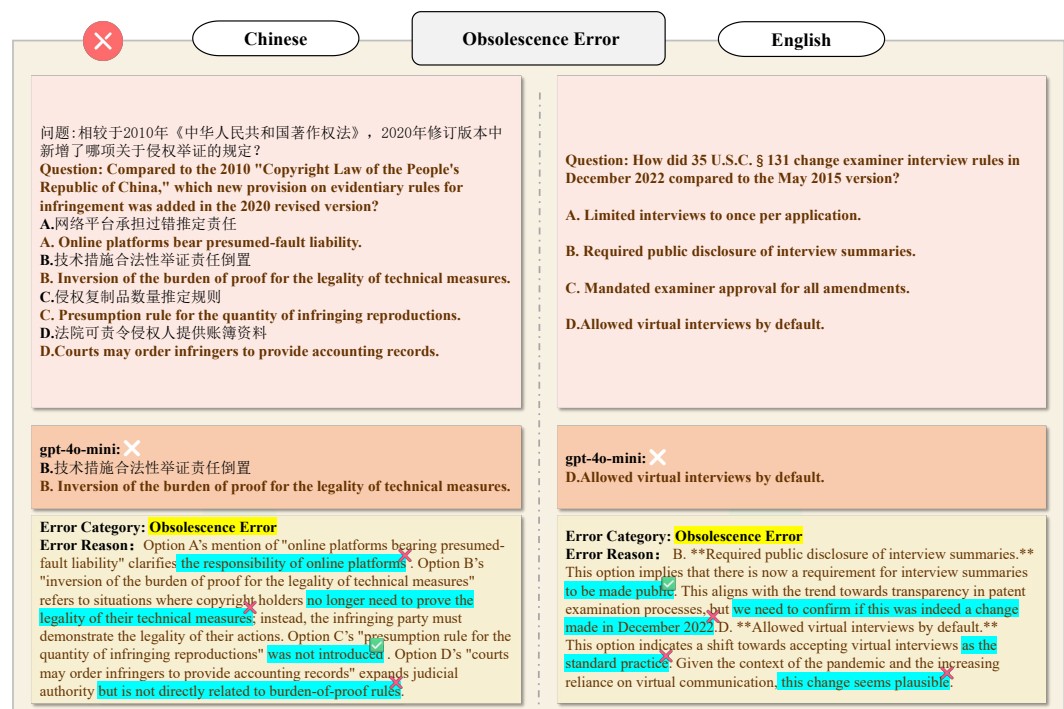

Figure 9: Obsolescence error case study.

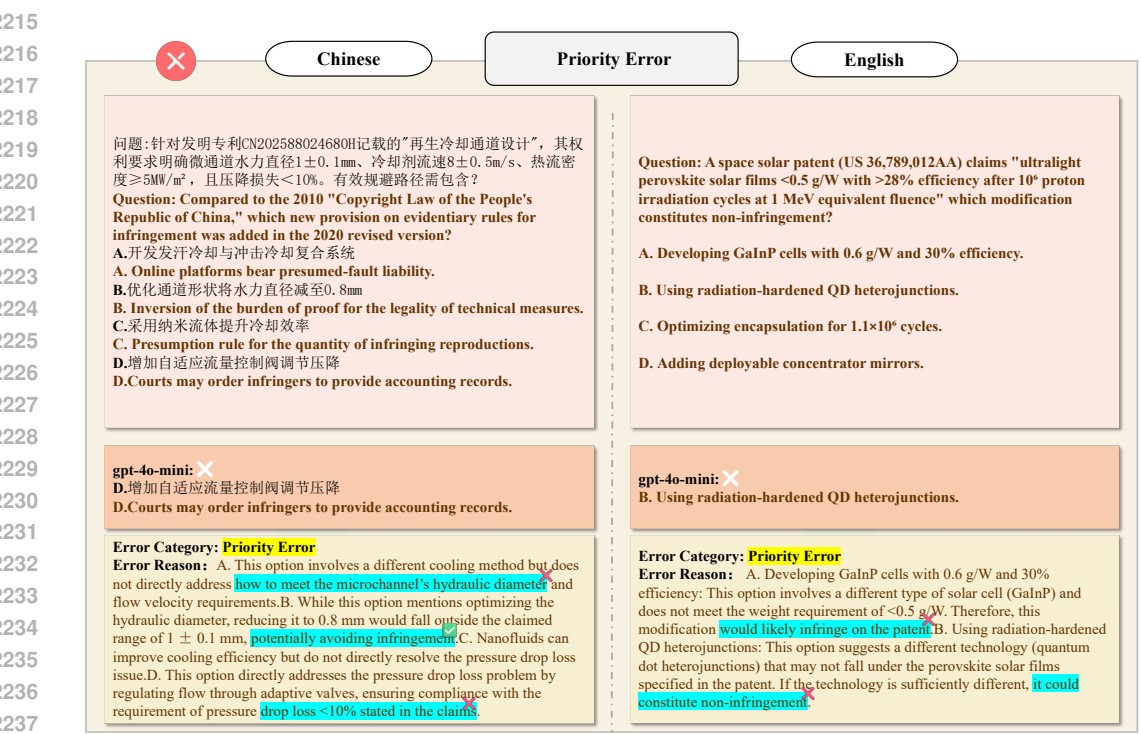

Figure 10: Priority error case study.

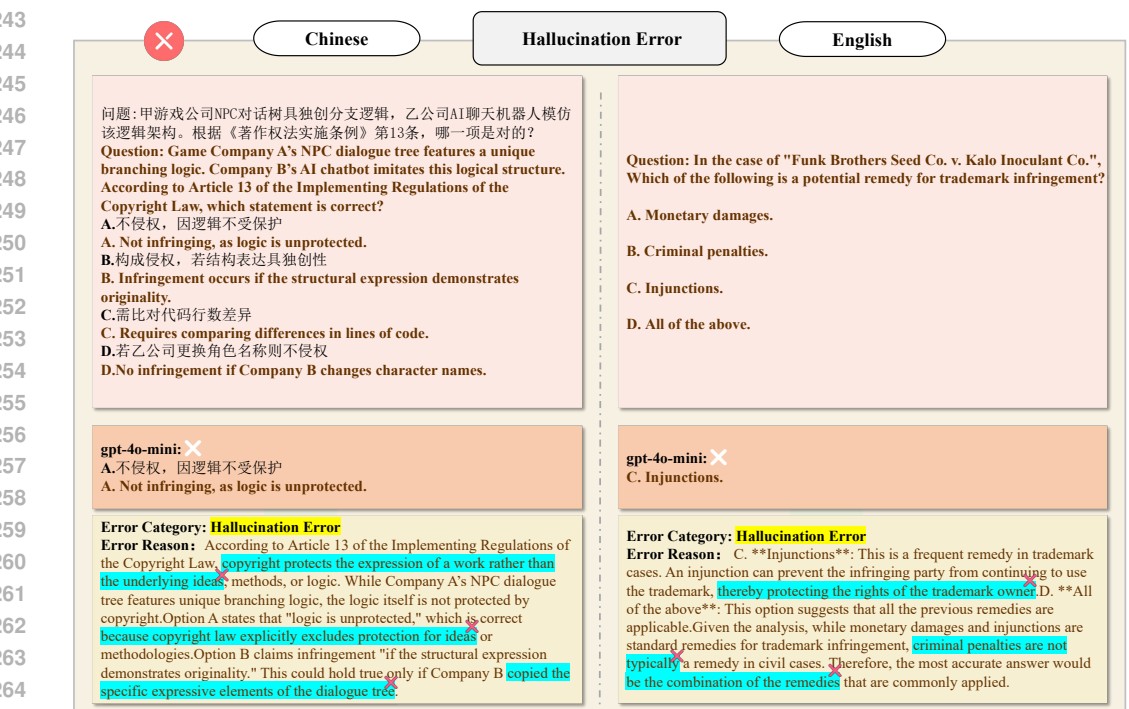

Figure 11: Hallucination error case study.

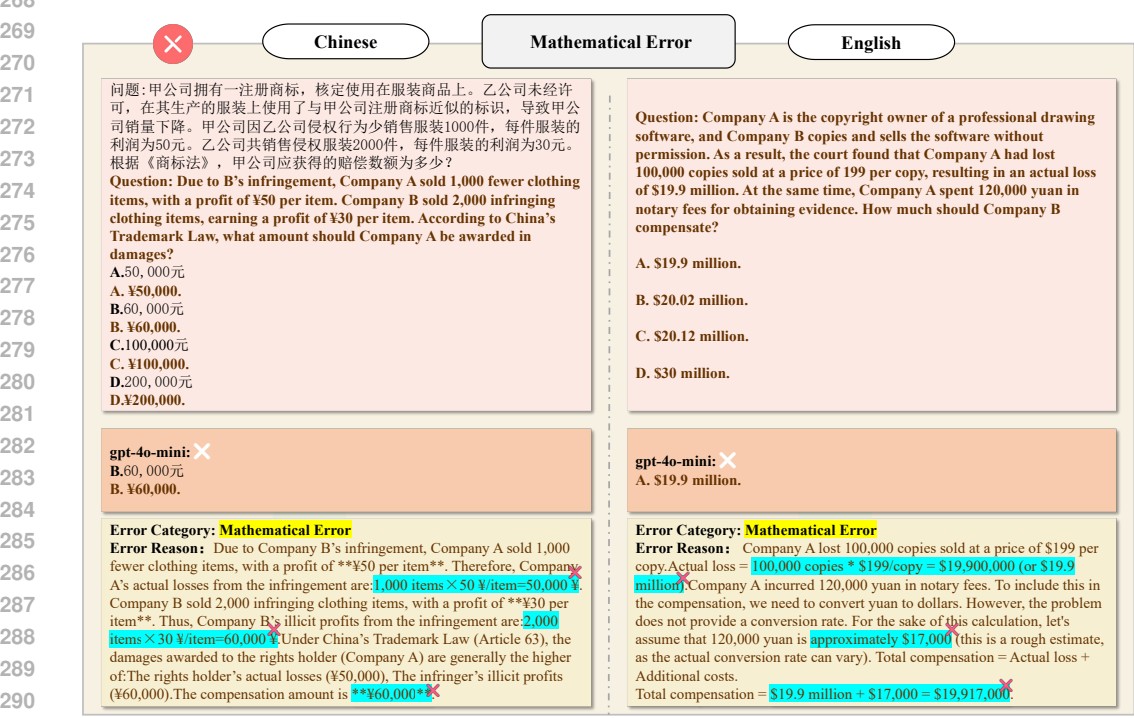

Figure 12: Mathematical error case study.

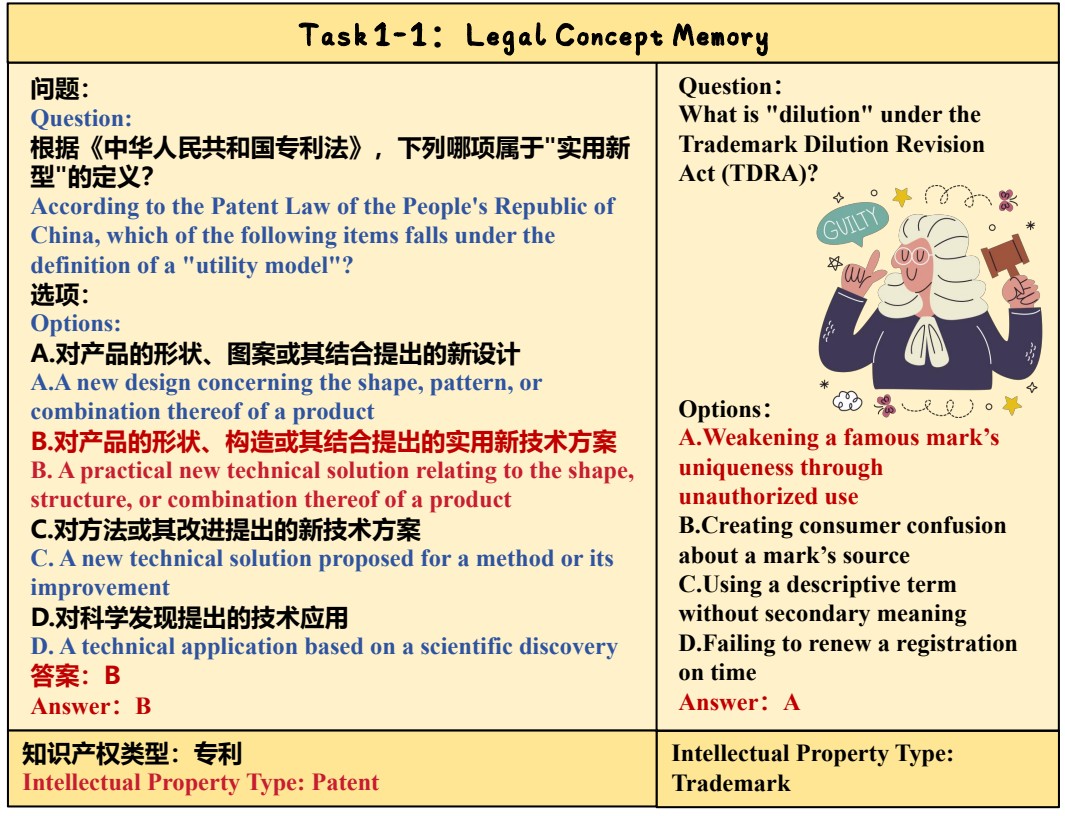

Figure 13: Data example of task 1-1.

---

### Task 1-2: Legal Clause Memory

**问题：**
**Question:**
根据《中华人民共和国商标法(2019年修正)》第10条，下列哪种标志可以作为商标使用?
According to Article 10 of the Trademark Law of the People's Republic of China (2019 Amendment), which of the following signs may be used as a trademark?

**选项：**
**Options:**
A.与外国军旗近似的图形
A.Signs that are similar to the military flags of foreign countries
B.带有民族歧视性的文字
B. Text or symbols with ethnic discrimination
C.县级以上行政区划地名但具有其他含义的
C. Names of administrative divisions at or above the county level that have other meanings
D.与"红十字"名称相同的标志
D. Signs identical to the name 'Red Cross'

**答案：C**
**Answer：C**

**知识产权类型：商标**
**Intellectual Property Type: Trademark**

**Question：**
Under 17 U.S.C. § 110(1), what is allowed in classroom teaching?

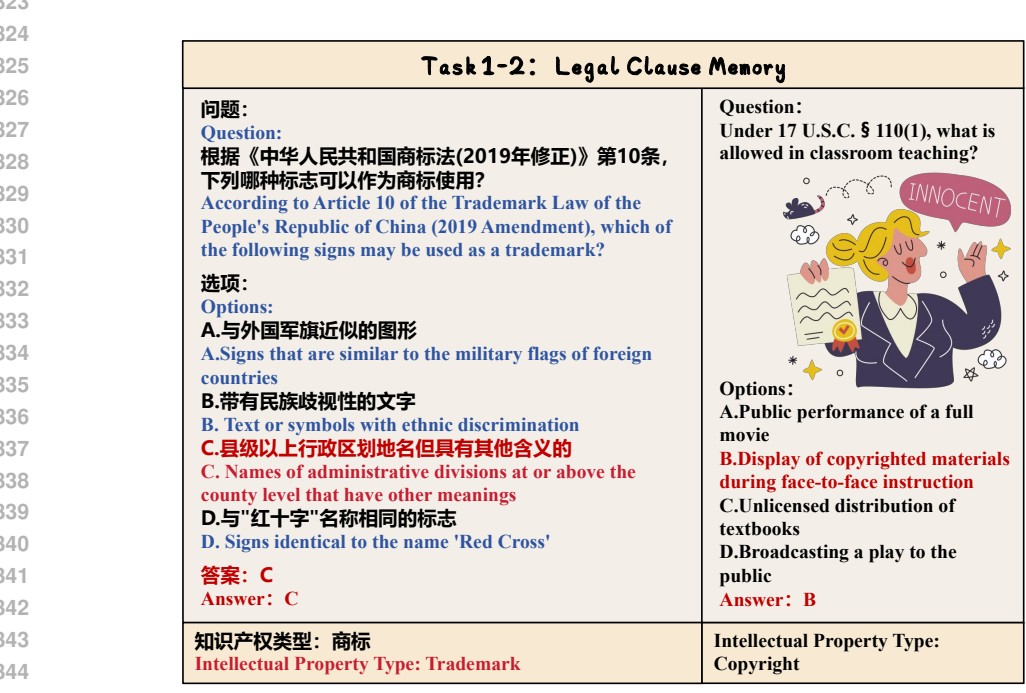

**Options：**
A.Public performance of a full movie
B.Display of copyrighted materials during face-to-face instruction
C.Unlicensed distribution of textbooks
D.Broadcasting a play to the public

**Answer：B**

**Intellectual Property Type: Copyright**

Figure 14: Data example of task 1-2.

---

### Task 1-3: Legal Evolution

**问题：**
**Question:**
相较于2008年《中华人民共和国专利法》，2020年修订版本新增了哪项关于药品专利期限补偿的制度?
Compared to the 2008 Patent Law of the People's Republic of China, what new pharmaceutical patent term extension system was introduced in the 2020 amended version?

**选项：**
**Options:**
A.允许专利期限延长至20年以上
A.Permits the extension of patent terms beyond 20 years
B.针对仿制药上市给予专利豁免期
B. Provides a patent exemption period for generic drug marketing approval
C.要求药品专利必须进行期限登记
C. Requires mandatory term registration for pharmaceutical patents
D.对创新药专利给予最长5年的期限补偿
D. Grants a maximum 5-year term extension for innovative drug patents

**答案：D**
**Answer：D**

**知识产权类型：专利**
**Intellectual Property Type: Patent**

**Question：**
Compared to 2015, what adjustment was made to patent term extension under 35 U.S.C. § 154(b)(1)(B) in 2022?

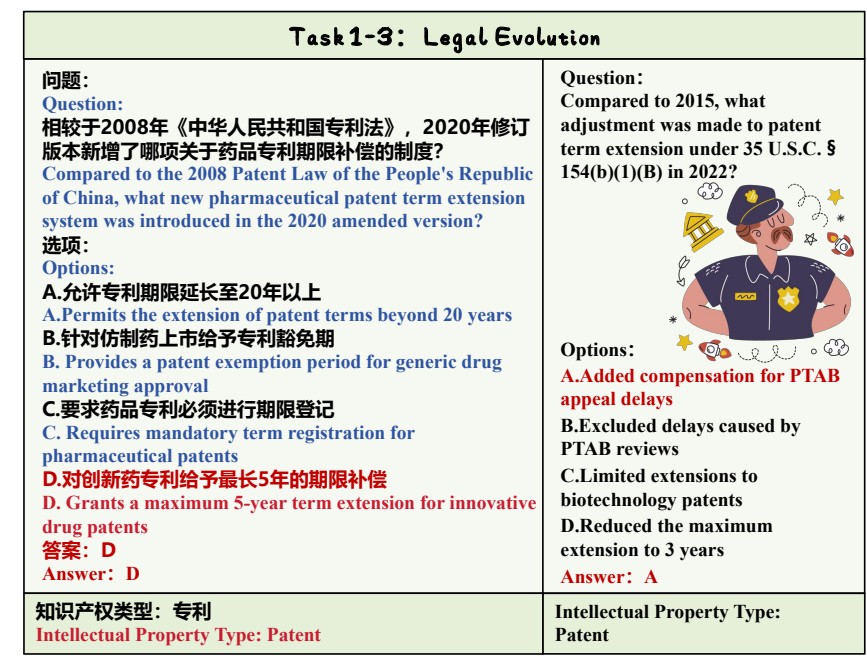

**Options：**
A.Added compensation for PTAB appeal delays
B.Excluded delays caused by PTAB reviews
C.Limited extensions to biotechnology patents
D.Reduced the maximum extension to 3 years

**Answer：A**

**Intellectual Property Type: Patent**

Figure 15: Data example of task 1-3.

| Task 1-4: Typical Case Memory | |
|---|---|
| 问题：
Question：
"郑州某研究所与陈某侵害植物新品种权纠纷案"中，陈某被认定侵权的行为是什么？
In the case of Zhengzhou Research Institute v. Chen regarding infringement of new plant variety rights dispute,What are the acts of infringement that Chen was found to have committed?
选项：
Options：
**A. 未经许可繁育"天使红"石榴新品种的繁殖材料**
**A.Propagation of reproductive materials of the new pomegranate variety 'Angel Red' without authorization**
B.销售假冒"天使红"石榴的果实
B. Sale of counterfeit fruits of the 'Angel Red' pomegranate variety
C.未经授权使用"天使红"商标
C. Unauthorized use of the 'Angel Red' trademark
D.未支付品种许可费用
D. Non-payment of variety licensing fees
答案：A
Answer：A | Question：
In the case of "Kewanee Oil Co. v. Bicron Corp.", what was the Supreme Court's decision regarding the preemption of state trade secret law by federal patent law?

Options：
A.The Court did not address the issue of preemption
B.The Court ruled that preemption applied only in specific cases
C.State trade secret law was preempted by federal patent law
**D.State trade secret law was not preempted by federal patent law**
Answer：D |
| 知识产权类型：植物新品种
Intellectual Property Type: New Plant Variety | Intellectual Property Type:
Trade Secret |

Figure 16: Data example of task 1-4.

| Task 1-5-1: Patent IPC Classification | |
|---|---|
| 问题：
Question：
标题：基于物联网的数字孪生城市交通灯控制系统
Title: IoT-based Digital Twin Urban Traffic Light Control System
摘要：本发明公开了基于物联网的数字孪生城市交通灯控制系统，涉及智能控制的技术领域，包括采集模块、分析模块和控制模块，计算拥挤度变化率，得到集中时间段，计算历史自然数据的权重，构造干扰函数，计算干扰时长和新的集中时间段，建立交通模型，将新的拥挤度变化率与拥挤度变化量阈值进行比较，执行第一操作，并建立第一映射关系。本发明通过实时监测和分析交通流量、拥挤度变化情况，以及考虑自然环境因素的影响，优化交通流的通行时间分配，动态调整交通信号灯的控制策略，通过历史交通数据和自然数据的分析，能够预测未来的交通情况，提前做好相应的交通信号调整和路线规划，物联网技术的应用使得交通灯控制系统能够实现远程监控和集中管理。
Abstract: The present invention discloses an IoT-based digital twin urban traffic signal control system...(Omit)
答案：G08G1/08
Answer：G08G1/08 | Question：
Title:System and method for migrating agents between mobile devices
Abstract:Mobile agents can be deployed to location aware mobile devices within specific regions of interest to achieve specific goals in respect of events occurring in the region of interest. In order to ensure that the agent can persist within the region of interest until the agent goals are achieved, the agent is configured to locate other devices within the region of interest and to propagate itself, by moving or copying itself, to those other devices. When a device hosting the agent exits the region of interest, the agent is terminated, thereby freeing device resources.
Answer：A01D34/43 |
| 知识产权类型：专利
Intellectual Property Type: Patent | Intellectual Property Type:
Patent |

Figure 17: Data example of task 1-5-1.

## Task 1-5-2: Patent CPC Classification

**Title:**
**Multipurpose machine for cultivating trees**

**Abstract：**
**A multipurpose machine for cultivating trees, comprising an inverted U-shape structure that enables the machine to pass over existing trees to carry out pruning, disinfection or fruit picking tasks, provided at the bottom with wheels, driven by at least one motor that autonomously facilitates the movement thereof, and respective upper frames that telescopically couple to each other, being driven by respective cylinders to move the portion of the structure on the right with respect to the one on the left in order to vary the width of the machine. Likewise, the machine has the ability to raise or lower the upper structure of the same to adapt it to the height of the trees to be cultivated.s.**

**Answer：A01D46/30**

**Intellectual Property Type:**
**Patent**

Figure 18: Data example of task 1-5-2.

## Task 1-6: IP Element Identifcation

问题：
**Question:**
餐饮连锁企业注册声音商标（特定叫卖声），竞争对手使用相似韵律不同歌词进行宣传。侵权认定关键在于什么?
**What is the key to determining infringement when a restaurant chain registers a sound trademark (a specific call), and its competitor uses a similar rhythm but different lyrics for promotion?**

选项：
**Options:**
**A. 商标显著性获得时间**
**A. Time of obtaining distinctiveness of the trademark**
**B. 声音商标的混淆可能性判定**
**B. Determination of Confusion Potential of Sound Trademarks**
**C. 描述性使用正当性**
**C. Legitimacy of Descriptive Use**
**D. 驰名商标跨类保护**
**D. Cross class protection of well-known trademarks**
**答案：B**
**Answer：B**

知识产权类型：专利
**Intellectual Property Type: Patent**

**Question:**
**Please select the correct answer from A, B, C, and D.**
**A lab engineers synthetic DNA sequences encoding Shakespearean sonnets. Competitors replicate them. What governs?**

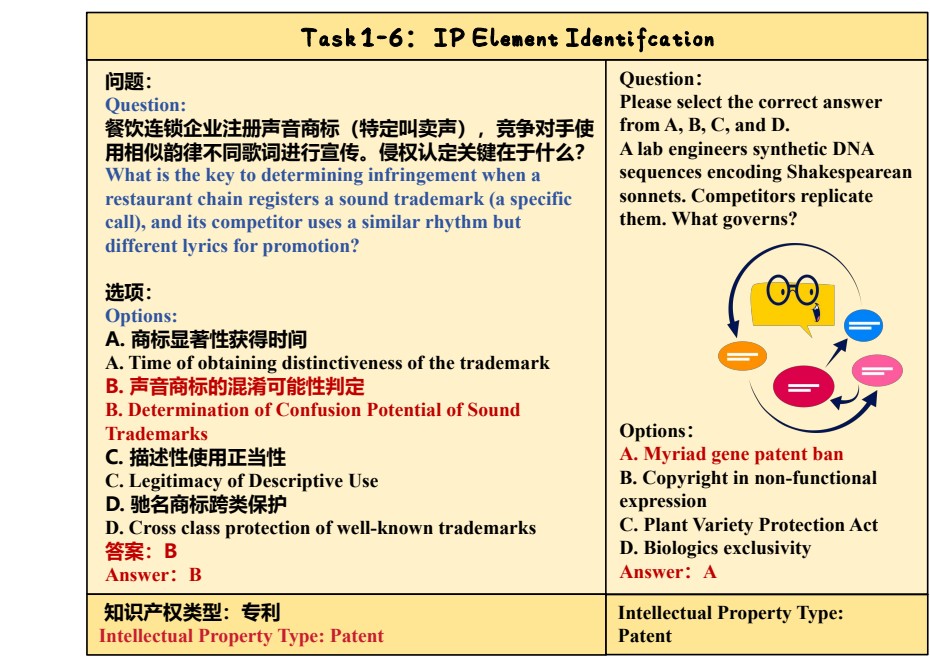

**Options:**
**A. Myriad gene patent ban**
**B. Copyright in non-functional expression**
**C. Plant Variety Protection Act**
**D. Biologics exclusivity**
**Answer：A**

**Intellectual Property Type:**
**Patent**

Figure 19: Data example of task 1-6.

| Task 1-7: Process Guidance | |
|---|---|
| 问题：
Question：
商标注册申请人可以通过什么方式提交《中华人民共和国商标法实施条例》规定的申请？
How can trademark registration applicants submit applications as stipulated in the Implementing Regulations of the Trademark Law of the People's Republic of China?

选项：
Options：
A. 只能通过纸质方式
A. Can only be done in paper form
B. 只能通过电子方式
B Can only be done through electronic means
C. 可以通过纸质或电子方式
C. Can be done through paper or electronic means
D. 只能通过代理机构提交
D Can only be submitted through an agency

答案：C
Answer：C | Question：
Please select the correct answer from A, B, C, and D.Who makes the decision that the international application is considered withdrawn?

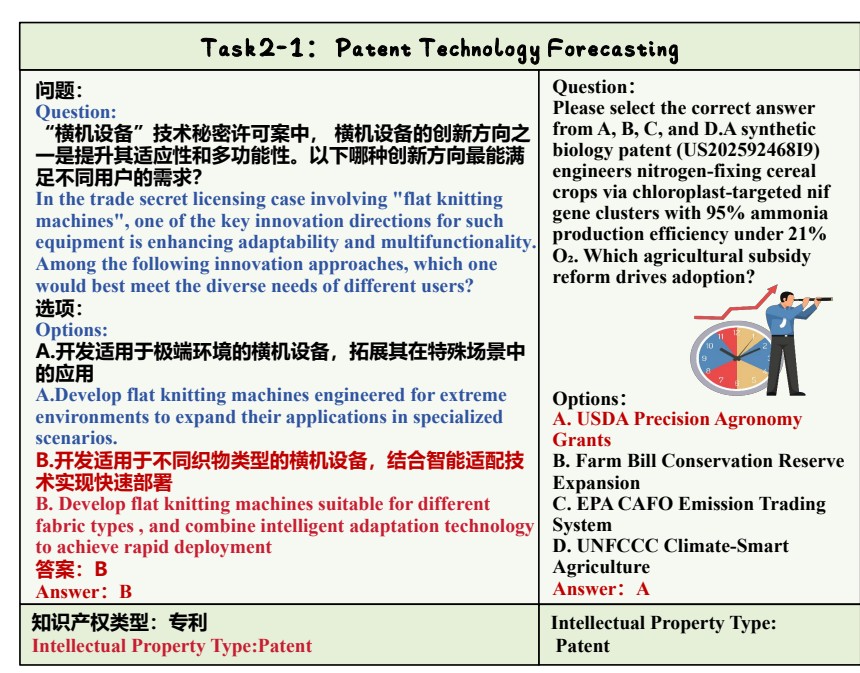

Options：
A. International Bureau
B. Receiving Office
C. International Searching Authority
D. Chinese
Answer：B |
| 知识产权类型：商标
Intellectual Property Type:Trademark | Intellectual Property Type:
Patent |

Figure 20: Data example of task 1-7.

| Task 2-1: Patent Technology Forecasting | |
|---|---|
| 问题：
Question：
"横机设备"技术秘密许可案中，横机设备的创新方向之一是提升其适应性和多功能性。以下哪种创新方向最能满足不同用户的需求？
In the trade secret licensing case involving "flat knitting machines", one of the key innovation directions for such equipment is enhancing adaptability and multifunctionality. Among the following innovation approaches, which one would best meet the diverse needs of different users?
选项：
Options：
A.开发适用于极端环境的横机设备，拓展其在特殊场景中的应用
A.Develop flat knitting machines engineered for extreme environments to expand their applications in specialized scenarios.
B.开发适用于不同织物类型的横机设备，结合智能适配技术实现快速部署
B. Develop flat knitting machines suitable for different fabric types , and combine intelligent adaptation technology to achieve rapid deployment
答案：B
Answer：B | Question：
Please select the correct answer from A, B, C, and D.A synthetic biology patent (US20259246819) engineers nitrogen-fixing cereal crops via chloroplast-targeted nif gene clusters with 95% ammonia production efficiency under 21% $O_2$. Which agricultural subsidy reform drives adoption?

Options：
A. USDA Precision Agronomy Grants
B. Farm Bill Conservation Reserve Expansion
C. EPA CAFO Emission Trading System
D. UNFCCC Climate-Smart Agriculture
Answer：A |
| 知识产权类型：专利
Intellectual Property Type:Patent | Intellectual Property Type:
Patent |

Figure 21: Data example of task 2-1.

## Task2-2: Infringement Behavior Determination

**问题:**
**Question:**
甲画家作品被用于训练AI模型，乙公司生成"梵高风格化"图片销售。根据《著作权法实施条例》第21条，哪一项是对的？
Artist A's works are used to train AI models, while Company B generates "Van Gogh stylized" images for sale. According to Article 21 of Regulation for the Implementation of the Copyright Law, which one is correct?

**选项:**
**Options:**
A.不侵权，因风格不受保护
A. Not infringing, style is not protected
B.构成侵权，若训练数据包含未授权作品
B. If the training data contains unauthorized works, it constitutes infringement
C.需比对笔触相似度
C. The similarity of brushstrokes needs to be compared
D.若声明"AI辅助创作"则不侵权
D. If it is declared as "AI-assisted creation", there is no infringement

答案: B
Answer: B

知识产权类型：著作权
Intellectual Property Type: Copyright

**Question:**
A company uses a logo that has a similar color scheme and general shape to another company's logo but different lettering. Is this trademark infringement?

**Options:**
A. Yes, if the overall similarity can lead to consumer confusion
B. No, because of the different lettering
C. Only if the other company's logo is very new
D. Only if the company uses the logo on a large number of products

Answer: A

Intellectual Property Type: Trademark

Figure 22: Data example of task 2-2.

## Task2-3: Compensation Calculation

**问题:**
**Question:**
某游戏公司侵权使用他人美术设计，法院认定：权利人许可费为每幅设计5万元；侵权使用设计20幅；侵权游戏月收入100万元（利润率60%）。赔偿金额应为？
A game company infringed upon others' art designs. The court ruled that the licensing fee for each design was 50,000 yuan. Twenty infringing designs were used; The monthly revenue of infringing games is 1 million yuan (with a profit margin of 60%). What should be the amount of compensation?

**选项:**
**Options:**
A. 20×5万＝100万
A. 20× 50,000 ＝1 million
B. 100万×60%×12月＝720万
B. 1 million ×60%×12 months = 7.2 million
C.max(100万,720万)＝720万
C. max(1 million, 7.2 million)= 7.2 million
D. 500万元以下（如500万）
D. Less than 5 million yuan (e.g. 5 million)

答案: A
Answer: A

知识产权类型：著作权
Intellectual Property Type: Copyright

**Question:**
The defendant's misappropriation of trade secrets resulted in a loss of $500,000 in the plaintiff's income and a profit of $300,000 for the defendant. The court awarded damages on the basis of "unjust enrichment". What is the maximum number of damages a plaintiff can receive under Section 3(b)(1) of the Uniform Trade Secrets Act (UTSA)?

**Options:**
A. 300,000
B. 500,000
C. 800,000
D. 1,500,000

Answer: B

Intellectual Property Type: Trade secret

Figure 23: Data example of task 2-3.

| Task2-4: Patent Valuation | |
|---|---|
| 问题：
Question:
**某抗癌药化合物专利剩余保护期5年，年销售额达12亿元。若该药被纳入国家医保集采目录导致单价下降60%，其专利价值趋势是？**
The remaining patent protection period of a certain anti-cancer drug compound is five years, with an annual sales volume reaching 1.2 billion yuan. If the drug is included in the national medical insurance centralized procurement list, resulting in a 60% drop-in unit price, what is the trend of its patent value?
选项：
Options:
**A.因销量暴增翻倍**
A. Because the sales volume has doubled sharply
**B.因利润压缩大幅贬值**
B. It depreciated significantly due to the compression of profits
**C.转为技术秘密后增值**
C. It increases in value after being converted into a technical secret
**D.因政策保护维持不变**
D. It remains unchanged due to policy protection
**答案：B**
Answer：B | Question:
**Tesla's electric vehicle charging patents (e.g., U.S. 20170171460) were licensed to Rivian under a fixed-fee model. Why might this structure reduce valuation risk for Tesla?**

Options:
A. Fixed fees guarantee upfront cash flow
B. Avoids dependency on fluctuating royalty rates
C. Eliminates litigation risks
D. Both A and B
Answer：D |
| 知识产权类型：专利
Intellectual Property Type: Patent | Intellectual Property Type: Patent |

Figure 24: Data example of task 2-4.

| Task2-5: Trade Secret Requirements |
|---|
| 问题：
Question:
**根据《反不正当竞争法》，下列哪项不属于商业秘密的构成要件？**
According to the Anti-Unfair Competition Law, which of the following is not a constituent element of trade secrets?

选项：
Options:
**A.不为公众所知悉**
A. Unknown to the public
**B.具有商业价值**
B. Have commercial value
**C.权利人已采取合理保密措施**
C. The right holder has taken reasonable confidentiality measures
**D.已向行政机关登记备案**
D. It has been registered and filed with the administrative authority
**答案：D**
Answer：D |
| 知识产权类型：商业秘密
Intellectual Property Type: Trade secret |

Figure 25: Data example of task 2-5.

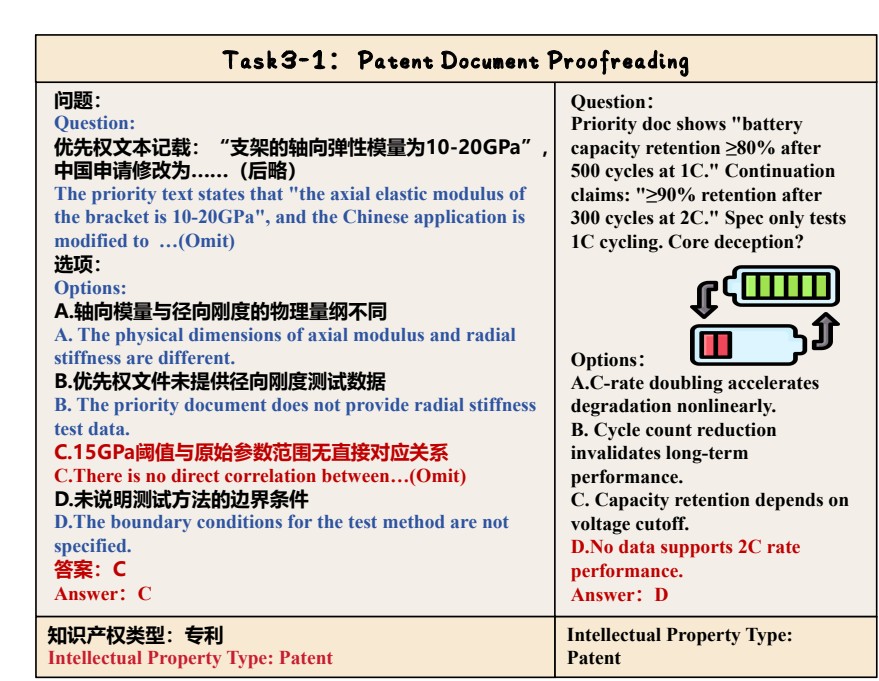

Figure 26: Data example of task 3-1.

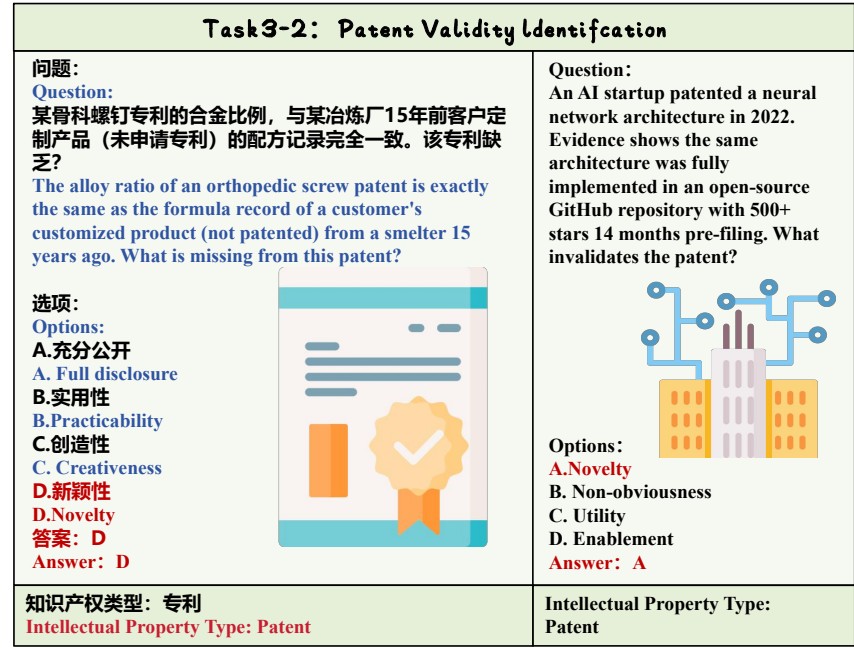

Figure 27: Data example of task 3-2.

| Task3-3: Patent Match | |
|---|---|
| 问题：
Question:
请从A,B,C,D四个选项中选出与下述专利最相似的专利序号，该序号是？……（后略）
Please select the patent number that is most similar to the following patent from the four options A, B, C and D. What is the number? …(Omit)
选项：
Options:
A.一种多功能洁面仪……（后略）
A. A multi-functional facial cleanser…(Omit)
B.一种具有吸取毛发功能的宠物毛刷及其使用方法……（后略）
B. A pet brush with hair absorption function and its use method…(Omit)
C.一种吸水刷头，包括……（后略）
C.A water-absorbing brush head, including…(Omit)
D.本发明公开了一种多功能化妆刷……（后略）
D.The invention discloses a multi-functional makeup brush…(Omit)
答案：C
Answer：C | Question：
Provided is an adipose tissue preservation solution, comprising: polyethylene glycol 400, human albumin, norfloxacin, low molecular dextran, and Ac-DEVD-CHO…(Omit)

Options：
A.Provided in the present invention are a mesenchymal stem cell injection…(Omit)
B. The present invention provides a mesenchymal stem cell injection…(Omit)
C. Provided in the present invention is a cell freezing medium for clinical use…(Omit)
D. The present invention provides a mesenchymal stem cell injection…(Omit)
Answer：D |
| 知识产权类型：专利
Intellectual Property Type: Patent | Intellectual Property Type: Patent |

Figure 28: Data example of task 3-3.

| Task3-4: Rights Attribution Analysis | |
|---|---|
| 问题：
Question:
丙公司技术员张某，完成本职工作之余，利用业余时间、自行购买材料研发 "新型焊接工具" ……（后略）
Question:Zhang, a technician of Company C, developed a "new welding tool" by himself using spare time and materials purchased by himself after completing his own work…(Omit)
选项：
Options:
A.丙公司，因张某是公司员工。
A. Company C, because…(Omit)
B.张某，因非职务发明创造。
B.Zhang, because it is not an invention made in the course of his duties.
C.双方共有，因张某员工身份关联。
C.The two parties share the same identity due to Zhang's employee status.
D.归当地政府，鼓励创新。
D.Return to the local government…(Omit)
答案：B
Answer：B | Question：
A biotechnology researcher invents a novel gene-editing tool while employed at University X, using university lab equipment and government grant funds. The researcher's employment contract states, "All inventions arising from university-funded projects belong to the institution." Who owns the patent?



Options：
A. The researcher individually.
B. University X.
C. The government funding agency.
D.Shared between the researcher and University X.
Answer：B |
| 知识产权类型：专利
Intellectual Property Type: Patent | Intellectual Property Type: Patent |

Figure 29: Data example of task 3-4.

**Task3-5: Patent Application Examination**

**Question:**
**METHOD FOR PREVENTING OVERLOAD IN MOBILE TELEPHONE NETWORKS BY USING 'ALWAYS-ON' IN THE CASE OF A CALL FROM A MOBILE TELEPHONE**
**Abstract:**
**The invention relates to a method for preventing overload in telecommunications networks with IMS by always-on for a call generated by a user, in which process said user makes a PDP Context request and a GGSN of said network provides a free IP address to him or her and the PDP Context becomes active; and wherein an S-CSCF of the network creates a record in which an association between said IP address and a characteristic identity of the IMS network is included…(Omit)**

**Options:**
**A. Allowed**
**B. Refuse**
**Answer: A**

**Intellectual Property Type:**
**Patent**

Figure 30: Data example of task 3-5.

**Task4-1: Abstract Generation**

**问题:**
**Question:**
**#权利要求**
**#Claims**
**1.一种水田搅浆平地机，包括牵引架（1）、挡泥罩（11）、平地刮板（12）、驱动装置（13）、搅拌轴（14）和搅浆刀（15），所述牵引架（1）与行走机械连接，所述牵引架（1）用于对挡泥罩（11）进行固定……（后略）**
**1. A paddy field mixing and leveling machine, comprising a traction frame (1), a mudguard cover (11), a leveling scraper (12), a driving device (13), a mixing shaft (14), and a mixing blade (15), wherein the traction frame (1) is connected to a walking machinery, and the traction frame (1) is used to fix the mudguard cover (11)... (Omitted later) application based on a scientific discovery…(Omit)**
**答案: 本发明涉及一种水田搅浆平地机，主要包括牵引架、挡泥罩……（后略）**
**Answer: The present invention relates to a paddy field mixing and leveling machine, which mainly includes components such as a traction frame, a mud blocking cover…(Omit)**

**知识产权类型: 专利**
**Intellectual Property Type: Patent**

**Question:**
**# Claims**
**1. A method comprising: obtaining circulating white blood cells from a subject; treating the white blood cells with a DNA damaging agent; performing a flow cytometry based functional variant analysis (FVA)...(Omit)**

**Answer:**
**The present invention relates to a method for assessing the functional status of DNA double strand break (DSB) repair pathway genes in circulating white blood cells obtained from a subject...(Omit)**

**Intellectual Property Type:**
**Trademark**

Figure 31: Data example of task 4-1.

<table>
<tr><td colspan="2">Task4-2: Dependent Claim Generation</td></tr>
<tr><td>

问题：
**Question:**
#独立权利要求项
**#Independent claims**
**1.自动麻将机，其特征在于，包括环形推牌装置，所述环形推牌装置包括推牌槽，……（后略）**
**1. An automatic mahjong machine, characterized by comprising a circular card pushing device, wherein the circular card pushing device comprises a card pushing groove….(Omit)**
**请根据给定的独立权利要求项生成其对应的所有从属权利要求。**
**Please generate all dependent claims corresponding to the given independent claims.**
答案：根据权利要求1所述的自动麻将机，其特征在于，所述环形推牌装置的推牌槽的基座上设有驱动装置，用于驱动推牌槽内的麻将牌沿推牌槽移动……（后略）
**Answer: The automatic mahjong machine according to claim 1, characterized in that a driving device is provided on the base of the pushing slot of the circular pushing device, for driving the mahjong tiles in the pushing slot to move along the pushing slot …(Omit)**

</td><td>

**Question:**
**# Independent Claim**
**1. An information processing device comprising: a processor; and a memory storing, movement information relating to movement of a vehicle that is not occupied by a user the movement of the vehicle including movement to change a parking position of the vehicle...**
**Answer:**
**Dependent Claims**
**The information processing device of claim 1, wherein the processor is further configured to receive the instruction from the external device via a wireless communication protocol.\n\n3. The information processing device of claim 1, wherein the memory further stores historical movement information of the vehicle…(Omit)**

</td></tr>
<tr><td>

知识产权类型：专利
**Intellectual Property Type: Patent**

</td><td>

**Intellectual Property Type: Trademark**

</td></tr>
</table>

Figure 32: Data example of task 4-2.

<table>
<tr><td colspan="2">Task4-3: Design-Around Solution Generation</td></tr>
<tr><td>

问题：
**Question:**
**请从A, B, C, D四个选项中选出题目对应的答案。**
**专利涉及散热器的蛇形迂回水道布局，下列哪种改进最可能规避侵权？**
**In the case of Zhengzhou Research Institute v. Chen regarding infringement of new plant variety rights dispute,What are the acts of infringement that Chen was found to have committed?**
选项：
**Options:**
**A. 将水道壁厚减少1mm**
**A.Reduce the wall thickness of the waterway by 1mm**
**B.设计树状分形分支水道**
**B. Design a tree like fractal branch waterway**
**C.增加水道内部纳米涂层**
**C. Add nano coating inside the waterway**
**D.加装水温LED指示灯**
**D. Install water temperature LED indicator light**
答案：B
**Answer: B**

</td><td>

**Question:**
**A synthetic biology patent (US 19,876,543) claims "CRISPRa activation system with dCas9-VPR fusion protein and modified sgRNA containing MS2 aptamers." Which redesign escapes infringement?**

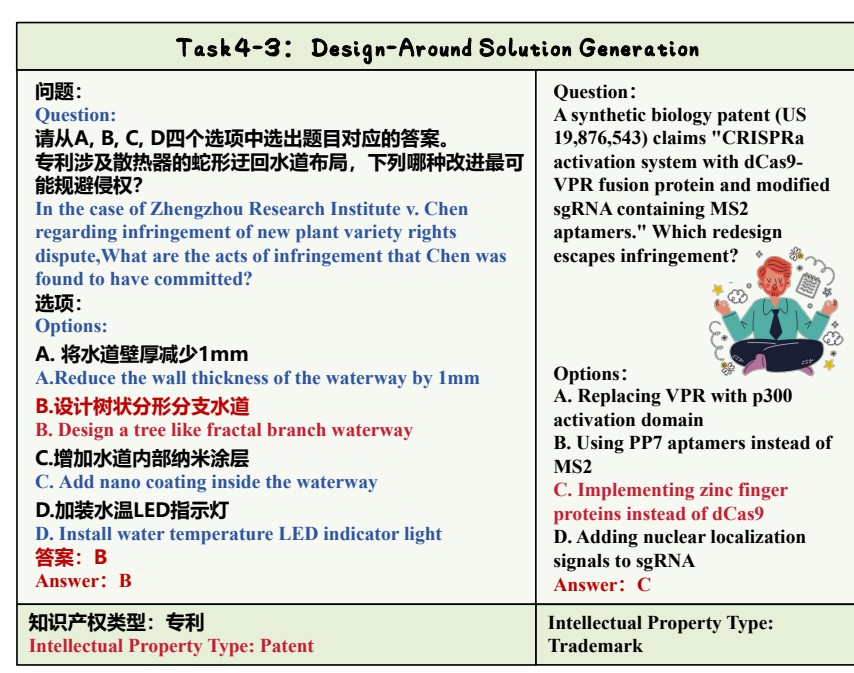

**Options:**
**A. Replacing VPR with p300 activation domain**
**B. Using PP7 aptamers instead of MS2**
**C. Implementing zinc finger proteins instead of dCas9**
**D. Adding nuclear localization signals to sgRNA**
**Answer: C**

</td></tr>
<tr><td>

知识产权类型：专利
**Intellectual Property Type: Patent**

</td><td>

**Intellectual Property Type: Trademark**

</td></tr>
</table>

Figure 33: Data example of task 4-3.

# N  CASE STUDY

We provide extensive case studies for each task, including both correct and erroneous responses in both Chinese and English, as shown from Figure 34 to Figure 75. These case studies offer deeper insight into the scope of the model's capabilities in the field of intellectual property.

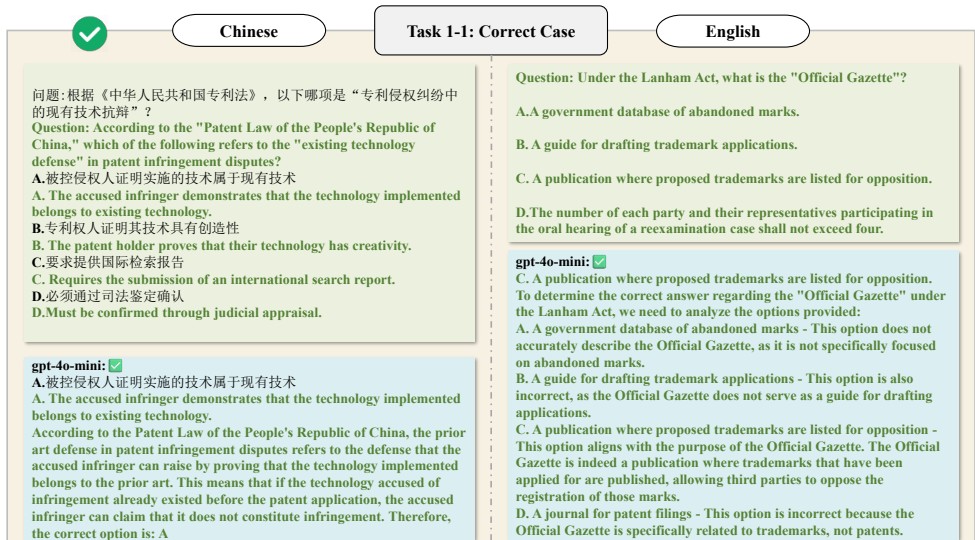

Figure 34: Correct case of task 1-1.

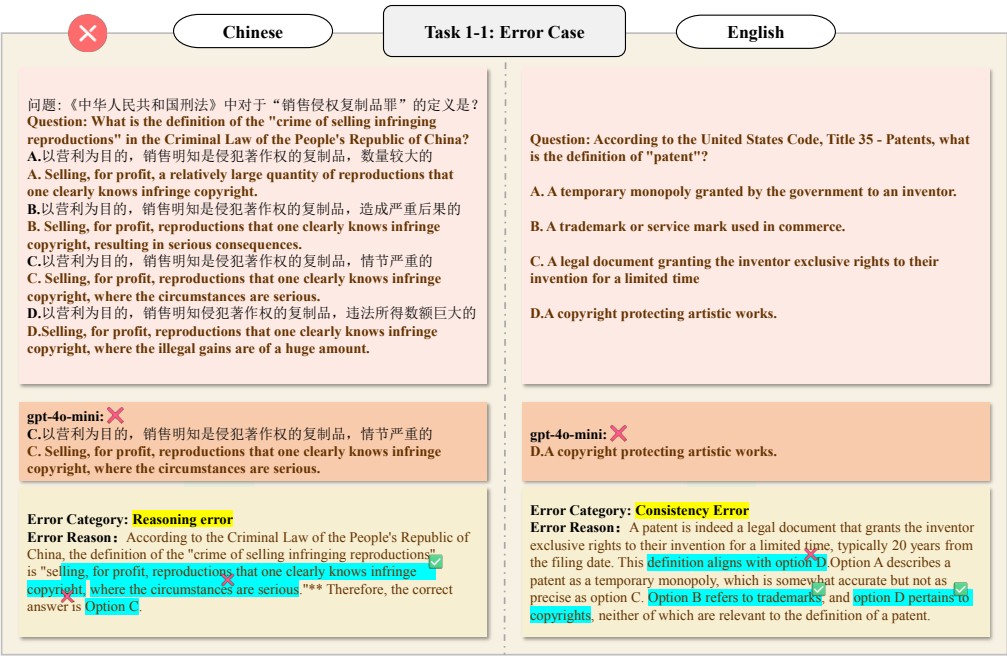

Figure 35: Error case of task 1-1.

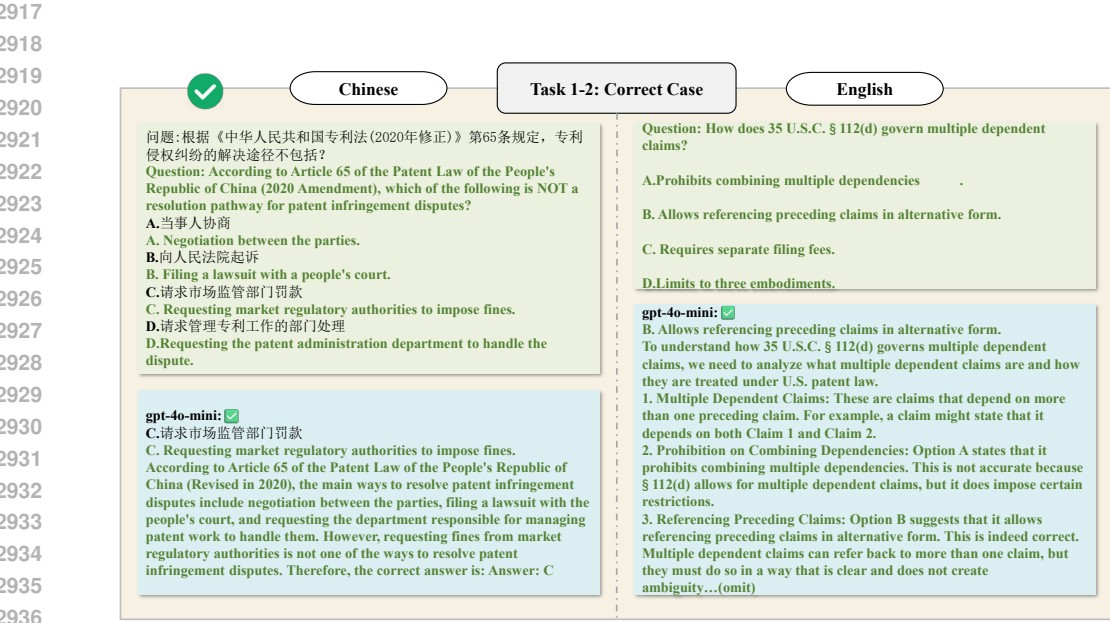

Figure 36: Correct case of task 1-2.

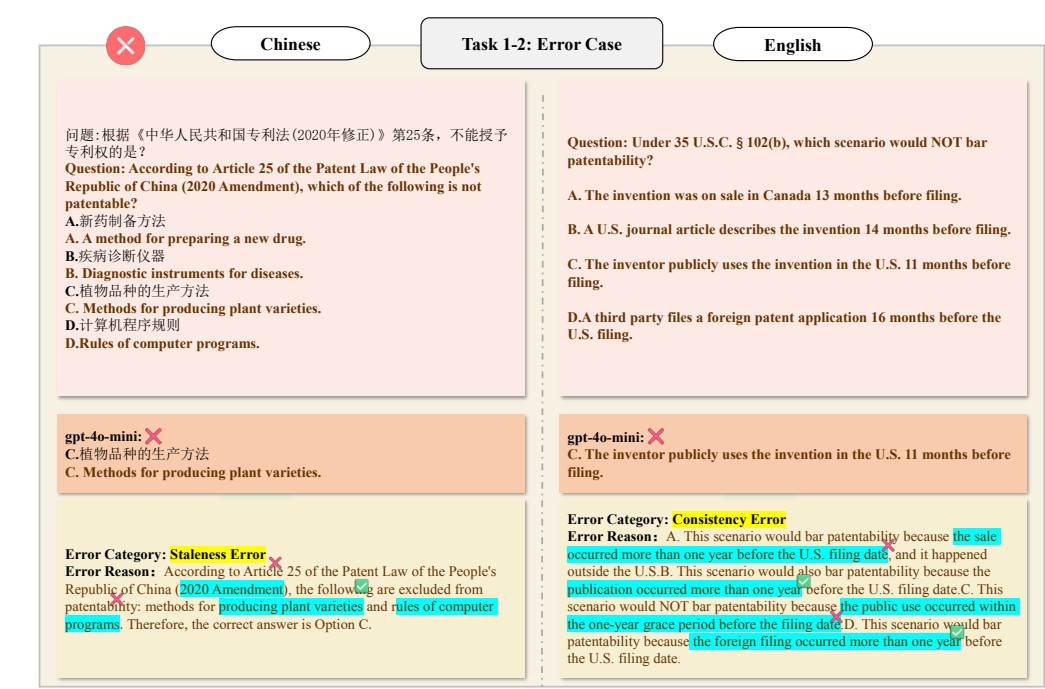

Figure 37: Error case of task 1-2.

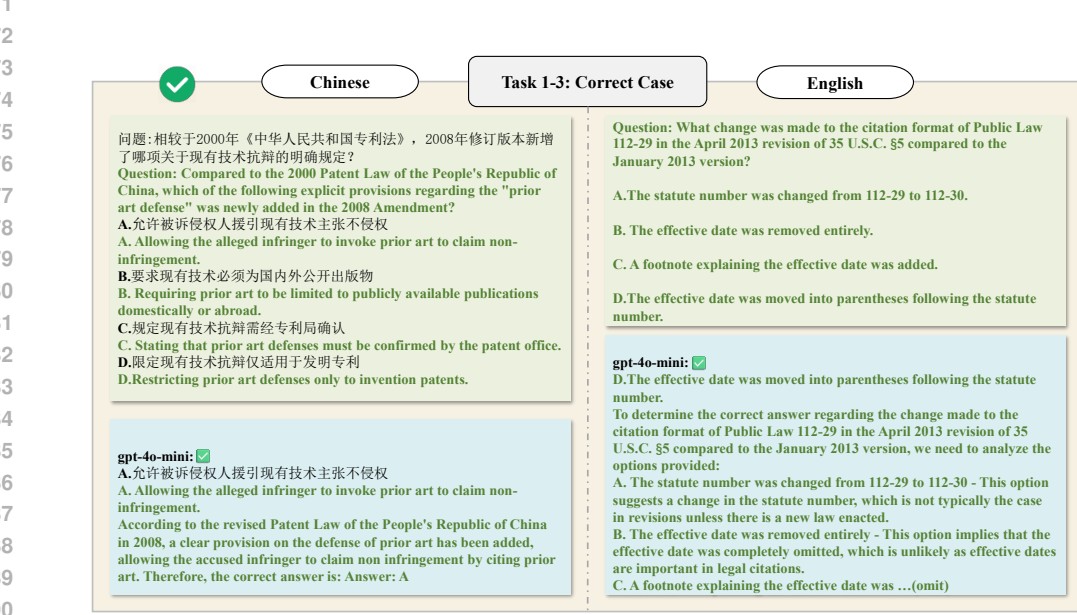

Figure 38: Correct case of task 1-3.

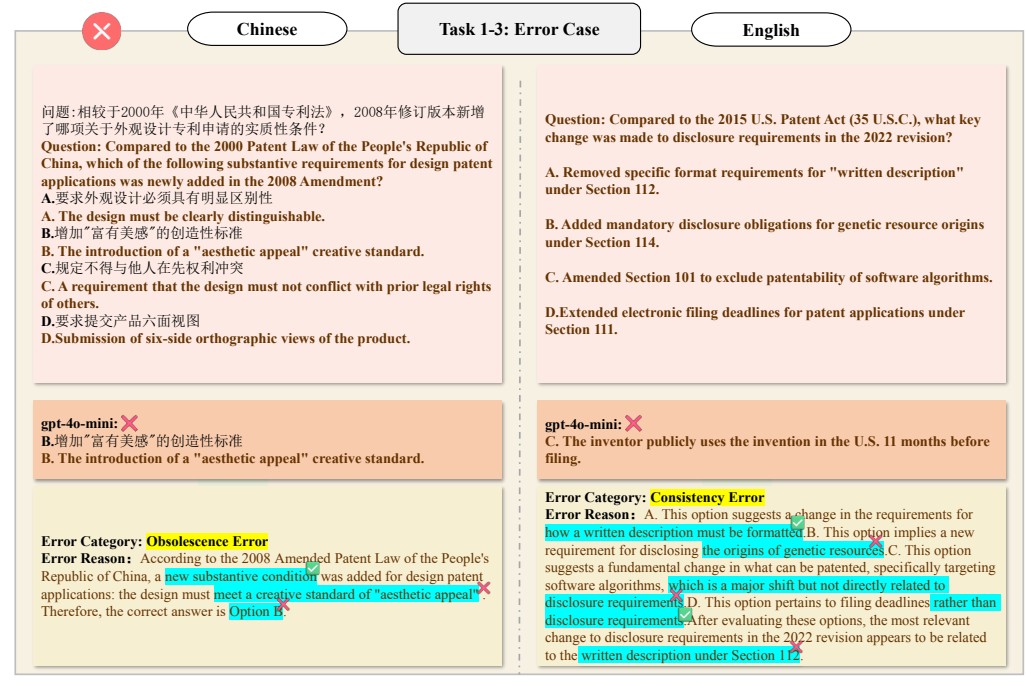

Figure 39: Error case of task 1-3.

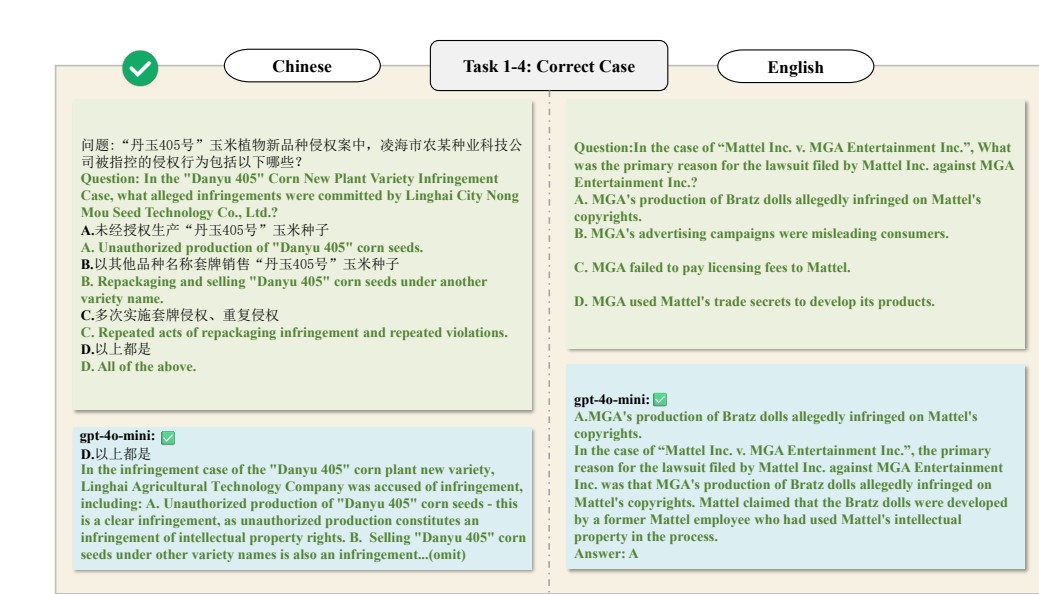

Figure 40: Correct case of task 1-4.

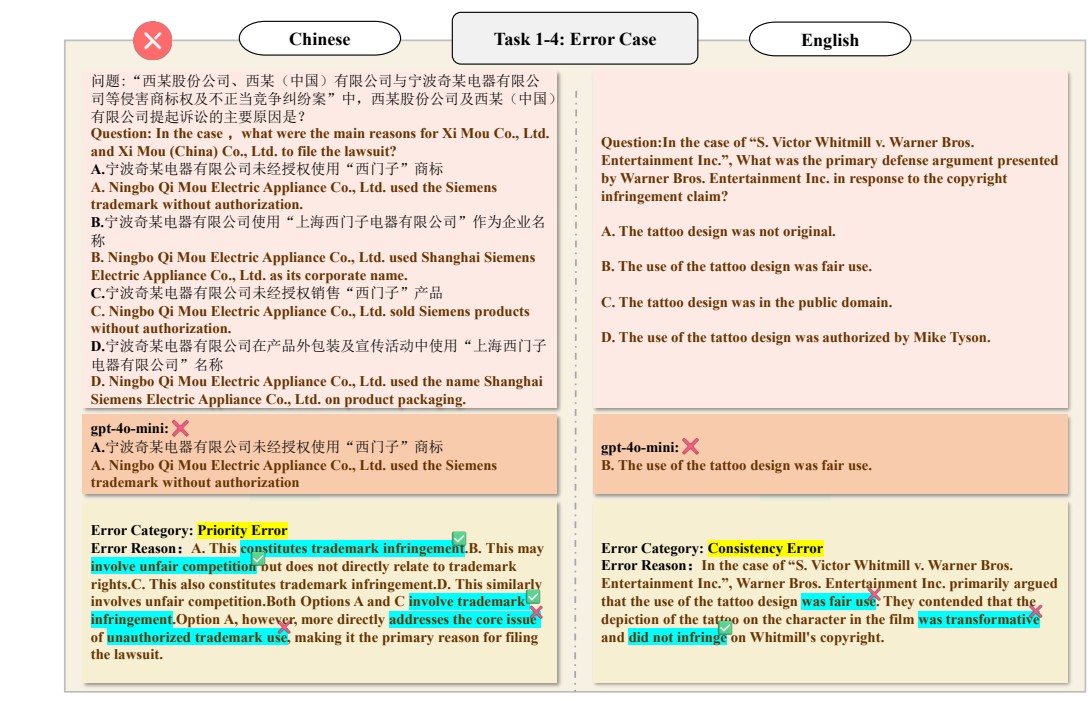

Figure 41: Error case of task 1-4.

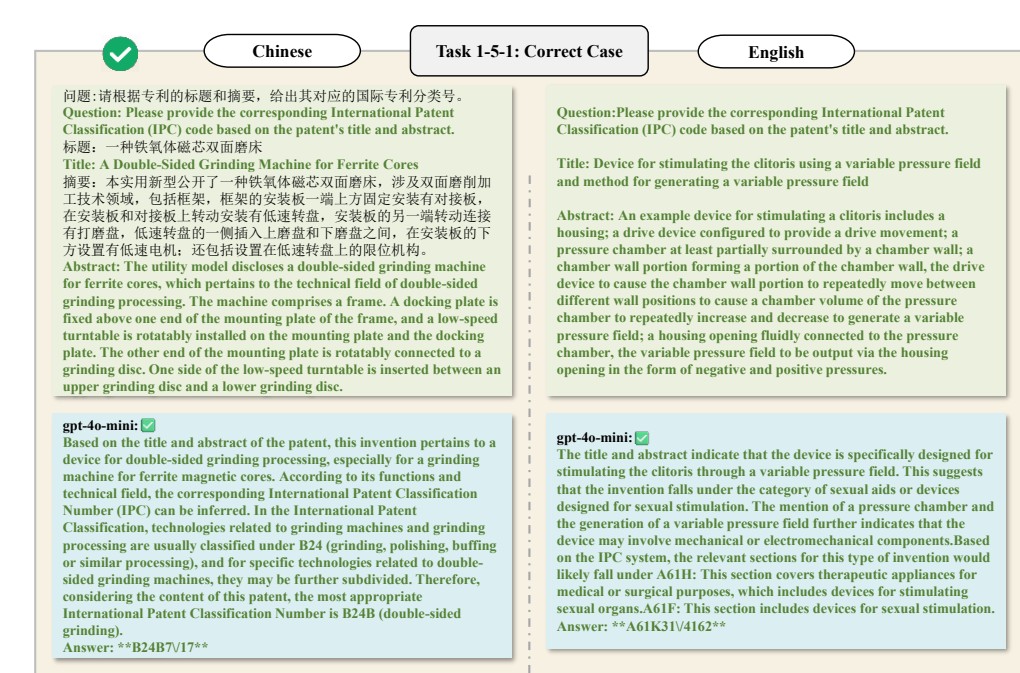

Figure 42: Correct case of task 1-5-1.

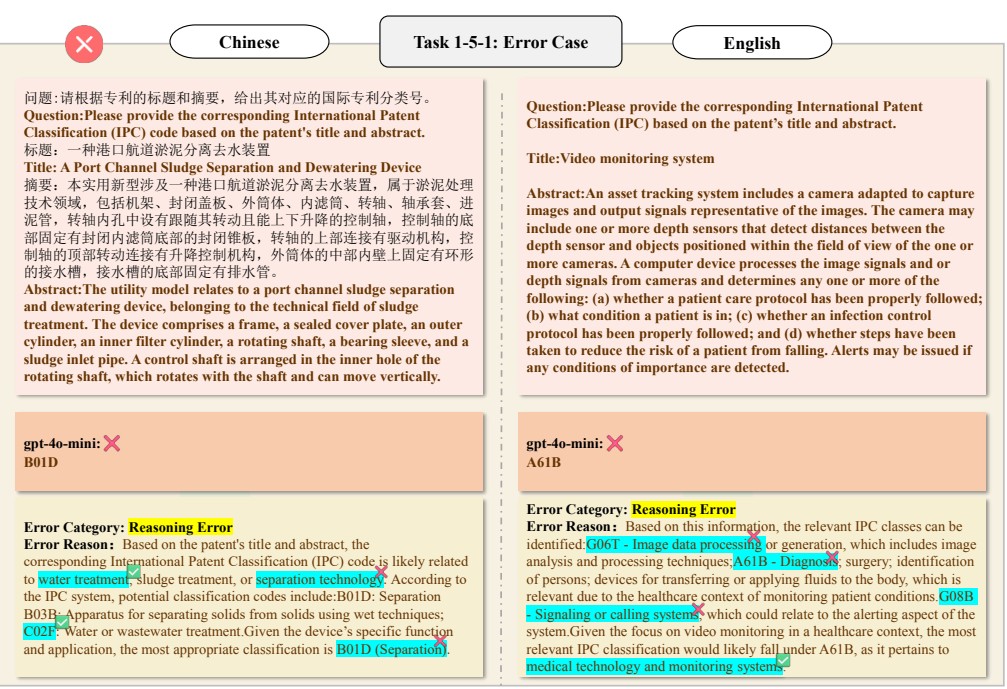

Figure 43: Error case of task 1-5-1.

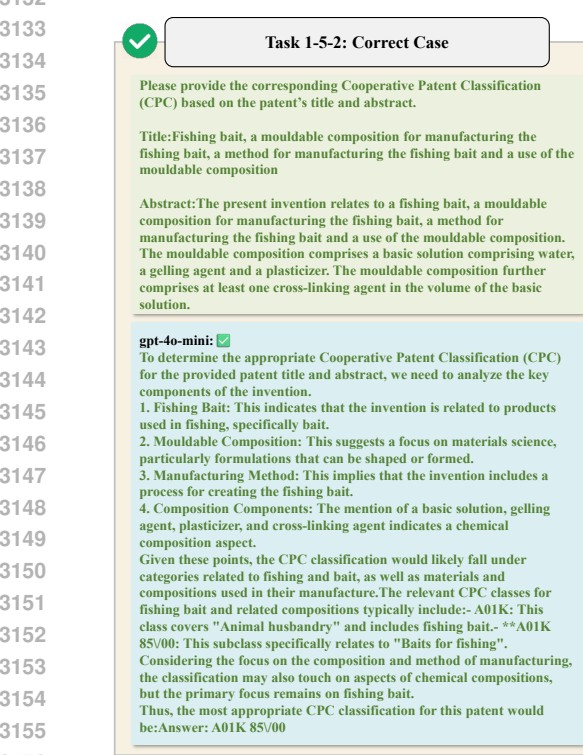

Figure 44: Correct case of task 1-5-2.

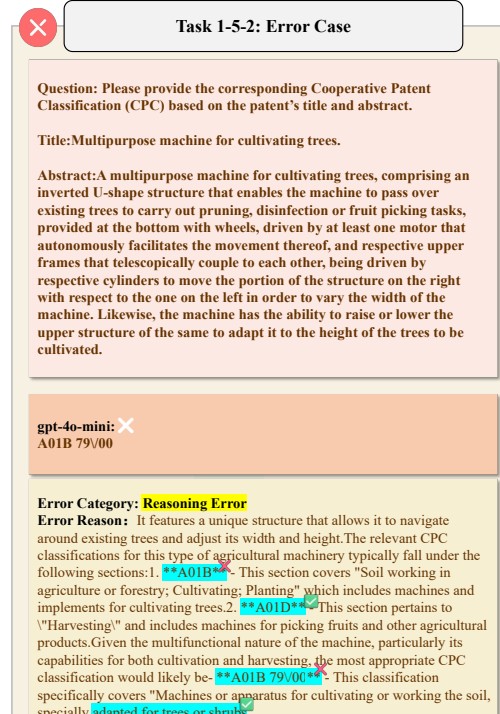

Figure 45: Error case of task 1-5-2.

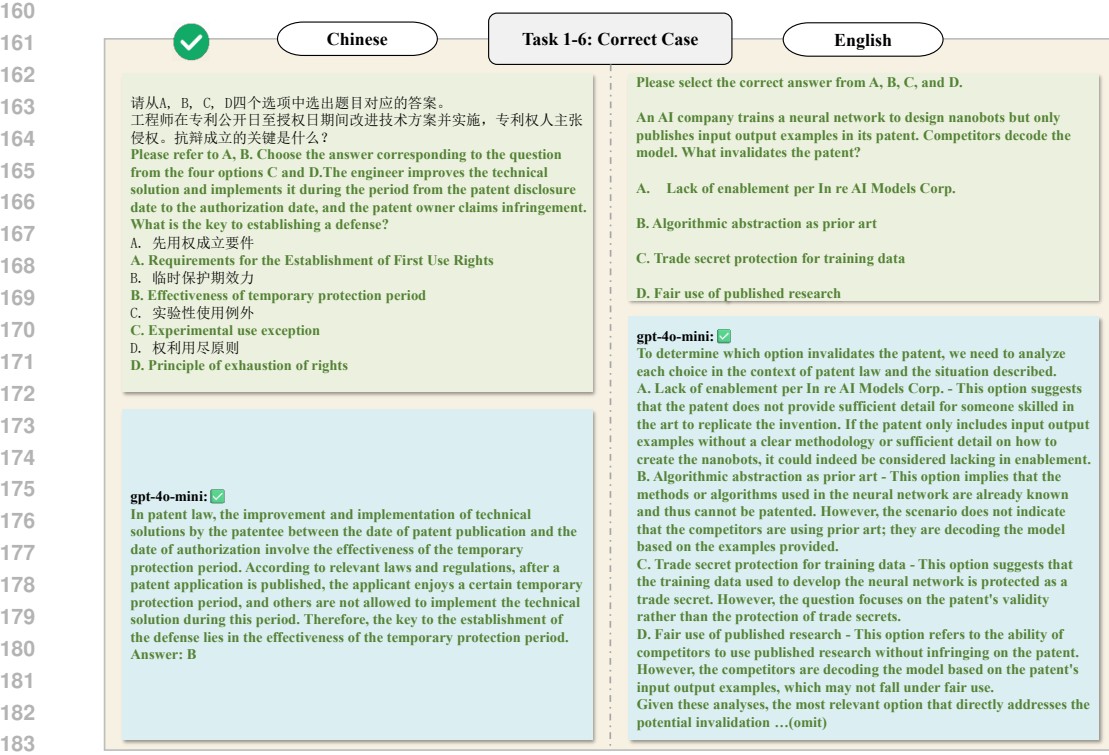

Figure 46: Correct case of task 1-6.

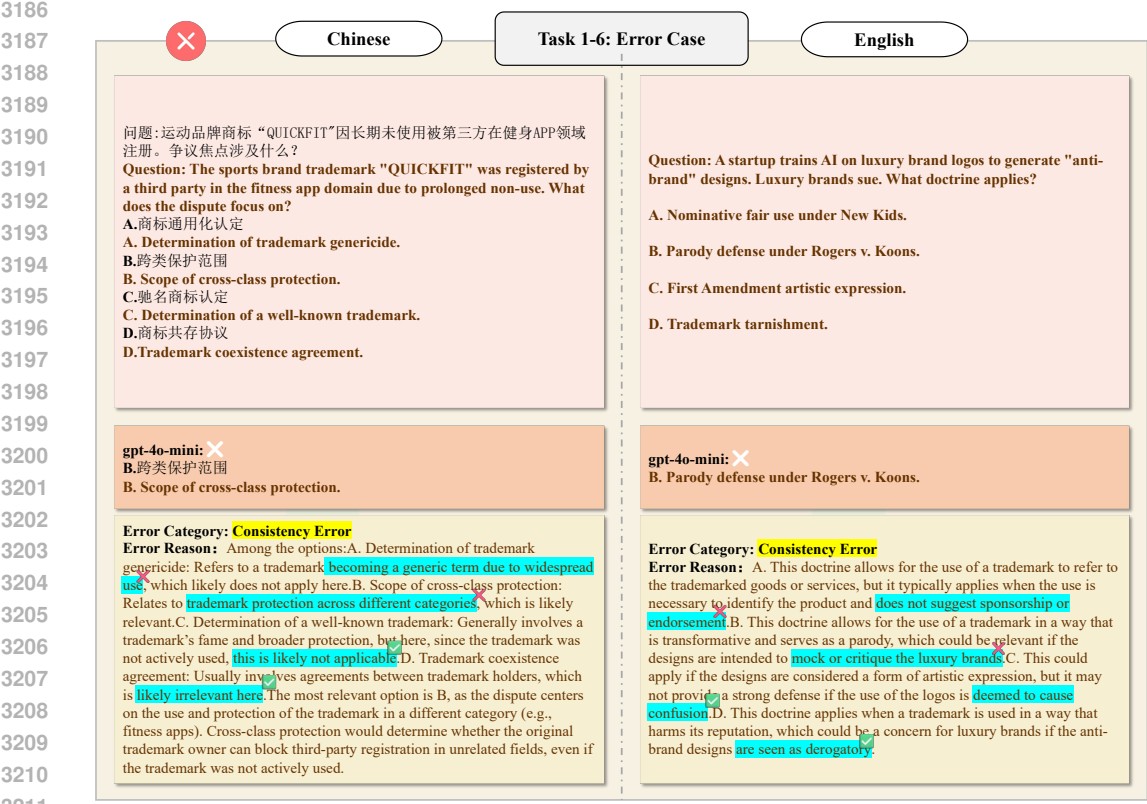

Figure 47: Error case of task 1-6.

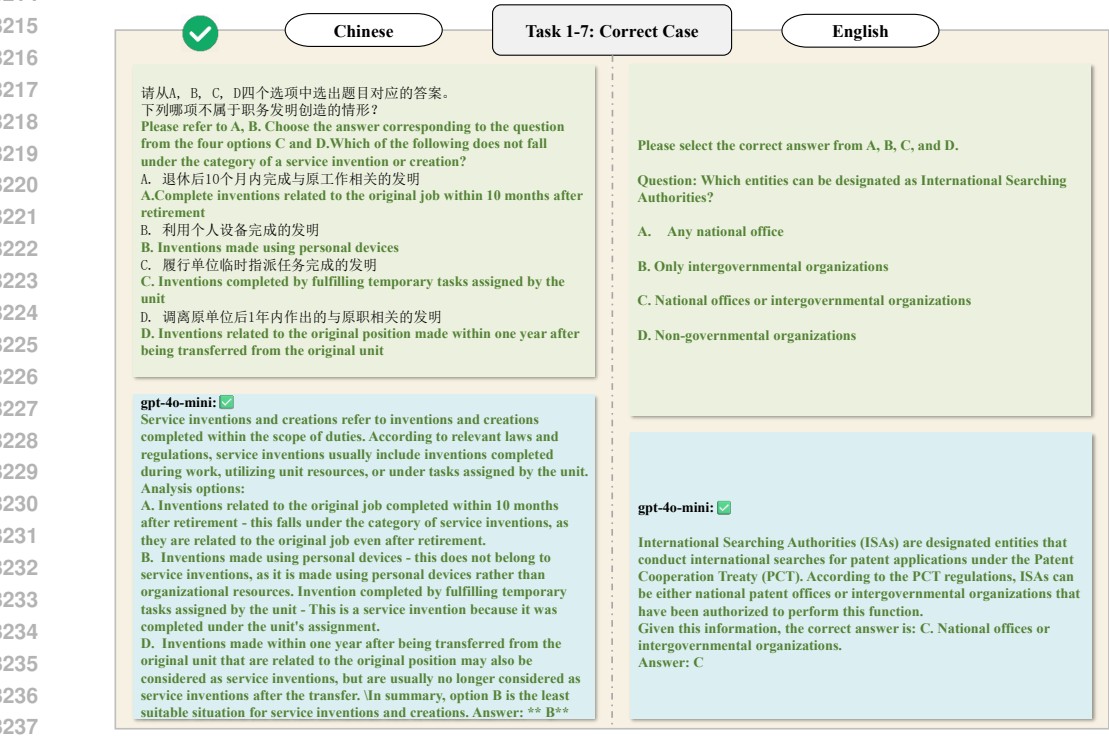

Figure 48: Correct case of task 1-7.

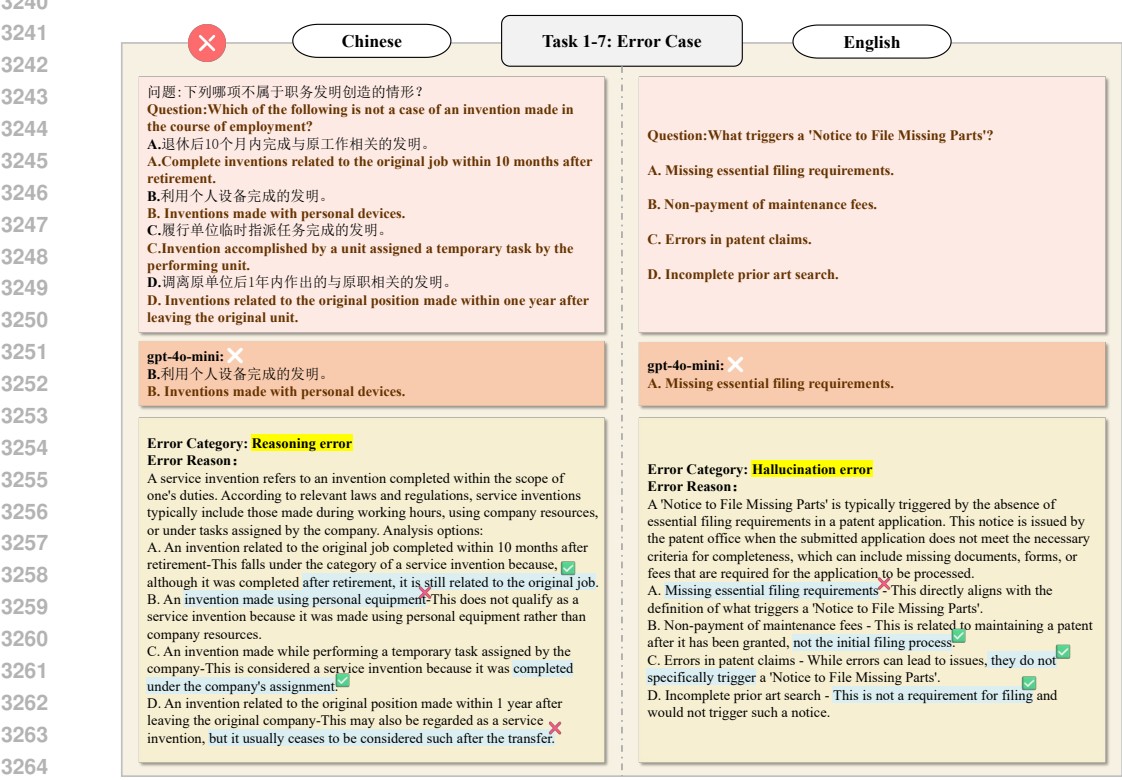

Figure 49: Error case of task 1-7.

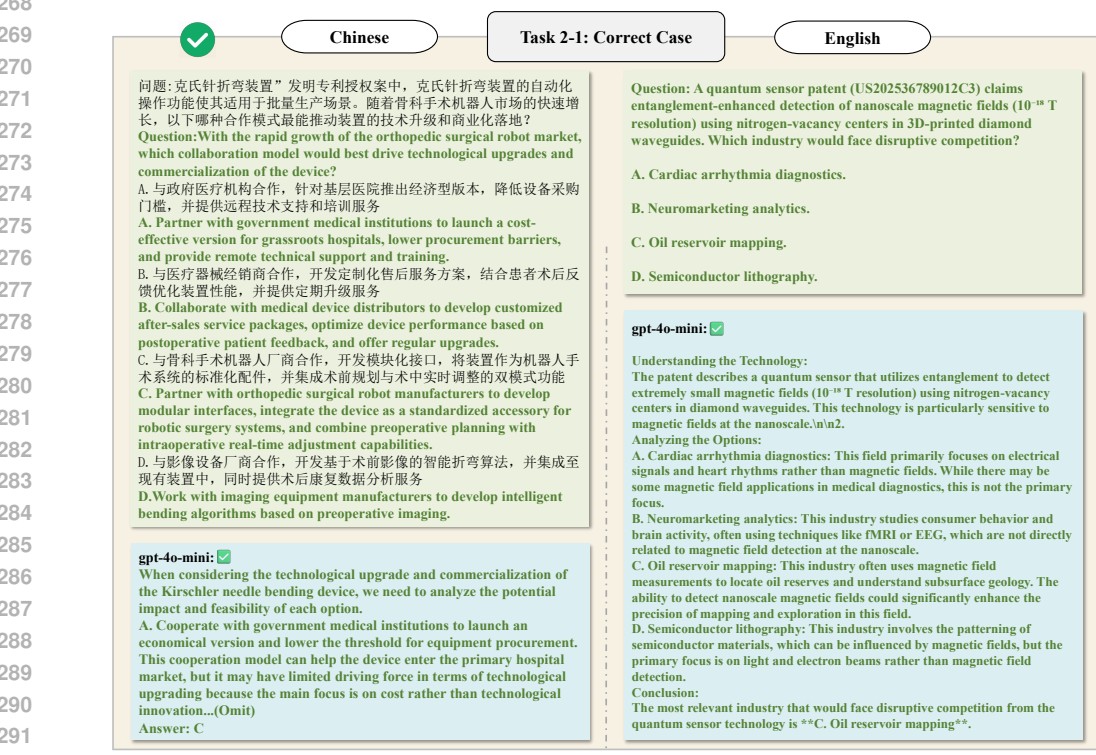

Figure 50: Correct case of task 2-1.

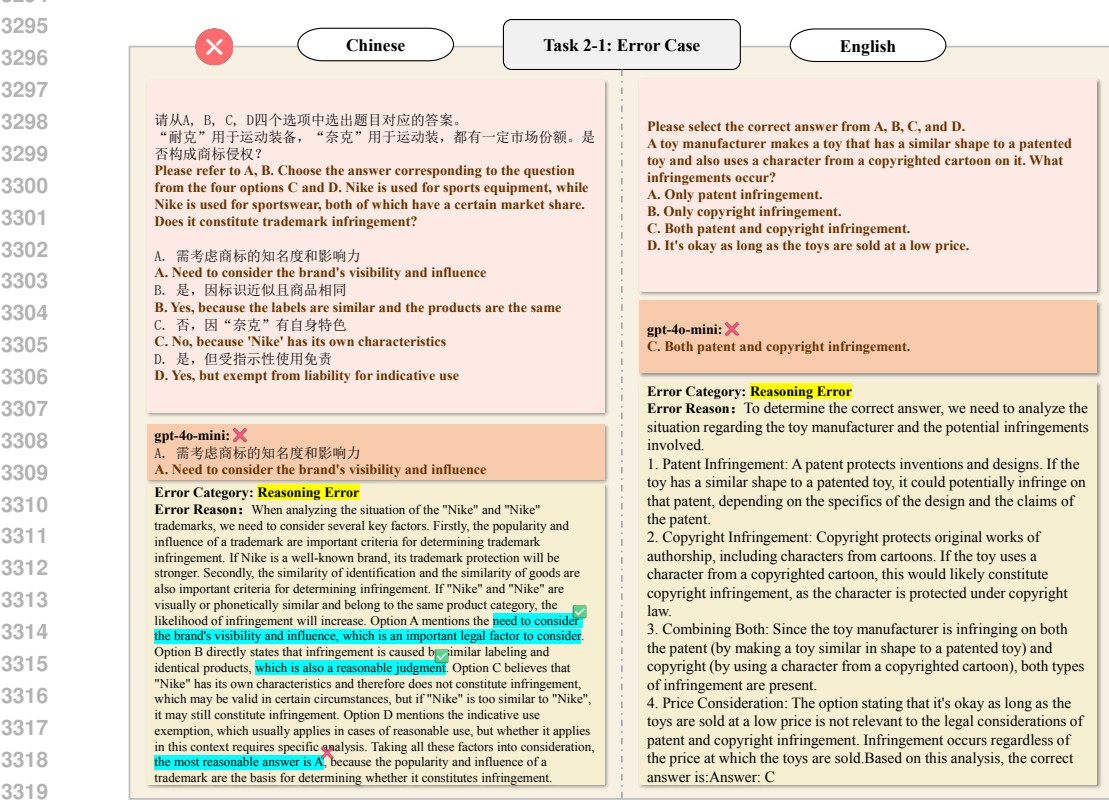

Figure 51: Error case of task 2-1.

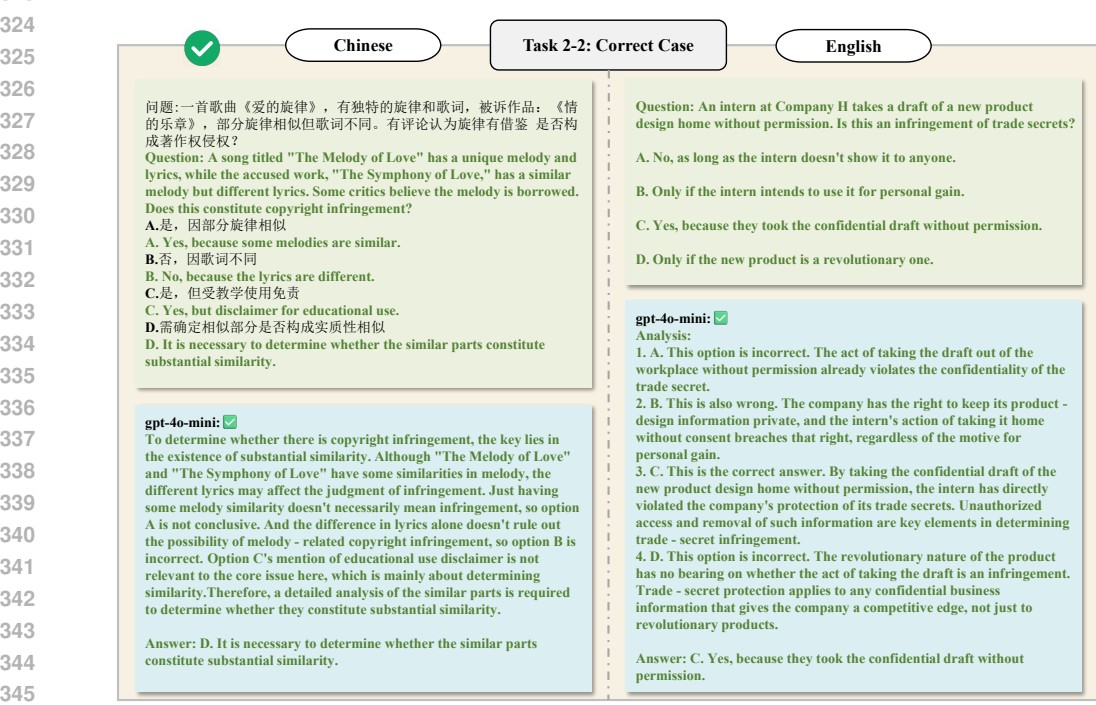

Figure 52: Correct case of task 2-2.

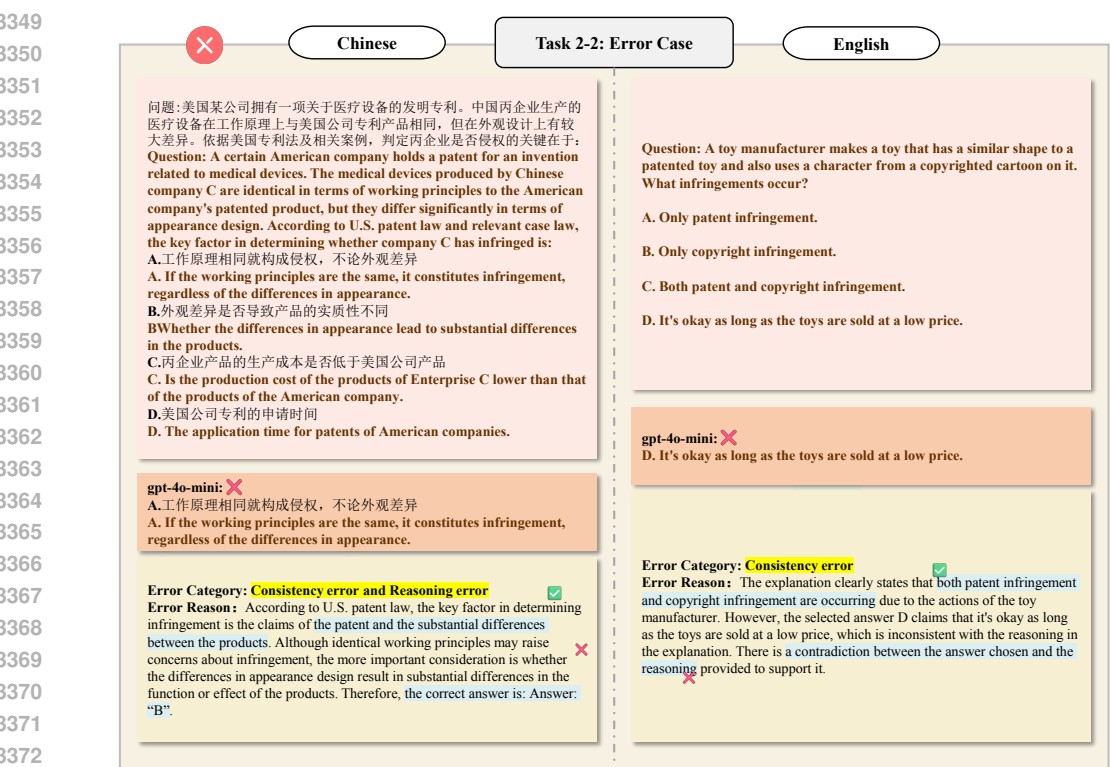

Figure 53: Error case of task 2-2.

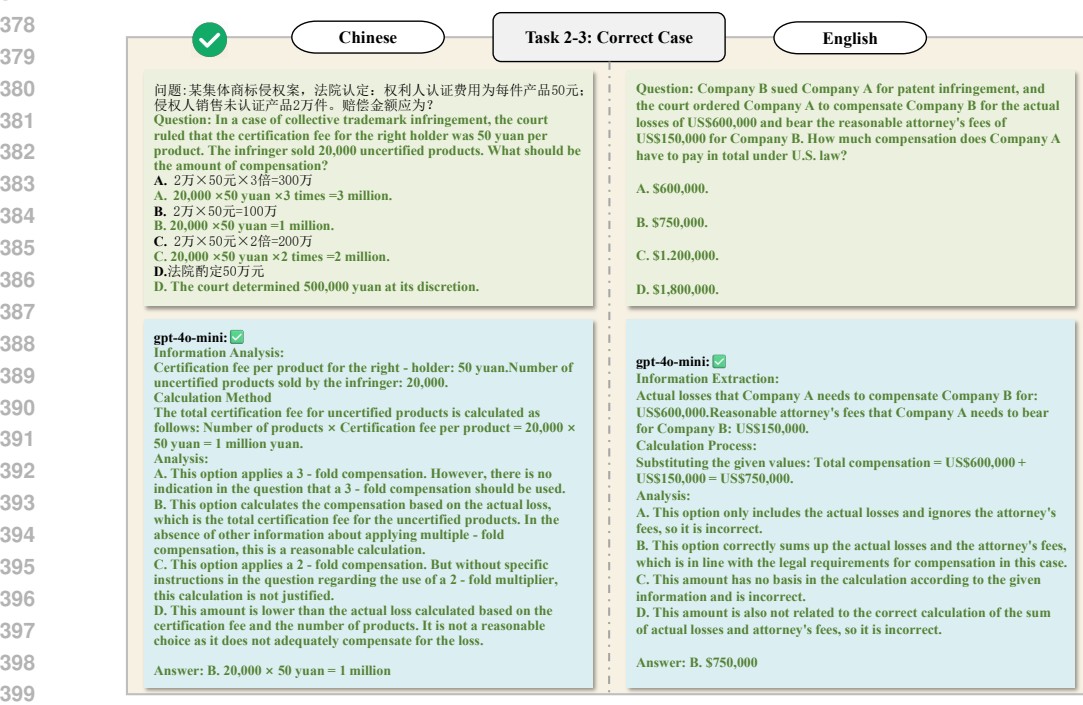

Figure 54: Correct case of task 2-3.

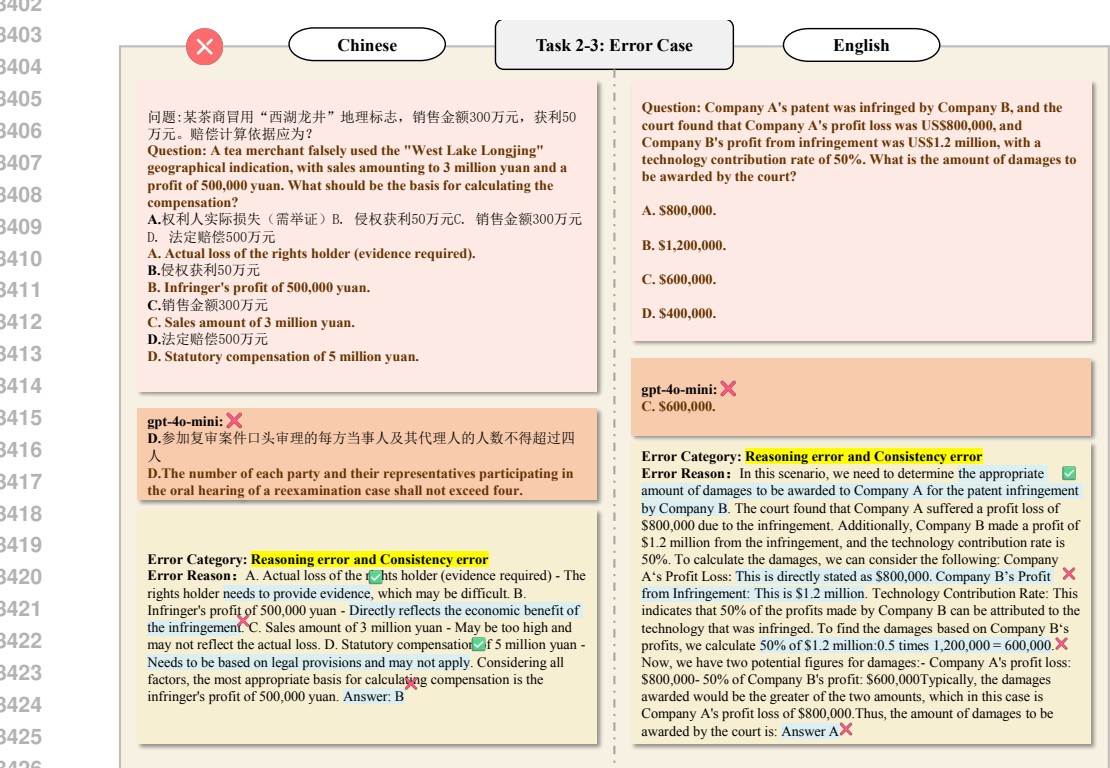

Figure 55: Error case of task 2-3.

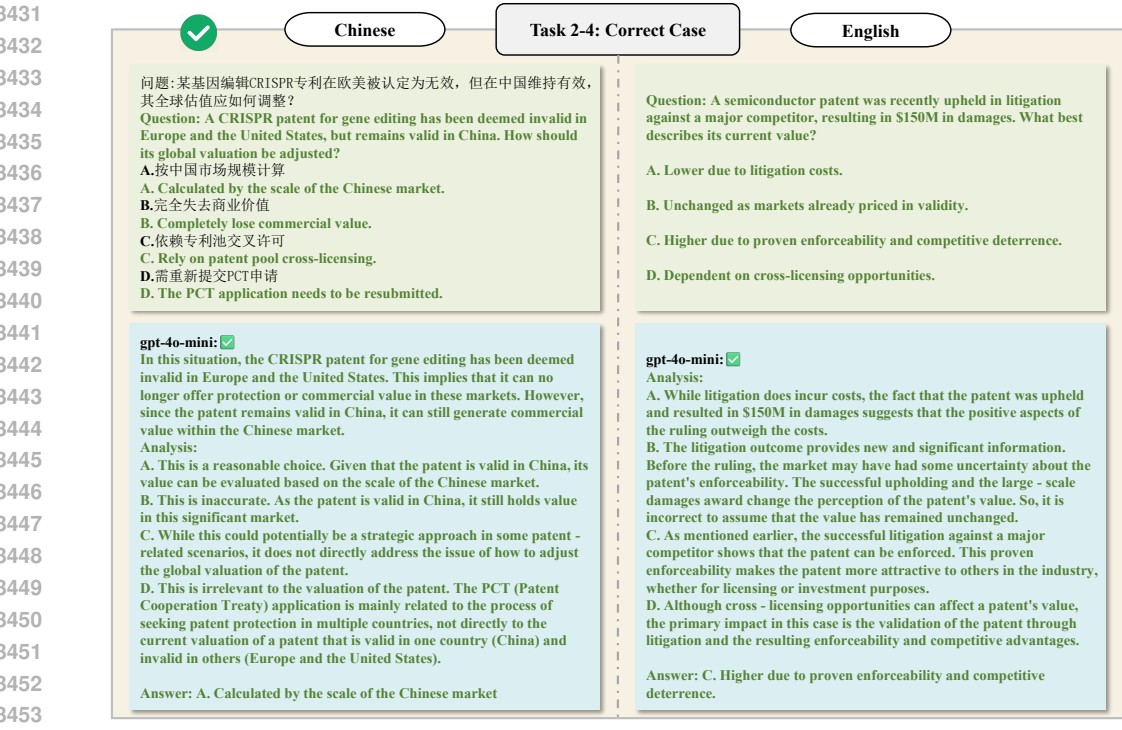

Figure 56: Correct case of task 2-4.

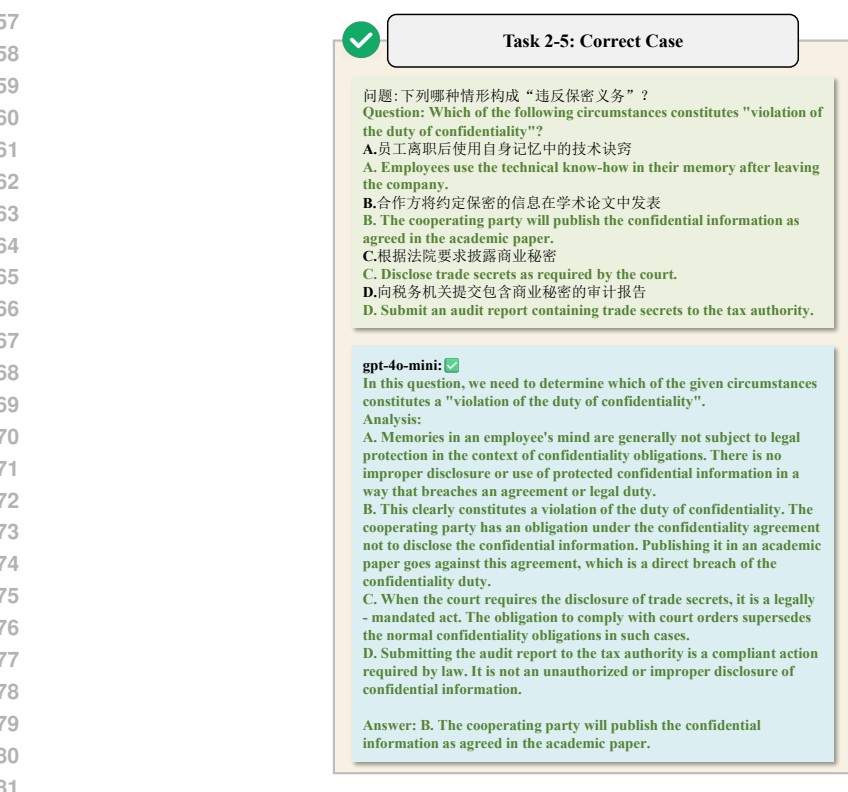

Figure 57: Correct case of task 2-5.

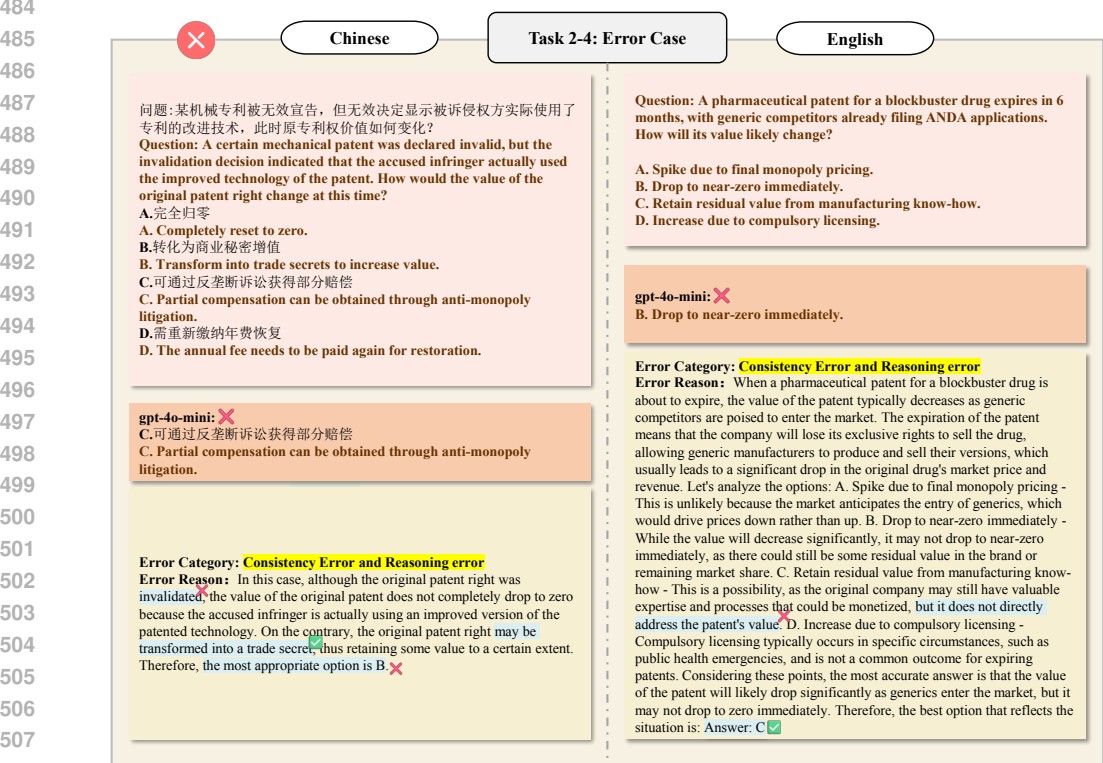

Figure 58: Error case of task 2-4.

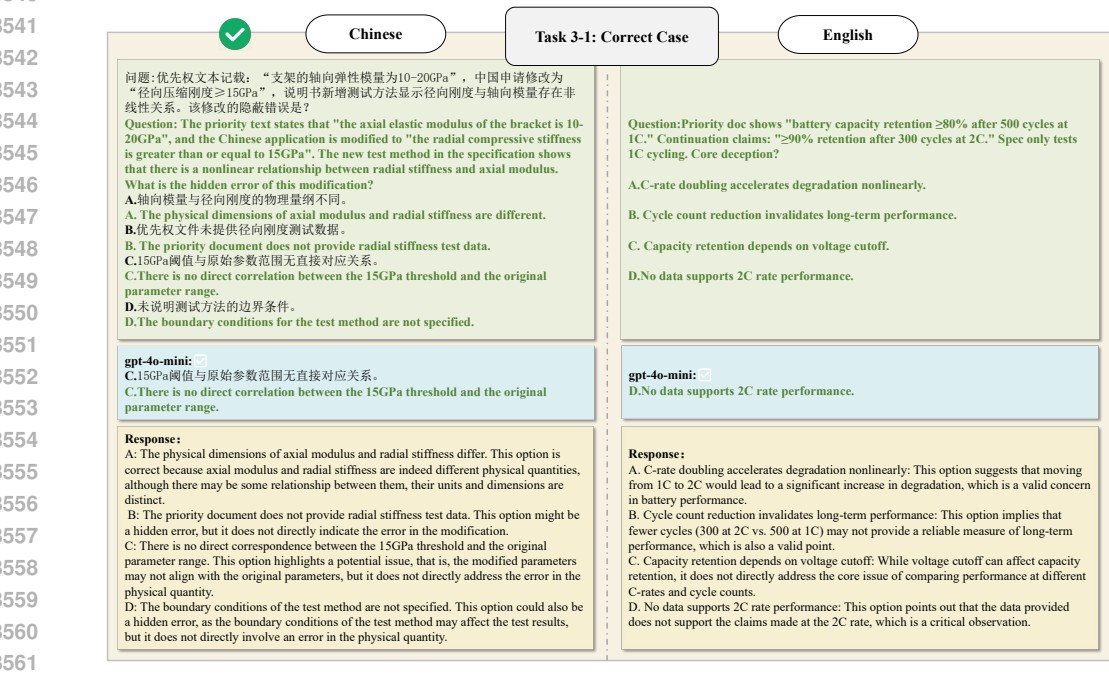

Figure 59: Error case of task 2-5.

Figure 60: Correct case of task 3-1.

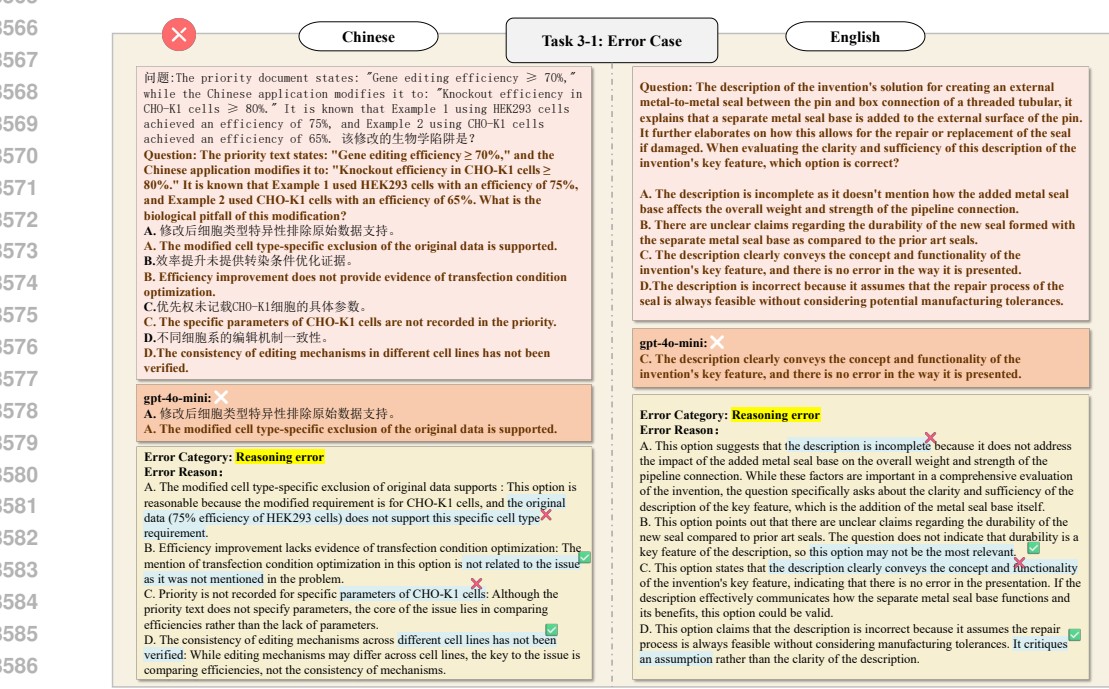

Figure 61: Error case of task 3-1.

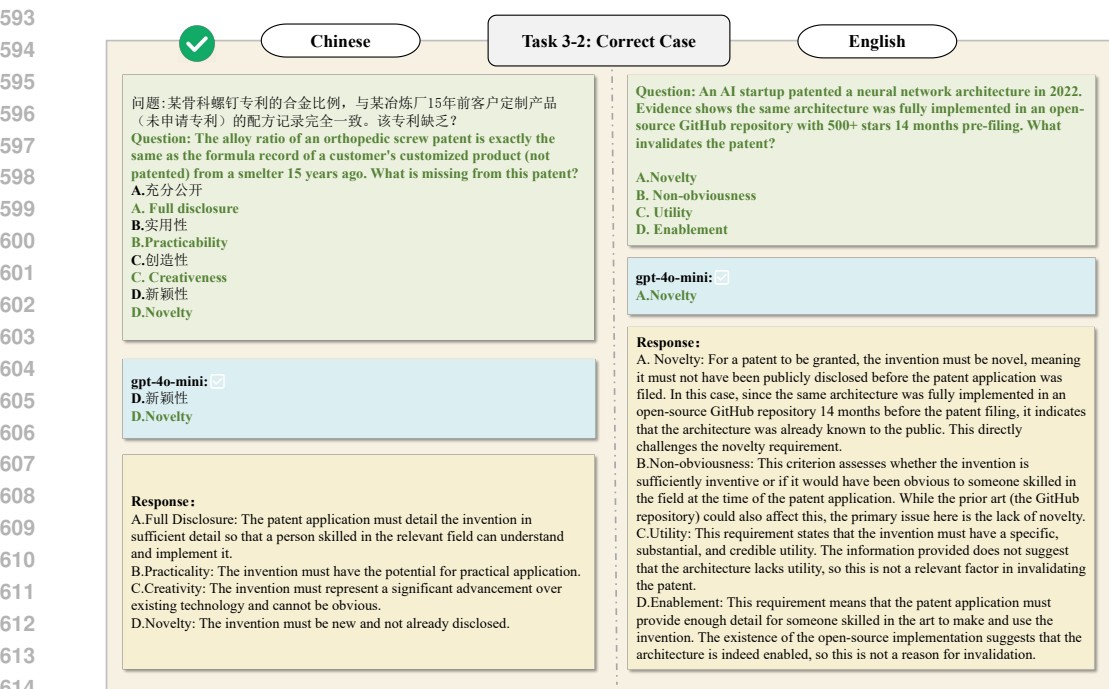

Figure 62: Correct case of task 3-2.

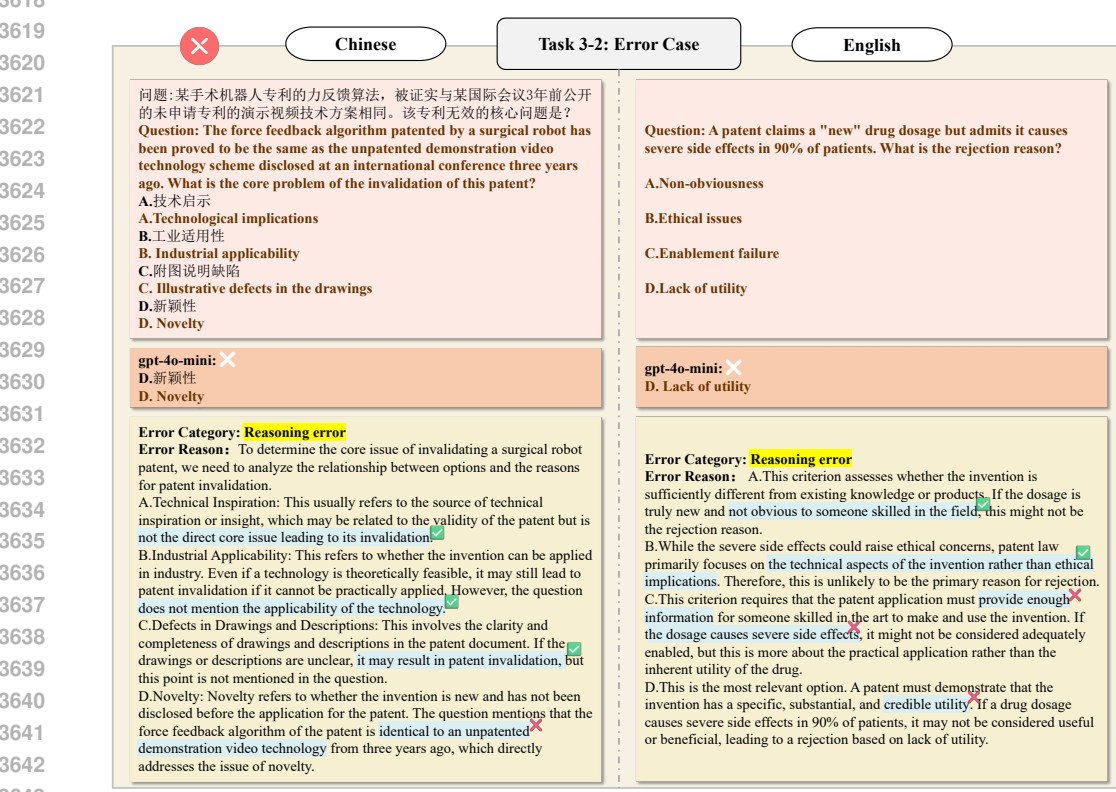

Figure 63: Error case of task 3-2.

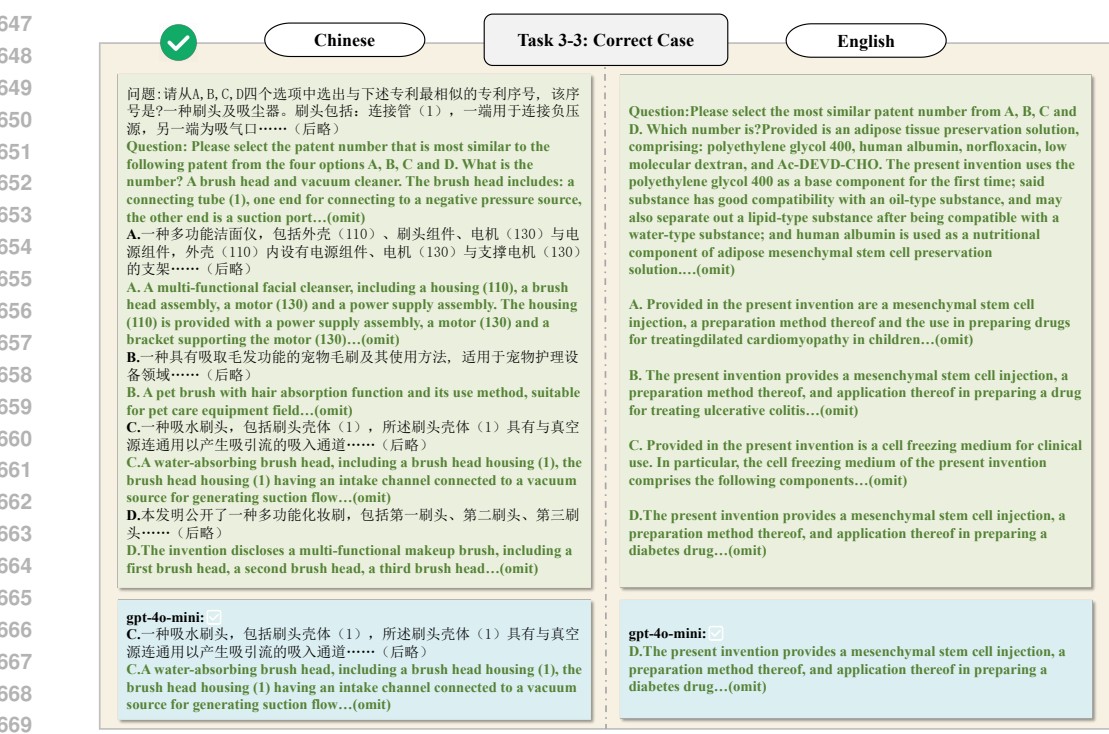

Figure 64: Correct case of task 3-3.

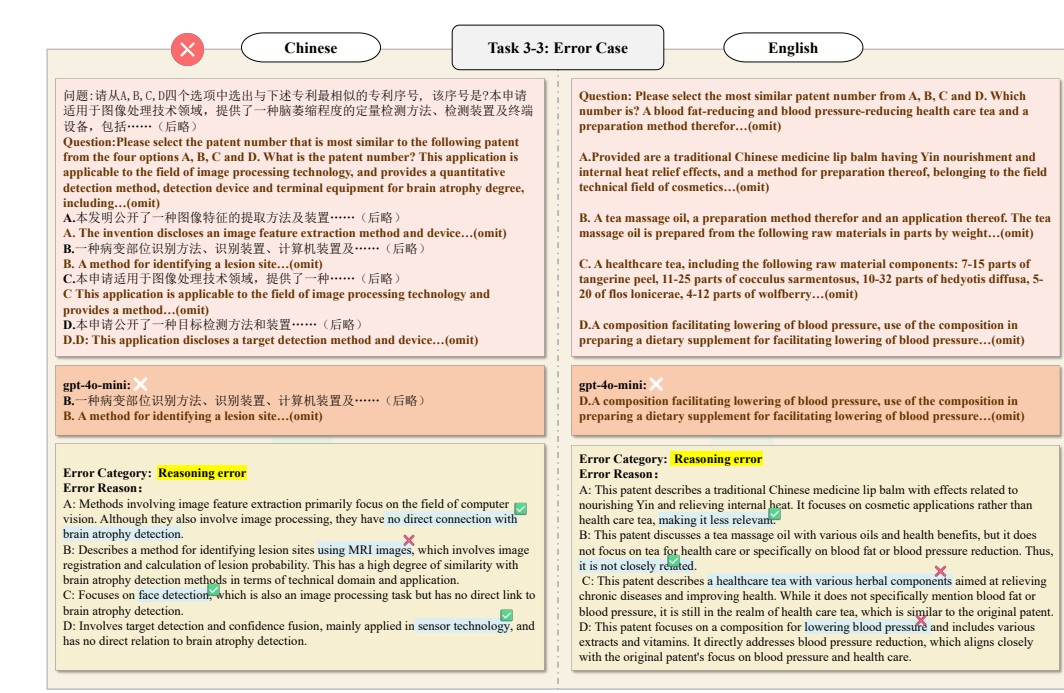

Figure 65: Error case of task 3-3.

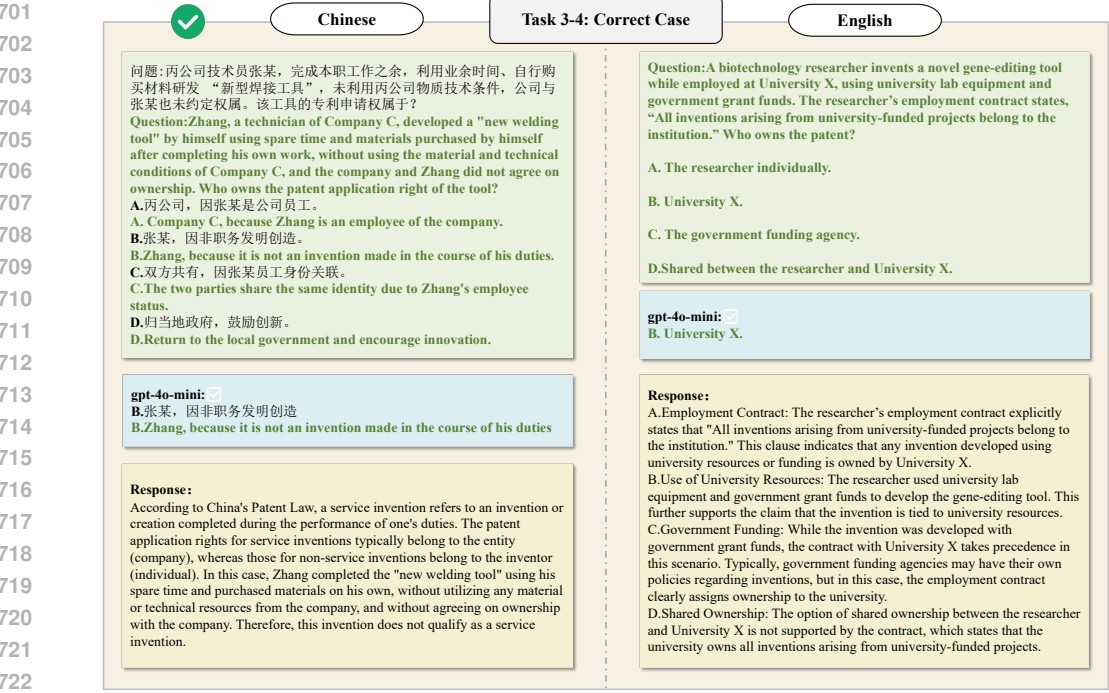

Figure 66: Correct case of task 3-4.

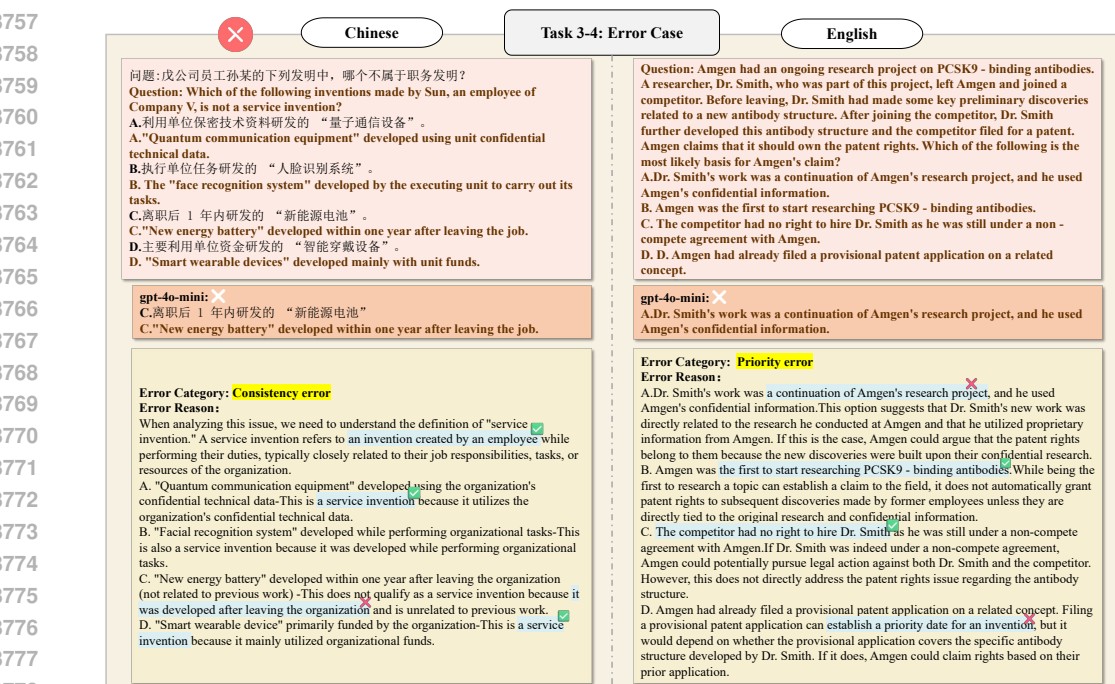

Figure 67: Correct case of task 3-5.

Figure 68: Error case of task 3-4.

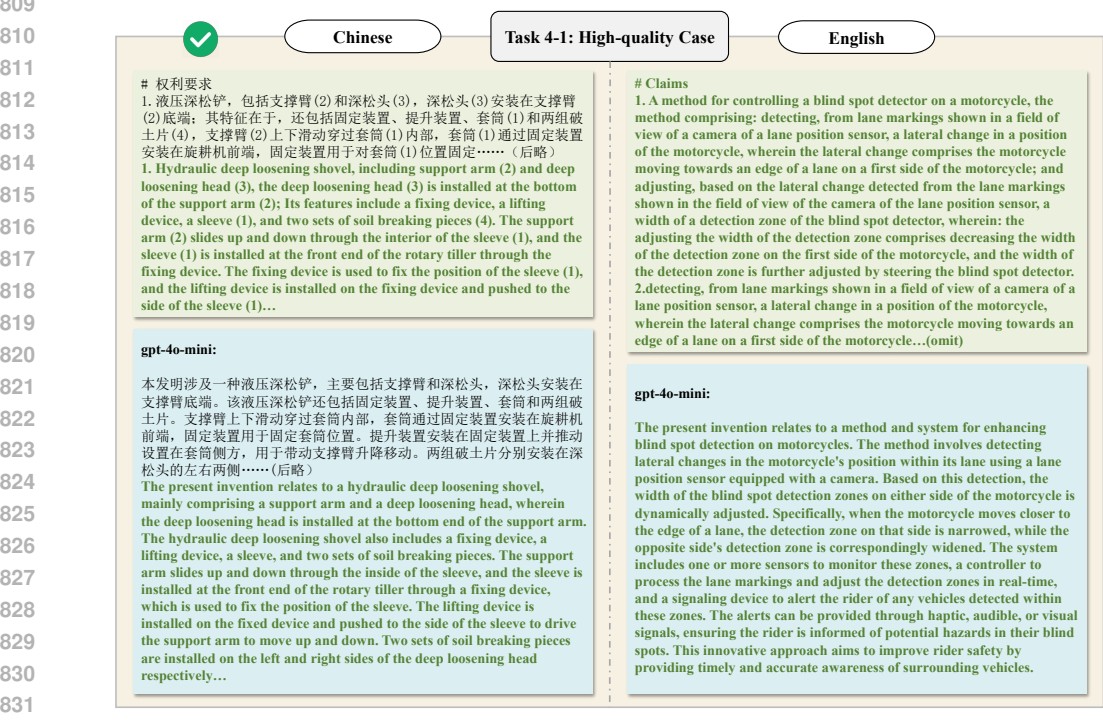

Figure 69: Error case of task 3-5.

Figure 70: High-quality case of task 4-1.

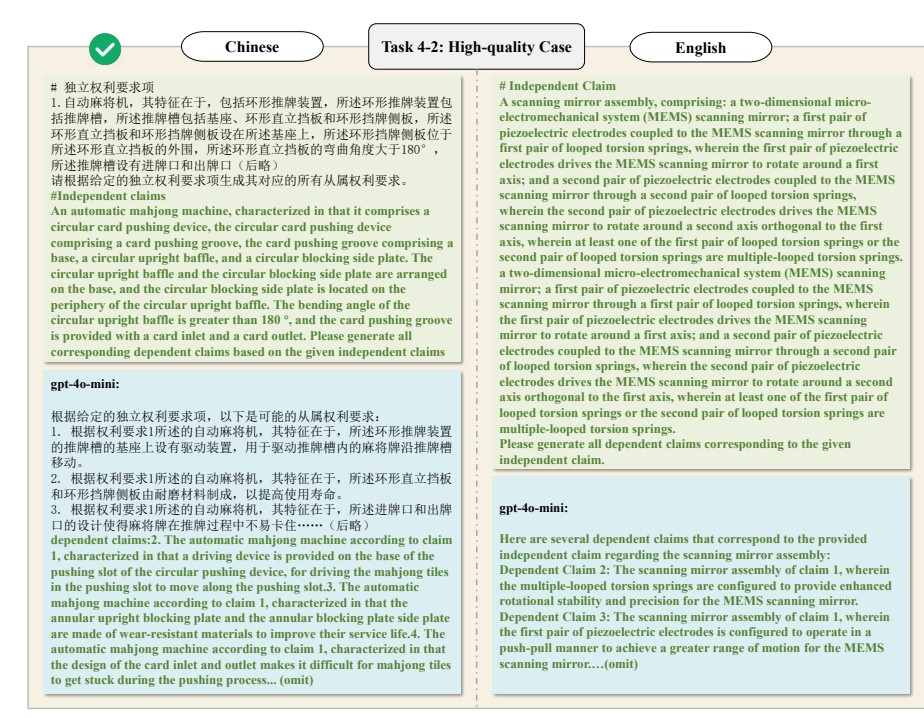

Figure 71: Low-quality case of task 4-1.

Figure 72: High-quality case of task 4-2.

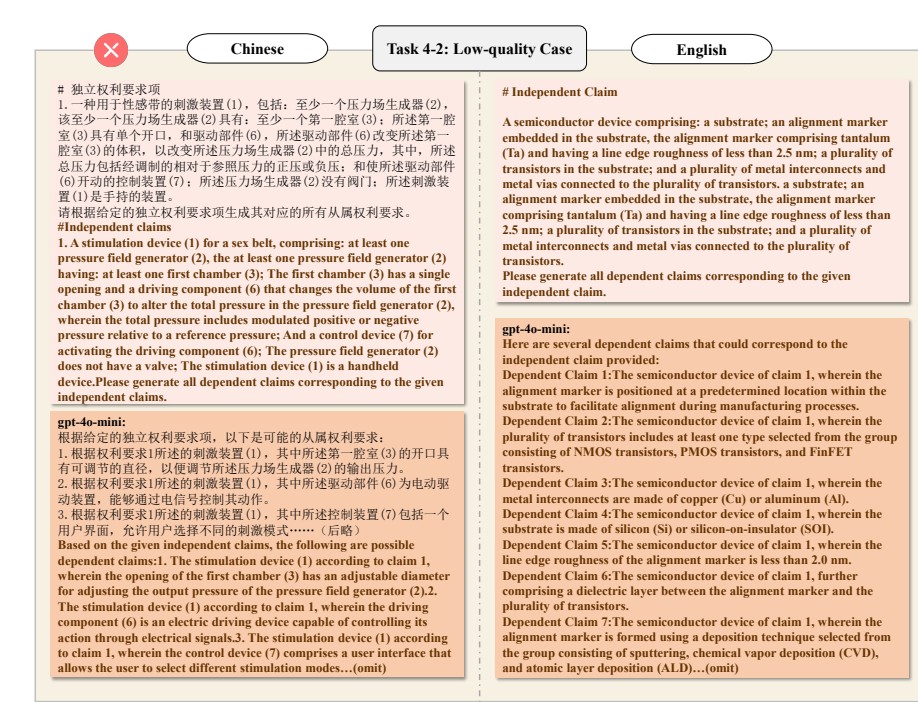

Figure 73: Low-quality case of task 4-2.

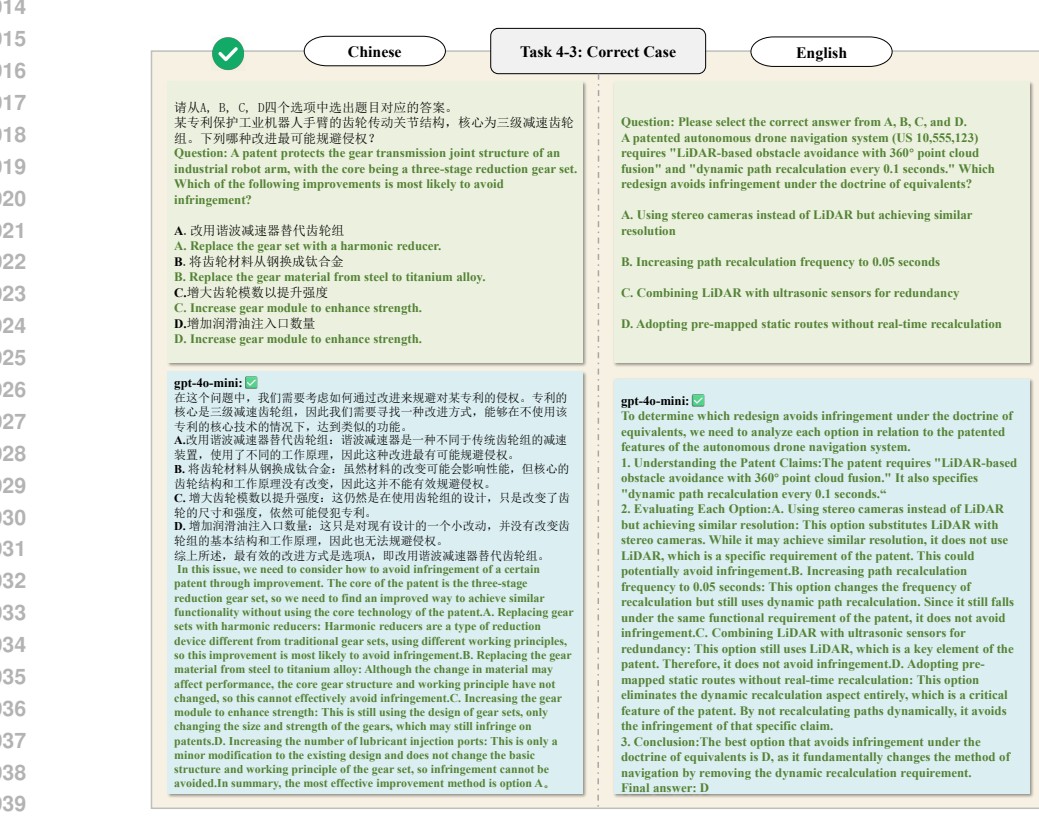

Figure 74: Correct case of task 4-3.

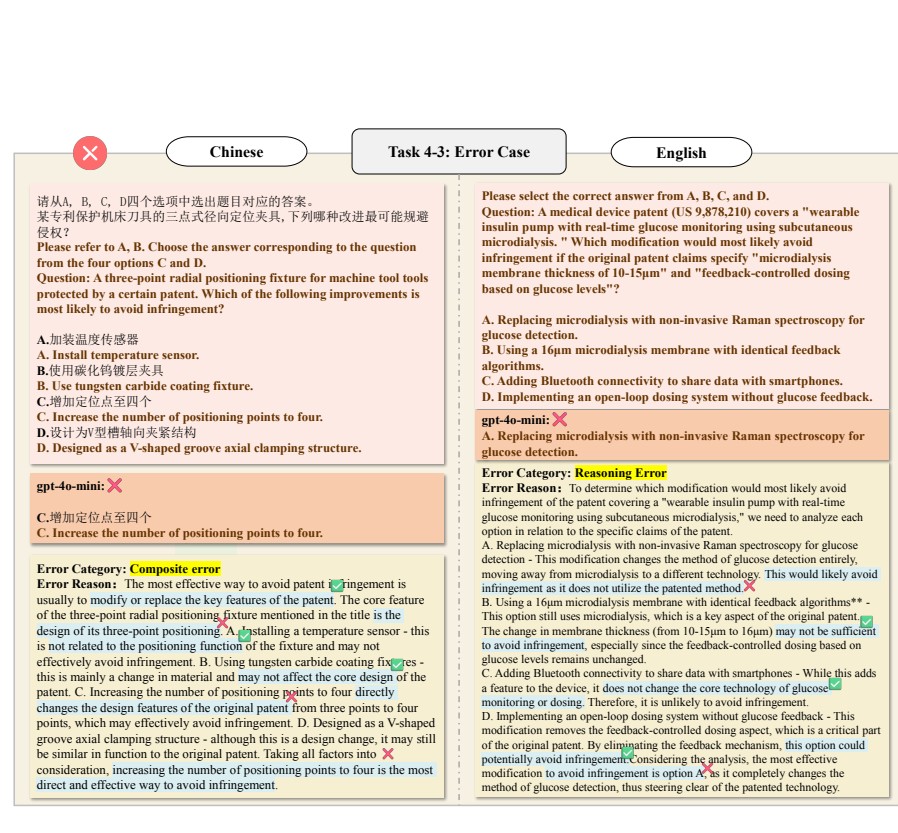

Figure 75: Error case of task 4-3.

