# OpenReview forum: "IPBench: Benchmarking the Knowledge of Large Language Models in Intellectual Property"
_ICLR.cc/2026/Conference — ICLR 2026 Conference Withdrawn Submission_

### Official Review · Reviewer_ZfoF · 2025-10-29

**Soundness:** 2
**Presentation:** 3
**Contribution:** 2
**Rating:** 2
**Confidence:** 4

**Summary:**

This paper proposes the IPBench, which is a large-scale bilingual benchmark for evaluating Large Language Models (LLMs) on real-world Intellectual Property (IP) tasks. IPBench includes 8 IP mechanisms, 20 distinct tasks and corresponding evaluation metrics. Then, this paper evaluates 17 major LLMs on the proposed IPBench. Moreover, this paper claims the release of IPBench to support future research in this specialized field.

**Strengths:**

This paper is well-articulated and clearly presents the construction process and methodological details of the benchmark. Moreover, this paper conducts comprehensive and extensive evaluation across a wide spectrum of models with multiple scales on the benchmark.

**Weaknesses:**

I have some concerns regarding the necessity of some tasks, the potential bias of some tasks, and the efficacy of some evaluation metrics for some tasks.

**Task 1**
I raise several concerns regarding the newly-added tasks in Task 1.​​
- First, the dataset scales for most tasks are limited to 500-600 samples. Given the huge number of clauses in relevant regulations and patent categories, I am concerned that such a restricted sample size may introduce evaluation bias and fail to adequately assess model performance.
- Considering the LLM hallucinations and the fact that RAG technology can now inexpensively retrieve precise legal texts, the utility and necessity of memory-related tasks for evaluating genuine model capabilities appear questionable, especially on such limited test sets.

**Task 2, 3, & 4-3**
- I question the validity of using multiple-choice questions to assess the model's reasoning abilities like this. In the provided task samples (e.g., P47-49), the logical reasoning required appears relatively simplistic and seems to rely primarily on the model's ability to memorize regulation clauses, raising concerns similar to those noted in Task 1 regarding evaluation bias.
- For tasks such as 2-1, it is debatable whether a standardized multiple-choice format is appropriate, as real-world legal analysis often lacks clear-cut answers.

Therefore, I think these tasks of the benchmark does not adequately evaluate the model's reasoning capabilities and achieve its goal.

**Task 4-1 & 4-2**
- I find the evaluation criteria for the two generation tasks to be insufficiently defined. Although GPT-based scoring is employed, the specific rubrics for each score level remain undisclosed. Consequently, while final scores are provided, they offer limited insight into the precise weaknesses leading to a particular rating. Moreover, the relative importance of each scoring dimension and their collective impact on the final assessment are not adequately detailed.

In summary, given the unavoidable hallucinations issue of LLMs and the fact that RAG can cheaply retrieve regulation clauses, I think a meaningful benchmark should prioritize evaluating the model's reasoning capabilities rather than its ability to memorize regulation clauses. However, the tasks in the IPBench rely more on the memory ability, leading to insensibility to evaluate model reason ability. Moreover, the unclear generation task criteria and potential bias further undermine the practical significance​​ of the benchmark. Therefore, I think this work requires further refinement in the current stage, especially for ensuring sufficient sensitivity in evaluating reasoning capabilities.

P.S. The establishment of benchmarks in patent-related tasks is undoubtedly a valuable direction for exploration. However, given that their design significantly influences the trajectory of research progress and, at the current stage, determines which specific capabilities of LLMs are prioritized for evaluation, I still think that benchmark establish should be very cautious. I think the proposed benchmark can evolve into better state with further refinement.

P.S.2 **Reasons of prioritization for reasoning tasks:** In real-world applications involving legal and regulatory contexts, **precise statutory provisions are necessary**. Due to the unpredictable and uncontrollable nature of hallucinations in LLMs, it is impractical to rely solely on a model’s memorization for legal reasoning. Therefore, direct reference to the original legal text remains irreplaceable. Since RAG technology enables low-cost access to relevant legal provisions, the demand for a model’s ability to memorize clauses is relatively low, whereas its capacity for accurate reasoning is considerably more critical. For these reasons, I contend that in benchmark construction, it is more sound to provide the corresponding legal texts together statute-related questions.

**Questions:**

Please see weakness section for details.

---

> ### Author Response · Authors · 2025-11-18
> **Response by Authors (Part 1)**
>
> **Dear Reviewer ZfoF:**
>
> We greatly appreciate these detailed suggestions, which we have treated as valuable guidance to further refine and strengthen our work. In the following pages, we respectfully present several clarifications and key points regarding our study. We kindly request your thoughtful reconsideration of our revised submission.
>
> > **Question 1: Potential bias of some tasks (data size and the necessity of law memory task setting)**
>
> **Answer 1:**
>
> - **Part 1: Limited Data Size.**
>
>    We thank the reviewer for this comment and fully appreciate the concern regarding the coverage of IP-related laws in our dataset. While we acknowledge that the sample size for each task is around 500 instances, we emphasise that this **does not imply limited or narrow legal coverage**. Our Level~1 tasks focus specifically on U.S.\ IP-related law and Chinese IP-related law (including Chinese Patent and Trademark Law). Within this intentionally defined yet comprehensive scope, we have invested substantial expert effort to include representative clauses, statutory provisions, and regulatory elements spanning diverse IP mechanisms.
>
>    As shown in the table below, we report the detailed **IP-related law coverage rates** across both U.S.\ and Chinese jurisdictions. The overall average coverage rate of **92.774\%** demonstrates that our law-memory tasks draw from a broad and representative range of legal sources, thereby substantially reducing the risk of evaluation bias. We will include this table in the appendix of the revised manuscript to further strengthen the transparency and rigor of our work.
>
>     | Country       | Legal Source                                                | Law Coverage |
>     |---------------|-------------------------------------------------------------|--------------|
>     | China         | Patent Law of the PRC                                       | 87.805%      |
>     |               | Trademark Law of the PRC                                    | 91.781%      |
>     |               | Copyright Law of the PRC                                     | 92.537%      |
>     |               | Implementing Regulations of the Patent Law                 | 92.617%      |
>     |               | Regulations on Computer Software Protection                | 93.939%      |
>     |               | Measures for the Protection of Geographical Indications    | 91.667%      |
>     |               | Regulations on the Protection of Layout-Designs of ICs     | 86.111%      |
>     |               | **Within-Country Average**                                   | **91.176%**  |
>     | United States | Patents (35 U.S.C.)                                        | 97.611%      |
>     |               | Lanham Act (15 U.S.C.)                                     | 93.407%      |
>     |               | Copyright Act (17 U.S.C.)                                  | 83.951%      |
>     |               | **Within-Country Average**                                   | **94.409%**  |
>     | **Average Coverage Rate** |                                                           | **92.774%**  |
>
>     Prior benchmarks, such as LexEval [1] and LegalBench [2], which cover a broad range of legal domains, also use roughly 500 instances for their Memorization tasks. Nevertheless, these benchmarks remain professional and highly useful. Additionally, prior work such as LIME [3], which enables more efficient evaluation with fewer samples, and LIMO [4], which has been shown to improve model reasoning ability with only 800 samples. Together, these results demonstrate that the current data size does not constitute a bottleneck.
>
>     We provide a detailed table below summarizing the statistics of representative general and domain-specific benchmarks, with particular emphasis on the data size per task. We will include this table in the appendix of the revised manuscript to further enhance the clarity, completeness, and rigor of our work.
>
>     | Benchmark                | Data Sample Size per Task | Domain    |
>     |--------------------------|--------------------------|-----------|
>     | GPQA [9]                 | 182/Discipline           | General   |
>     | SuperGPQA [10]           | 93/Discipline            | General   |
>     | LexEval [1]              | 615                      | Law       |
>     | LegalBench [2]           | 500-600                  | Law       |
>     | E-Eval [6]               | 399                      | Education |
>     | PatentEval [7]           | 200                      | IP        |
>     | MoZIP [8]                | 267                      | IP        |
>     | IPBench (**_Ours_**)     | 500                      | IP        |

---

> ### Author Response · Authors · 2025-11-18
> **Response by Authors (Part 2)**
>
> **Dear Reviewer ZfoF:**
>
> - **Part 2: The necessity of law memory task.**
>
>     We thank the reviewer for the detailed consideration of our legal memory task design, which will help us refine the manuscript to more clearly highlight the importance of memory-oriented evaluations in the IP domain. Our taxonomy is grounded in Webb’s Depth of Knowledge (DOK) framework, a well-established cognitive theory in which the first level of knowledge corresponds to the **recall and reproduction** of concrete facts, definitions, and foundational legal concepts.
>
>     As noted in **lines 161--162**, we consider the IP domain a technical-legal hybrid field. Motivated by these considerations, and inspired by legal-specific benchmarks such as LexEval [1] and LawBench [5], we incorporate legal memory tasks into our taxonomy to provide a more complete and faithful evaluation of LLMs in the IP domain, rather than focusing solely on higher-level reasoning. In particular, Tasks 1-3 require models not only to recall the relevant legal provisions, but also to distinguish between different law versions—an ability that is fundamental for accurate technical-legal decision-making.
>
>     We appreciate the reviewer’s thoughtful suggestion regarding the potential use of a RAG system for clause retrieval. While valuable, this reflects a **different perspective from the primary objective of our benchmark**. The core function of a RAG system is to ***retrieve existing provisions***, whereas the legal memory tasks in IPBench are designed to evaluate ***the model’s native knowledge accuracy and its ability to differentiate between law versions***. This capability is fundamental in many real-world IP scenarios—such as preliminary patent examination without access to retrieval tools or assessing the applicability of legal provisions across different versions—and therefore cannot be substituted by a retrieval-based system.
>
> Overall, we sincerely appreciate the reviewer’s careful consideration of our memory task design. While we value the suggestion, we continue to view the memory component as an essential element of our taxonomy, which is grounded in Webb’s DOK cognitive framework. We will integrate this clarification into the revised manuscript to further strengthen the clarity, motivation, and rigor of our task design.
>
> > **Question 2: Efficacy of some evaluation metrics for some tasks (especially, MCQA for reasoning task)**
>
> **Answer 2:**
>
> Thank you for this insightful comment. We agree that the single-answer MCQ format has certain limitations. We adopted this format primarily to ensure **objective**, **scalable**, and most importantly, **accurate** evaluation. This design choice helps avoid the ambiguity and substantial cost associated with parsing and grading open-ended responses, thereby supporting evaluation **consistency**, as also noted by the reviewer in Question 3.
>
> We are grateful to the reviewer for carefully examining our provided large set of cases, and we believe there may have been a slight misunderstanding. As discussed in Question 1, we view the IP domain as a technical-legal hybrid field. Consequently, when constructing the Level-2 Logical Reasoning tasks, our priority was to select questions grounded in real IP-related cases. In the examples provided, these tasks consistently integrate technical descriptions with the need to reason under the relevant legal provisions, thereby **emphasizing applied reasoning rather than simple memorization of clauses**. We will clarify this motivation more explicitly in the Evaluation Setup section and consider the development of an open-ended version as a meaningful direction for future work.

---

> ### Author Response · Authors · 2025-11-18
> **Response by Authors (Part 3)**
>
> **Dear Reviewer ZfoF:**
>
> > **Question 3: More details about evaluation criteria for the two generation tasks**
>
> **Answer 3:**
>
> Thank you for this helpful suggestion. We acknowledge that the specific criteria for each LLMScore dimension were not fully detailed in the current version, as we initially prioritized concise and clear presentation. Due to space constraints, we provide here a brief version of the LLMScore rubric for Task 4-2, and we will include the complete, fully articulated version in the revised manuscript to enhance transparency and evaluative rigor.
>
> **Brief Version of Prompt**
>
> ---
> You are an experienced intellectual property expert specializing in assessing the quality of patent dependent claims. Please objectively evaluate the dependent claims written by the AI assistant, acting as a fair and rigorous judge. When evaluating, you should score it based on the following five dimensions:
> 1. **Accuracy** for the task of generating dependent claims based on a given independent claim evaluates the linguistic correctness, structural fidelity, and legal coherence of the generated claims. It emphasizes the following key aspects: (Omit)
> 2. **Relevance** for the task of generating dependent claims based on a given independent claim evaluates whether the content of the dependent claims remains aligned with the technical subject matter disclosed in the independent claim and the overall invention. This ensures that the claim set does not deviate from the invention’s disclosed scope. This metric specifically emphasizes: (Omit)
> 3. **Completeness** for the task of generating dependent claims based on a given independent claim assesses whether the dependent claims meaningfully expand upon the independent claim by introducing additional technical features or specific limitations. It ensures that each claim contributes to a more comprehensive protection of the invention. This metric focuses on the following key aspects: (Omit)
> 4. **Consistency** for the task of generating dependent claims based on a given independent claim assesses whether the generated claims logically and legally align with their referenced claims and maintain internal coherence within the claim set. It includes the following key evaluation aspects: (Omit)
> 5. **Language-Style** for the task of generating dependent claims based on a given independent claim evaluates the linguistic quality and drafting style of the claims, focusing on clarity, precision, and conciseness in accordance with standard patent drafting conventions. This includes the following key evaluation aspects: (Omit)
>
> We will provide the following materials: the patent independent claims, the ground-truth dependent claims and the dependent claims written by the AI assistant based on independent claims. When starting your evaluation, you need to follow the reasoning steps below:
> 1. Compare the AI assistant’s dependent claims with the ground-truth dependent claims, pointing out the shortcomings of the AI assistant’s answer and explaining them in detail.
> 2. Evaluate the AI assistant’s dependent claims according to the dimensions mentioned above, giving a score from 1 to 10 for each dimension.
> 3. Based on the scores for each dimension, calculate the overall score for the dependent claims written by the AI assistant (1-10 points).
> 4. Your scoring should be as strict as possible, and you must follow the scoring rules below: The higher the quality of the response, the higher the score.
>
> **Scoring Standards for Dependent Claim Generation:**
> (Omit)
>
> ---
>
> Our additional experimental results on the correlation between LLMScore and human judgment in **Table 6 (Page 8)** show that LLMScore achieves higher consistency with human evaluations and lower p-values, indicating a more reliable and stable automated assessment. Furthermore, we have provided detailed results across the different LLM-as-a-judge dimensions **in Appendix K.4** of the initial version, which help illustrate the model’s fine-grained generation weaknesses on these tasks.
>
> We thank the reviewer for the positive feedback regarding our benchmarking of patent-related tasks. Our IPBench is grounded in Webb’s DOK cognitive theory, whose taxonomy offers a structured framework for evaluating abilities ranging from factual recall to higher-level reasoning. We kindly request the reviewer’s thoughtful reconsideration of our revised submission.
>
> We look forward to your response and sincerely appreciate the time and effort devoted to evaluating our work.

---

> ### Author Response · Authors · 2025-11-18
> **Response by Authors (Part 4: Reference)**
>
> **Reference**
>
> [1] Li H, Chen Y, Ai Q, et al. Lexeval: A comprehensive chinese legal benchmark for evaluating large language models[J]. Advances in Neural Information Processing Systems, 2024, 37: 25061-25094.
>
> [2] Guha N, Nyarko J, Ho D, et al. Legalbench: A collaboratively built benchmark for measuring legal reasoning in large language models[J]. Advances in neural information processing systems, 2023, 36: 44123-44279.
>
> [3] Zhu K, Zang Q, Jia S, et al. Lime: Less is more for mllm evaluation[C]//Findings of the Association for Computational Linguistics: ACL 2025. 2025: 9086-9121.
>
> [4] Ye Y, Huang Z, Xiao Y, et al. LIMO: Less is More for Reasoning[C]//Proceedings of the Second Conference on Language Modeling. 2025.
>
> [5] Fei Z, Shen X, Zhu D, et al. Lawbench: Benchmarking legal knowledge of large language models[C]//Proceedings of the 2024 conference on empirical methods in natural language processing. 2024: 7933-7962.
>
> [6] Hou J, Ao C, Wu H, et al. E-EVAL: A Comprehensive Chinese K-12 Education Evaluation Benchmark for Large Language Models[C]//Findings of the Association for Computational Linguistics ACL 2024. 2024: 7753-7774.
>
> [7] Zuo Y, Gerdes K, Clergerie É, et al. PatentEval: Understanding errors in patent generation[C]//Proceedings of the 2024 Conference of the North American Chapter of the Association for Computational Linguistics: Human Language Technologies (Volume 1: Long Papers). 2024: 2687-2710.
>
> [8] Ni S, Tan M, Bai Y, et al. MoZIP: A Multilingual Benchmark to Evaluate Large Language Models in Intellectual Property[C]//Proceedings of the 2024 Joint International Conference on Computational Linguistics, Language Resources and Evaluation (LREC-COLING 2024). 2024: 11658-11668.
>
> [9] Rein D, Hou B L, Stickland A C, et al. Gpqa: A graduate-level google-proof q\&a benchmark[C]//First Conference on Language Modeling. 2024.
>
> [10] Du X, Yao Y, Ma K, et al. SuperGPQA: Scaling LLM Evaluation across 285 Graduate Disciplines[C]//The Thirty-ninth Annual Conference on Neural Information Processing Systems Datasets and Benchmarks Track. 2025.
>
> ---
>
> We kindly request your thoughtful reconsideration of our submission, and we look forward to your response.

---

> ### Author Response · Authors · 2025-11-26
> **Sincerely Looking Forward to Your Response**
>
> **Dear Reviewer ZfoF**,
>
> We hope this message finds you well. We understand that you may have a busy schedule, so we would like to kindly remind you about our responses to your comments.
>
> We sincerely appreciate the time and effort you have taken to provide valuable feedback on our manuscript. We deeply value your feedback and believe that our responses address the concerns you raised. If there are any additional questions or clarifications needed, we would be more than happy to provide further details promptly.
>
> If you find our clarifications satisfactory, we kindly request your thoughtful reconsideration of our submission, and we look forward to your response.
>
> Best Regards,
>
> Authors of #6607

---

> ### Author Response · Authors · 2025-11-28
> **Gentle Reminder. Discussion Period Ending Soon.**
>
> **Dear Reviewer ZfoF**,
>
> I hope this message finds you well. As the discussion period is nearing its end, with **less than one week remaining**, we would like to ensure that we have addressed all your concerns satisfactorily. If there are any additional points or feedback you would like us to consider, please let us know. Your feedback is invaluable to us and greatly helps improve our work.
>
> If you find our clarifications satisfactory, we kindly request your thoughtful reconsideration of our submission, and we look forward to your response.
>
> Best Regards,
>
> Authors of #6607

---

### Official Review · Reviewer_EVqu · 2025-10-31

**Soundness:** 2
**Presentation:** 3
**Contribution:** 3
**Rating:** 6
**Confidence:** 4

**Summary:**

This paper presents IPBench, the first large-scale, bilingual (Chinese–English) benchmark designed to evaluate Large Language Models (LLMs) on real-world Intellectual Property (IP) tasks beyond patents, including trademarks, copyrights, and trade secrets. The benchmark is notable for its breadth (20 tasks across 8 IP mechanisms), depth (cognitive complexity from information recall to creative generation), and linguistic diversity (Chinese and English). The authors evaluate 17 models under multiple prompting conditions and find that general-purpose closed-source models significantly outperform specialized legal/IP models. They also introduce LLMScore, a structured LLM-as-judge metric for evaluating generative outputs, which is a thoughtful and practical contribution. Overall, this work establishes a strong foundation for future research on legal reasoning, domain-specific LLMs, and multimodal IP understanding.

**Strengths:**

- This paper presents a novel contribution to the evaluation of LLMs in the legal and IP domain. Its originality lies in two aspects: IPBench is the first benchmark to systematically evaluate LLMs across a broad range of real-world IP mechanisms, including not just patents, but also trademarks, copyrights, trade secrets. In addition, the authors introduce a cognitive-complexity-based taxonomy of tasks (Depth of Knowledge framework), which offers a structured and generalizable way to assess model capabilities.

- The benchmark is also carefully designed, with 20 tasks covering diverse formats and jurisdictions, and evaluated across 17 strong LLM baselines. The "LLMScore" is a thoughtful methodological contribution that addresses known weaknesses in automatic metrics.

- The paper also demonstrates strong clarity in its writing, task definitions, and experimental analysis. Figures and tables are well-organized, and the empirical findings and the limited impact of CoT prompting are clearly presented.

- Finally, the paper fills a notable gap in LLM evaluation benchmarks by addressing a domain with high societal, legal, and economic relevance.

**Weaknesses:**

- First, although 17 models are evaluated in the paper, including general-purpose, legal-specialized, and IP-specific variants, it offers little rationale for why these particular models were included or excluded. Some well-known legal-domain baselines such as Legal-BERT or CaseLaw-BERT are absent, without discussion. Moreover, the prompting strategy is applied uniformly across all tasks. While this ensures consistency within model types, it overlooks the high diversity of IPBench tasks, which range from factual recall to creative generation. For example, CoT prompting is used even on simple memory-based tasks, where it is shown to slightly degrade performance, and few-shot examples are randomly sampled without semantic relevance to the current task. Although the authors acknowledge that CoT may conflict with model preferences on certain tasks, they do not analyze CoT effectiveness across task types or cognitive complexity levels.

- Second, the related work section primarily focuses on patent-specific datasets, but omits discussion of broader legal benchmarks like LexGLUE, LegalBench, or LawBench. Including such comparisons would help clarify how IPBench differs in complexity, task format, or real-world grounding.

- Third, although LLMScore is a promising and thoughtfully designed metric, its empirical validation remains limited. The authors conduct a small-scale human evaluation on two generation tasks, comparing LLMScore with three human raters using correlation metrics (Kendall, Spearman, and Pearson). While this shows moderate alignment, the study is narrow in scope, covering only a subset of tasks and omitting key indicators like inter-rater agreement or detailed error analyses. A more extensive human validation across diverse task types would strengthen the credibility and generalizability of the metric.

**Questions:**

- Could you clarify the rationale behind your model selection and exclusions? While the paper evaluates 17 models, it is unclear why certain widely used legal-domain baselines were omitted. Clarifying your selection criteria like model size, instruction tuning availability and performance cutoffs would help contextualize your results.

- Have you considered analyzing CoT prompting effectiveness by task type or complexity level? You find that CoT prompting slightly degrades performance overall, but it is applied uniformly across all tasks. Since IPBench is structured around cognitive complexity , have you tested whether CoT helps on higher-level tasks, e.g., legal reasoning, generation, but hurts on lower-level ones, e.g., concept recall?

- Can you elaborate on the validation of LLMScore beyond the two generative tasks? The correlation analysis between LLMScore and human judgments is helpful, but it covers only two tasks and three models. Do you plan to extend this to more tasks? Also, could you report inter-rater agreement among the human evaluators?

---

> ### Author Response · Authors · 2025-11-18
> **Response by Authors (Part 1)**
>
> **Dear Reviewer EVqu:**
>
> We sincerely appreciate the time and effort you have invested in providing suggestions for our paper. We are grateful for your positive feedback on our paper's novelty, IP mechanisms coverage, taxonomy, LLMScore and presentation. We are very encouraged by your positive evaluation and your appreciation for the contributions of the IPBench benchmark. We will carefully address your questions below.
>
> > **Question 1: Model selection principle.**
>
> **Answer 1:**
>
> Thank you for your insightful comment. We agree that clarifying the criteria for selecting and excluding evaluated models will help readers better understand the scope of our benchmark. As shown in **Table 20 (Page 25)**, we provide a comprehensive summary for model’s size, context window size and accessing method. We will provide more detailed explanation for model selection below.
>
> -  **Open-sourced and commercial close-sourced models**: We include only models with publicly available APIs (GPT series and DeepSeek Series) or checkpoints that allow reproducible evaluation (Qwen Series, Llama Series and so on).
> - **Domain specialization**: Our IPBench is a vertical-domain benchmark for intellectual property, a domain that inherently integrates both technical and legal aspects. Therefore, we select the only publicly available IP-oriented LLM, MoZi. Unfortunately, other IP-focused foundation models such as PatentGPT are not open-sourced. To mitigate this issue and provide a more comprehensive evaluation, we additionally included open-source legal LLMs such as DISC-LawLLM and HanFei, since IPBench contains tasks with legal attributes, enabling comparisons between domain-specific and general foundation models. The performance of legal-domain LLMs is not our focus; instead, we emphasize that the IP domain requires its own powerful, domain-specific LLMs. We agree that legal and IP-oriented LLMs share certain characteristics, and analyzing their differences will be pursued as our future work to develop more capable, practice-ready IP-oriented LLMs.
> - **Model size and representativeness**: We aim to cover a wide range of model scales (e.g., 7–8B, 27–32B, 70–72B, and 671B) and diverse architecture families, including the GPT, Gemma, and Qwen series, among others.
> - **Chat model and reasoning model**: Our IPBench is rooted in Webb’s DOK cognitive theory, encompassing memory, classification, and generation tasks, as well as logic and reasoning tasks. Accordingly, we select two main types of models: chat models and large reasoning models, such as the powerful DeepSeek-R1 and QwQ.
>
> We believe that our IPBench could serve as a guide for the development of IP-oriented LLMs, similar to how LegalBench serves law-oriented LLMs. We will also include newly released models from the community, both general and IP-oriented, in future work to further support the development of the domain.

---

> ### Author Response · Authors · 2025-11-18
> **Response by Authors (Part 2)**
>
> **Dear Reviewer EVqu:**
>
> > **Question 2: More analysis on the efficacy of CoT prompting.**
>
> **Answer 2:**
>
> Thank you for your insightful comment. We provide the performance differences between CoT prompting and zero-shot prompting, as shown in the table below (CoT minus zero-shot). We observe that for the majority of level-1 tasks, and especially for legal memory tasks, the use of CoT prompting results in a decline in model performance. Notably, for Task 2-3, which requires models to calculate compensation in IP cases, most models exhibit substantial improvement under CoT prompting. In particular, the IP-oriented LLM MoZi achieves the largest performance gain.
>
> This phenomenon aligns with our statements in **lines 376–377**, which emphasize the need for future models to integrate both intuitive (System 1) and analytical (System 2) capabilities in the IP domain. We appreciate the reviewer’s valuable feedback. We will incorporate these additional analytical results into the appendix of our revised manuscript, as they offer useful insights for guiding the development of IP domain–specific LLMs.
>
> | Model           | OA    | 1-1   | 1-2   | 1-3   | 1-4   | 1-6   | 1-7   | 2-1   | 2-2   | 2-3   | 2-4   | 2-5   | 3-1   | 3-2   | 3-3   | 3-4   | 3-5   | 4-3   |
> |-----------------|-------|-------|-------|-------|-------|-------|-------|-------|-------|-------|-------|-------|-------|-------|-------|-------|-------|-------|
> | GPT-4o-mini     | -0.6  | 0     | -1.6  | -2.2  | -1.8  | 1.1   | -0.2  | 1.2   | -1.4  | 2.5   | -2    | -3    | -1.7  | -3.9  | -0.5  | 0.3   | 0.9   | 0.6   |
> | Qwen2.5-7B-it   | -0.4  | -3.4  | -0.4  | -2    | -1    | -1.1  | 0.9   | -1.4  | -0.2  | 1     | -0.7  | 2.7   | -1    | -1    | -0.7  | 0.3   | 5.4   | 0     |
> | Llama3.1-8B-it  | -0.4  | -6.3  | -6.4  | -0.6  | -1.1  | 0.5   | -0.9  | 1.8   | 3.1   | 1.2   | -5    | -4    | 2.3   | -2.3  | 4.2   | -4.2  | 1.9   | 1.8   |
> | Gemma-2-9B-it   | -0.6  | -4.6  | -6    | -7.6  | 4.7   | -4.7  | -3.7  | -0.2  | 1.2   | 1     | 0.7   | -4    | 0.3   | 1     | 6     | -2.5  | 0     | 3     |
> | Mistral-7B-it   | -0.3  | 1     | -0.6  | 3     | 2.4   | 3.1   | 0     | -1.2  | -1.4  | 1.9   | 0     | -1.2  | 0.7   | 5.2   | -2.9  | 1.4   | -7.6  | -6.6  |
> | MoZi-Qwen       | -4.7  | -0.8  | -3.4  | -5    | -0.8  | -8.1  | -2.4  | -5.4  | -5.6  | 3.5   | -1    | -4    | -9.7  | -1.6  | -6.9  | -8    | 0     | -13   |
> | DISC-LawLLM     | -14.6 | -13.6 | -8    | -19   | -20.8 | -11.6 | -10.6 | -15.4 | -25.6 | -5.4  | -27.9 | -2    | -27   | -19.5 | -10.7 | -32.3 | 0     | -4.8  |
> | Hanfei          | -9.9  | -21   | -17.9 | -19.4 | -11.3 | -9.1  | -20.4 | -2    | -16   | -12.3 | -14.9 | -24.9 | -8.3  | -1.9  | 5.7   | -17   | 0     | 5.8   |
>
> > **Question 3: More details about LLMScore.**
>
> **Answer 3:**
>
> Thank you for your detailed consideration. Our LLMScore is primarily designed for IP-related generation tasks, providing fine-grained and interpretable evaluation based on human-crafted criteria. We will continue to introduce additional tasks as new IP-related generation tasks emerge in future work.
>
> We provide the **Krippendorff’s Alpha between human experts’ judgement** in Table below. The results demonstrate the high consistency among our three human experts. And we will assess the compliance of publicly available annotation data. If permitted, we will make the human experts’ annotations and human judgments on generation tasks publicly available as open-source upon acceptance.
>
> | Metric               | Task 4-1 | Task 4-2 |
> |----------------------|----------|----------|
> | Krippendorff's Alpha | 0.708    | 0.948    |
>
> > **Question 4: Provide more discussion of broader legal benchmarks.**
>
> **Answer 4:**
>
> Thank you for your insightful comment. Our IPBench is intentionally designed for the IP domain, which contains distinct legal attributes and task structures. In the revised manuscript, we will explicitly cite LexGLUE, LegalBench, LawBench, and other relevant legal benchmarks to clearly delineate the differences between general legal tasks and the specialised IP domain, thereby improving clarity and positioning within the broader benchmarking landscape.
>
> ---
>
> Thank you again for your detailed and careful review. We look forward to your response.

---

> > ### Comment · Reviewer_EVqu · 2025-11-26
> >
> > I thank the authors for their comments and decide to stick with my score.

---

### Official Review · Reviewer_4pw2 · 2025-10-31

**Soundness:** 3
**Presentation:** 3
**Contribution:** 2
**Rating:** 6
**Confidence:** 3

**Summary:**

The paper introduces 20 Intellectual Property (IP)-related tasks designed to benchmark LLMs. The authors evaluate 17 LLMs and provide insights on:
* the underperformance of IP-oriented models compared to general-purpose ones,
* differences in performance between English and Chinese evaluations,
* the effects of zero-shot, few-shot, and chain-of-thought (CoT) prompts,,
* the use of LLM-as-a-judge for generative task evaluation, and
* the poor performance of models on patent classification tasks.

**Strengths:**

* The task construction and evaluation follow community standards. While this makes the work sound, it also means it is not particularly original (aside from focusing on the IP domain).
* The range of tasks appears reasonably broad (within IP), at least from my limited familiarity with IP benchmarks.
* The presentation is clear and well-structured.

**Weaknesses:**

* Significance: It’s difficult for me to assess the overall significance of this benchmark. The IP domain is relatively niche, but it’s also one where LLMs could have substantial impact.
* None of the findings are particularly surprising, something the authors themselves acknowledge. (For example: domain-specific models tend to underperform generalist models; the relative strength of chat vs. reasoning models depends on the task; model performance on English/Chinese correlates with the model’s primary pre-training language; more few-shot examples generally help; and CoT can underperform depending on the task.) On the positive side, the paper confirms that intuitions from other domains transfer to IP-related tasks.
* Appendix E does not detail how each specific task was constructed.
* The paper provides little context for interpreting model accuracy. What accuracy would be considered acceptable for practical deployment of LLMs in these tasks? What is the human inter-annotator agreement for each?
* The results for LLM-as-a-judge show relatively low correlation with human judgments, even though they outperform alternative approaches.

**Questions:**

* A few tasks exceed 90% accuracy and several exceed 80%. Does this imply that LLMs are suitable for practical use in those tasks?
* Please provide more details on the cosine similarity–based quality filtering process.
* It is unclear how human judgment is evaluated in Table 6, was it binary (correct/incorrect) or based on a rubric score?
* It would be interesting to include the rank correlation between tasks.
* Please include the LLM-as-a-judge prompts in the Appendix.
* Please include the cost of evaluating each model on IPBench, at least for those accessed via API (e.g., GPT-4o).
* Each example seems to have been checked by multiple annotators. Releasing these individual annotations (not just the final consensus) could make for a valuable dataset for research on LLM-based annotation.
* The use of Webb’s Depth of Knowledge (DOK) framework doesn’t add much value to me, though it’s not necessarily a weakness either.
* The acronyms IPC/CPC are not introduced.

---

> ### Author Response · Authors · 2025-11-18
> **Response by Authors (Part 1)**
>
> **Dear Reviewer 4pw2:**
>
> We sincerely appreciate the time and effort you have invested in providing suggestions for our paper. We are grateful for your positive feedback on our paper's task construction and evaluation process, task coverage and the presentation. We are very encouraged by your positive evaluation and your appreciation for the contributions of the IPBench benchmark. We will carefully address your questions below.
>
> > **Question 1: Does high accuracy imply that LLMs are suitable for practical use in those tasks?**
>
> **Answer 1:**
>
> Thank you for your careful consideration of models’ performance on IPBench. High accuracy on certain tasks does not necessarily indicate that LLMs are practically ready for real-world IP applications. We have made substantial efforts to increase the difficulty and realism of our data, resulting in the best-performing model, DeepSeek-V3, achieving only a 75.8 average overall score. Our IPBench is rooted in Webb’s DOK cognitive theory, which considers memory skills as the foundational basis for processing domain-specific practices. Although these memory tasks achieve high accuracy, their performance contrasts sharply with that on logical reasoning and patent classification tasks (even 0% exact match), which require higher-order reasoning abilities. Compared with real-world IP tasks, the tasks in our benchmark are designed at a finer granularity, allowing for a more detailed and precise evaluation of LLMs’ capabilities across different dimensions of IP. These results  demonstrate that these LLMs **still lack much of the domain-specific knowledge and practical experience required for real-world IP scenarios**. We believe that our IPBench could serve as a guide for the development of IP-oriented LLMs, similar to how LegalBench serves law-oriented LLMs.
>
> > **Question 2: Please provide more details on the cosine similarity–based quality filtering process.**
>
> **Answer 2:**
>
> Thank you for your detailed consideration. We will provide more information on this process in our revision. As mentioned **in lines 223–229**, we provide expert annotators with strict guidelines, apply a similarity-based quality filtering procedure to prevent data duplication, and ensure the overall quality of the content through human expert review.
>
> To further clarify the similarity-based filtering: we use the BGE-M3 model to compute dense semantic representations of each question–options pair. These embeddings are indexed using FAISS to efficiently handle large-scale vector storage and retrieval. For every new data, we compute its cosine similarity with previously accepted data. If the similarity exceeds a predefined threshold (0.8), the data is considered semantically redundant and filtered out. Ultimately, we obtained a series of low-redundancy clean data, ensuring that the final dataset maintains semantic diversity while avoiding approximate duplicate content. We will refine this section in the revision to improve clarity.
>
> > **Question 3: It is unclear how human judgment is evaluated in Table 6.**
>
> **Answer 3:**
>
> Thank you for your detailed consideration of our human judgment settings. For the two generation tasks, we employ three human experts with experience in the IP domain. Each expert is provided with the gold-standard target abstracts and claims, together with evaluation rubrics aligned with our LLMScore framework. We report the **Krippendorff’s Alpha among the three human experts** in the table below, and the results indicate a high level of consistency in their judgments.
>
> | Metric               | Task 4-1 | Task 4-2 |
> |----------------------|----------|----------|
> | Krippendorff's Alpha | 0.708    | 0.948    |
>
> > **Question 4: Include the rank correlation between tasks.**
>
> **Answer 4:**
>
> Thank you for the helpful suggestion. We provide the rank correlation between tasks for each model, as shown in the table below. The correlation analysis shows that these models exhibit strong consistency in their rankings, highlighting the effectiveness of our evaluation. We will include it in the revision to enhance clarity and rigor.
>
> | Metric   | OA      | Level 1 | Level 2 | Level 3 |
> |----------|---------|---------|---------|---------|
> | Kendall  | 0.6901  | 0.7983  | 0.7003  | 0.5529  |
> | Spearman | 0.8420  | 0.9241  | 0.8553  | 0.7226  |
>
> Notably, we provide a model performance heatmap in the initial version (**Figure 5, Page 28**), which can also serve as an additional illustration of the models’ relative performance in terms of ranking.

---

> ### Author Response · Authors · 2025-11-18
> **Response by Authors (Part 2)**
>
> **Dear Reviewer 4pw2:**
>
> > **Question 5: Include the LLM-as-a-judge prompts in the Appendix.**
>
> **Answer 5:**
>
> Thank you for your helpful suggestion. We provide a brief version of LLMScore prompt for Task 4-2 here due to the length constraints and will include the complete version in the revision.
>
> **Brief Version of Prompt**
>
> ---
> You are an experienced intellectual property expert specializing in assessing the quality of patent dependent claims. Please objectively evaluate the dependent claims written by the AI assistant, acting as a fair and rigorous judge. When evaluating, you should score it based on the following five dimensions:
> 1. **Accuracy** for the task of generating dependent claims based on a given independent claim evaluates the linguistic correctness, structural fidelity, and legal coherence of the generated claims. It emphasizes the following key aspects: (Omit)
> 2. **Relevance** for the task of generating dependent claims based on a given independent claim evaluates whether the content of the dependent claims remains aligned with the technical subject matter disclosed in the independent claim and the overall invention. This ensures that the claim set does not deviate from the invention’s disclosed scope. This metric specifically emphasizes: (Omit)
> 3. **Completeness** for the task of generating dependent claims based on a given independent claim assesses whether the dependent claims meaningfully expand upon the independent claim by introducing additional technical features or specific limitations. It ensures that each claim contributes to a more comprehensive protection of the invention. This metric focuses on the following key aspects: (Omit)
> 4. **Consistency** for the task of generating dependent claims based on a given independent claim assesses whether the generated claims logically and legally align with their referenced claims and maintain internal coherence within the claim set. It includes the following key evaluation aspects: (Omit)
> 5. **Language-Style** for the task of generating dependent claims based on a given independent claim evaluates the linguistic quality and drafting style of the claims, focusing on clarity, precision, and conciseness in accordance with standard patent drafting conventions. This includes the following key evaluation aspects: (Omit)
>
> We will provide the following materials: the patent independent claims, the ground-truth dependent claims and the dependent claims written by the AI assistant based on independent claims. When starting your evaluation, you need to follow the reasoning steps below:
> 1. Compare the AI assistant’s dependent claims with the ground-truth dependent claims, pointing out the shortcomings of the AI assistant’s answer and explaining them in detail.
> 2. Evaluate the AI assistant’s dependent claims according to the dimensions mentioned above, giving a score from 1 to 10 for each dimension.
> 3. Based on the scores for each dimension, calculate the overall score for the dependent claims written by the AI assistant (1-10 points).
> 4. Your scoring should be as strict as possible, and you must follow the scoring rules below: The higher the quality of the response, the higher the score.
>
> **Scoring Standards for Dependent Claim Generation:**
> (Omit)
>
> ---

---

> ### Author Response · Authors · 2025-11-18
> **Response by Authors (Part 3)**
>
> **Dear Reviewer 4pw2:**
>
> > **Question 6: Include the cost of evaluating each model (accessed via API) on IPBench.**
>
> **Answer 6:**
>
> Thank you for the helpful suggestion. We provide the costs of the evaluated models on IPBench, which are accessed via API, as shown in the table below. We will include it in the revision to enhance clarity and rigor.
>
> | Model         | GPT-4o   | GPT-4o-mini | DeepSeek-V3 | DeepSeek-R1 |
> |---------------|----------|-------------|-------------|-------------|
> | Cost          | 43.902 $ | 2.001 $     | 2.618 $    | 21.437 $   |
>
> > **Question 7: Release annotations of individuals.**
>
> **Answer 7:**
>
> Thank you for your helpful suggestion. We will assess the compliance of publicly available annotation data. If permitted, we will make the human experts’ annotations and human judgments on generation tasks publicly available as open-source upon acceptance.
>
> > **Question 8: Role of Webb’s DOK framework.**
>
> **Answer 8:**
>
> Thank you for the comment. We chose to adopt the DOK framework because it provides a systematic and principled way to capture abilities ranging from foundational knowledge to higher-order reasoning, which is particularly relevant in the IP domain. Workflows in this field naturally reflect the progression described in the DOK hierarchy: law and patent retrieval (remembering) → novelty assessment (understanding) → claim drafting (applying) → infringement analysis (reasoning). Accordingly, this framework aligns closely with the skill requirements of real-world IP tasks, offering both rigor and clarity in our benchmark’s task taxonomy and making it more representative of actual professional workflows and human cognitive capabilities.
>
> > **Question 9: The acronyms IPC/CPC are not introduced.**
>
> **Answer 9:**
>
> Thank you for your detailed consideration. Although we have already introduced the acronyms IPC/CPC in **line 258**, we will revise their first occurrence, as this may help enhance clarity.
>
> ---
>
> Thank you again for your detailed and careful review. We look forward to your response.

---

### Official Review · Reviewer_TeDk · 2025-11-02

**Soundness:** 3
**Presentation:** 2
**Contribution:** 3
**Rating:** 4
**Confidence:** 4

**Summary:**

This paper introduces IPBench, a large-scale, bilingual, and comprehensive benchmark tailored for evaluating LLMs on real-world intellectual property tasks. The benchmark is structured around a hierarchical cognitive taxonomy inspired by educational theory, spanning 20 distinct tasks across 8 IP mechanisms and totaling over 10,000 data points. The authors assess 17 prominent LLMs under various evaluation paradigms, providing granular performance and error analyses. They also propose LLMScore, an LLM-as-a-judge metric for evaluating generative output.

**Strengths:**

1.  The benchmark covers a broader IP tasks, including trademarks, trade secrets, and legal reasoning, than prior datasets, providing a comprehensive benchmark in the vertical domain.
2. The authors provide fine-grained error analysis, giving insights into model behavior and failure modes, such as dominance of reasoning or hallucination errors.
3. Introduction of LLMScore, an LLM-as-a-judge metric validated against human judgments and correlated via multiple statistical metrics.

**Weaknesses:**

1. Limited Discussion of Data Coverage Bias: While the dataset construction is described as broad and rigorous, and the distribution plots in Figure 2 and Tables 10–12 show diversity, there remains a heavy numerical dominance of patent tasks (67% of datapoints, per Figure 2b), which may reinforce the very bias the paper claims to correct.
2. The benchmark's data selection in the IP domain (US/China only) contradicts its goal as a general evaluation tool. IP law is not universal; it is highly nation-dependent. Therefore, an IP benchmark lacking broad jurisdictional diversity has limited generalizability and utility.
3. Only one open, IP-specialized LLM is evaluated (MoZi), due to ecosystem limitations. This restricts the ability to judge the gaps and needs between domain-adapted and general foundation models (and also proves the previous weakness))

**Questions:**

Please check the weakness.

---

> ### Author Response · Authors · 2025-11-18
> **Response by Authors**
>
> **Dear Reviewer TeDk:**
>
> We sincerely appreciate the time and effort you have devoted to providing constructive suggestions for our paper. We are grateful for your positive feedback on our extensive coverage of IP mechanisms, our fine-grained error analysis, and our LLM-as-a-judge metric, LLMScore. Below, we respectfully present several points for your consideration regarding our work, and we kindly request your thoughtful reconsideration of our revised submission.
>
> > **Question 1: Limited Discussion of Data Coverage Bias.**
>
> **Answer 1:**
>
> We acknowledge the phenomenon highlighted by the reviewer: patent-related data indeed constitutes a substantial portion of our dataset (approximately 67%). However, this distribution reflects, to a meaningful extent, the actual structural composition of the IP domain, where patent documents dominate publicly available corpora from institutions such as the USPTO and CNIPA. We therefore view the higher proportion of patent-related tasks not as a source of bias, but rather as **a natural reflection of real-world IP data distributions**.
>
> Moreover, our task-oriented taxonomy is explicitly designed to mitigate potential biases across mechanisms; for instance, Task 2-5 primarily focuses on trade secret types. To further ensure transparency, we provide detailed statistics for each IP mechanism across all tasks, as presented in **Table 10 and Table 11 (Page 20)**.
>
> > **Question 2: The benchmark's data selection in the IP domain (US/China only) contradicts its goal as a general evaluation tool.**
>
> **Answer 2:**
>
> We appreciate the reviewer’s insightful comment and fully acknowledge the point that the IP domain is highly nation-dependent. Indeed, the current version of IPBench primarily focuses on the U.S.\ and China. This design choice was **intentional**: these two jurisdictions together account for the majority of global IP filings (over 65%, according to WIPO IP Indicators 2025 [2]) and represent **two contrasting legal traditions**—common law and civil law—which allows us to capture a broad and representative spectrum of IP reasoning patterns.
>
> We also explicitly acknowledge this limitation in both the **Limitations (lines 680-685)** and **More Discussion (lines 1422-1425)** sections, where we note that IPBench is **extensible** and can be expanded to incorporate additional jurisdictions such as the EU, Japan, and Korea to support richer cross-jurisdictional evaluation.
>
> Therefore, we view the current scope as a pragmatic and representative first version, rather than a contradiction to the general benchmarking purpose.
>
> > **Question 3: Only one open, IP-specialized LLM is evaluated (MoZi), due to ecosystem limitations.**
>
> **Answer 3:**
>
> We thank the reviewer for highlighting in their comment the reason why we use only one MoZi, namely **ecosystem limitations**. As we explicitly noted in the Limitations section (**lines 691-695**), MoZi is currently the only publicly available IP-oriented LLM, while other IP-focused foundation models such as PatentGPT [1] are not open-sourced. To mitigate this constraint and provide a more comprehensive evaluation, we additionally included open-source legal LLMs such as DISC-LawLLM and HanFei, given that IPBench contains tasks with legal attributes. This allows us to compare domain-specific IP models with broader legal foundation models. Beyond this, we believe that IPBench can serve as a valuable guide for future development of IP-oriented LLMs, analogous to how benchmarks such as GPQA inform the progress of general-domain LLMs.
>
> We understand that the reviewer may have overlooked some explicit content in our responses to Question 2 and Question 3, in part due to the length and detail of the manuscript. Importantly, we regard these limitations not only as current constraints but also as meaningful opportunities to extend the scope, depth, and impact of IPBench in future iterations.
>
> We kindly request your thoughtful reconsideration of our submission, and we look forward to your response.
>
> ---
>
> **Reference**
>
> [1] Bai Z, Zhang R, Chen L, et al. Patentgpt: A large language model for intellectual property[J]. arXiv preprint arXiv:2404.18255, 2024.
>
> [2] WIPO. (2025). World Intellectual Property Indicators 2025. World Intellectual Property Organization,. https://doi.org/10.34667/TIND.58959

---

> > ### Comment · Reviewer_TeDk · 2025-11-20
> >
> > I appreciate the authors' clarification on Question 3 regarding the limited availability of open-source IP LLMs. I had previously overlooked this discussion because it was in the Appendix, and the Appendix is quite long.
> >
> > Thanks to the authors' hard work, all my concerns have been addressed. I have increased my score.

---

> > > ### Author Response · Authors · 2025-11-21
> > > **Sincerely Appreciate Reviewer TeDk's Positive Feedback**
> > >
> > > We sincerely thank you for taking the time to review our paper. Your continued engagement and the valuable concerns you've highlighted are greatly appreciated. We will carefully revise the final manuscript based on the constructive feedback from you and the other reviewers to ensure a comprehensive and improved presentation.

---

### Note · Authors · 2025-12-06

I have read and agree with the venue's withdrawal policy on behalf of myself and my co-authors.